# Beyond Anticoagulation: A Comprehensive Review of Non-Vitamin K Oral Anticoagulants (NOACs) in Inflammation and Protease-Activated Receptor Signaling

**DOI:** 10.3390/ijms25168727

**Published:** 2024-08-10

**Authors:** Shirin Jannati, Rajashree Patnaik, Yajnavalka Banerjee

**Affiliations:** 1Yajnavalkaa Banerrji Research Group, College of Medicine and Health Sciences, Mohammed Bin Rashid University of Medicine and Health Sciences (MBRU), Dubai Health, Dubai P.O. Box 505055, United Arab Emirates; shirin.jannati@students.mbru.ac.ae (S.J.); rajashree.patnaik@mbru.ac.ae (R.P.); 2Centre for Medical Education, University of Dundee, Dundee DD1 4HN, UK

**Keywords:** NOAC, anti-inflammation, protease-activated-receptor signaling, factor Xa, thrombin

## Abstract

Non-vitamin K oral anticoagulants (NOACs) have revolutionized anticoagulant therapy, offering improved safety and efficacy over traditional agents like warfarin. This review comprehensively examines the dual roles of NOACs—apixaban, rivaroxaban, edoxaban, and dabigatran—not only as anticoagulants, but also as modulators of inflammation via protease-activated receptor (PAR) signaling. We highlight the unique pharmacotherapeutic properties of each NOAC, supported by key clinical trials demonstrating their effectiveness in preventing thromboembolic events. Beyond their established anticoagulant roles, emerging research suggests that NOACs influence inflammation through PAR signaling pathways, implicating factors such as factor Xa (FXa) and thrombin in the modulation of inflammatory responses. This review synthesizes current evidence on the anti-inflammatory potential of NOACs, exploring their impact on inflammatory markers and conditions like atherosclerosis and diabetes. By delineating the mechanisms by which NOACs mediate anti-inflammatory effects, this work aims to expand their therapeutic utility, offering new perspectives for managing inflammatory diseases. Our findings underscore the broader clinical implications of NOACs, advocating for their consideration in therapeutic strategies aimed at addressing inflammation-related pathologies. This comprehensive synthesis not only enhances understanding of NOACs’ multifaceted roles, but also paves the way for future research and clinical applications in inflammation and cardiovascular health.

## 1. Introduction

### 1.1. Mechanism and Regulation of the Blood Coagulation Cascade

Blood coagulation is an innate response to vascular injury that results from a series of amplified reactions in which zymogens of serine proteases circulating in the plasma are sequentially activated by limited proteolysis, leading to the formation of blood clots, preventing loss of blood [1]. It is initiated through the extrinsic pathway [1]. As shown in Figure 1, membrane-bound tissue factor (TF), exposed during vascular injury, interacts with factor VIIa (FVIIa), which pre-exists in the plasma (at 1–2% of total factor VII) [2], and forms the extrinsic tenase complex.

This complex activates factor X (FX) to FXa. In association with its cofactor factor Va (FVa), FXa performs a proteolytic cleavage of prothrombin to thrombin. Thrombin cleaves fibrinogen to fibrin, promoting the formation of a fibrin clot, and activates platelets for inclusion in the clot. The TF·FVIIa complex can also activate factor IX (FIX) to factor IXa (FIXa), thus facilitating the propagation of the coagulation cascade through the intrinsic pathway [3]. The coagulation cascade is under tight regulation. Regulation of this cascade involves several natural anticoagulants, such as antithrombin, protein C, and protein S, which inhibit various clotting factors, as well as fibrinolytic mechanisms that degrade fibrin clots, maintaining a delicate balance between clot formation and dissolution. Any imbalance in its regulation could lead to either unclottable blood, resulting in excessive bleeding during injuries, or unwanted clot formation, resulting in death and debilitation due to vascular occlusion with the consequence of myocardial infarction, stroke, pulmonary embolism, or venous thrombosis [4].

### 1.2. Anticoagulants and Non-Vitamin K Oral Anticoagulants (NOACs)

Anticoagulants are pivotal for the prevention and treatment of thromboembolic disorders. The community pharmacy oral anticoagulant safety audit in the UK found that the three most frequently prescribed anticoagulants were apixaban (41.4%), rivaroxaban (25.2%), and warfarin (20.8%) [5]. Another study analyzing trends in oral anticoagulant use among 436,864 patients with atrial fibrillation (AF) in the US found that overall anticoagulation rates increased from 56.3% in 2011 to 64.7% in 2020, suggesting that a majority of high-risk patients with atrial fibrillation in the US are receiving anticoagulant therapy [6].

For decades, warfarin, heparin, and its low-molecular-weight derivatives such as enoxaparin have held the reins as the mainstay of oral anticoagulant therapy, for the management of thromboembolic events. However, despite their effectiveness, these anticoagulants have pitfalls that need to be considered. Heparin can lead to heparin-induced thrombocytopenia [7], a potentially serious condition where the body attacks its own platelets, increasing the risk of blood clots. Warfarin, known for its narrow therapeutic window and inter-individual variability within patient response, requires careful monitoring to avoid bleeding complications [8,9]. Additionally, warfarin interacts with many drugs and foods, necessitating close attention to prevent adverse effects [10,11]. Enoxaparin, while effective, can also lead to bleeding complications, especially if not dosed correctly in obese patients [12]. These pitfalls have fueled the quest for safer alternatives [13]. This pursuit culminated in the emergence of NOACs, also known as direct oral anticoagulants (DOACs), fundamentally transforming the anticoagulation landscape [14].

In contrast to warfarin’s method of indirectly inhibiting vitamin K-dependent clotting factors, NOACs directly target specific enzymes within the coagulation cascade (either FXa or thrombin), offering a series of significant advantages [15]. This direct approach enables predictable dosing, eliminating the need for the routine monitoring that is mandatory with warfarin therapy, thus simplifying treatment protocols and enhancing medication compliance [16,17]. Furthermore, most NOACs have minimal food interactions, providing patients with a much-appreciated dietary freedom. In comparison to warfarin, which has a wide range of drug interactions, NOACs have a notably reduced profile of interactions with other medications [18]. Additionally, certain NOACs come with the option of specific antidotes for rapid reversal in the event of bleeding complications, offering an extra layer of safety in emergency situations [19]. NOACs targeting factor Xa include the drugs rivaroxaban, apixaban, edoxaban, and betrixaban. These drugs work by inhibiting the cleavage of prothrombin to thrombin, a key step in the coagulation cascade (Figure 1). Thrombin-targeting NOACs include dabigatran, which directly binds to both clot-bound and free thrombin, to prevent blood clotting (Figure 1). It is important to note that NOACs are inhibitors of coagulation factors rather than inactivators. They compete with substrates for binding at the active site of the target protease, and that binding is reversible [20]. This reversible binding mechanism of NOACs provides the advantage of a predictable and controlled anticoagulant effect, which can be easily managed in the event of bleeding or when surgical procedures are necessary, as the anticoagulation can be quickly reversed [21].

Beyond their critical roles in blood coagulation, FXa and thrombin orchestrate a myriad of physiological processes, predominantly mediated through a family of G-protein coupled receptors protease-activated receptors (PARs). These include influencing inflammation, cell proliferation, and tissue repair mechanisms, highlighting their importance in both hemostasis and broader cellular and systemic functions [22]. In line with this, recent research has identified potential off-label uses of NOACs beyond traditional anticoagulation, particularly highlighting their anti-inflammatory and anti-atherosclerotic effects [23]. The molecular underpinnings of these off-label therapeutic or “Pleiotropic Effects” effects seem to reside in the interactions between NOACs, and the coagulation protease they bind to, with the concomitant effect of this binding on the activation of PARs. PARs, particularly PAR-1 and PAR-2, are known to play crucial roles in the pathophysiology of inflammation and atherosclerosis. The activation of these receptors’ triggers signaling pathways implicated in vascular inflammation, endothelial dysfunction, and the promotion of atherogenesis, also augmenting the inflammatory milieu in specific diseases, such as osteoarthritis (OA). For example, according to Ito et al. [24], rivaroxaban reduced inflammation and atherosclerosis by reducing coagulation and possibly altering PAR-2 signaling. It also significantly reduced the atherosclerotic area in the aorta of ApoE−/− mice fed a high-fat diet and reduced FXa-induced macrophage autophagy inhibition. This study suggests that rivaroxaban may help prevent atherosclerosis by stopping FXa-PAR2-mediated macrophage autophagy and inflammasome activity, in addition to its ability to stop blood clots. Building on this premise, this review critically evaluates the dual roles of FXa and thrombin as both coagulants and mediators of inflammation via PAR signaling, alongside the emerging evidence supporting the anti-inflammatory effects of NOACs, focusing on rivaroxaban, apixaban, edoxaban, and dabigatran.

By synthesizing current research, it posits that NOACs, traditionally used for their anticoagulant properties, may also modulate inflammation through PAR-signaling pathways, suggesting a broader clinical utility in managing inflammatory conditions and atherosclerosis. This concise exploration not only provides a fresh perspective on NOACs beyond anticoagulation, but also underscores the potential for innovative therapeutic strategies in disease treatment. The review aims to inform and inspire ongoing research and clinical practice, making it a significant contribution to the existing body of literature.

## 2. Exploring NOACs: From Pharmacokinetics to Clinical Efficacy and Pivotal Trials

Building on our exploration of NOACs, Table 1 offers a comprehensive summary of the NOACs rivaroxaban, apixaban, edoxaban, and dabigatran. It covers essential aspects such as pharmacology, the necessity of and approach to laboratory monitoring, guidelines for peri-operative management, and their clinical profiles. Furthermore, the table delves into the advantages and potential challenges associated with each NOAC, providing a well-rounded view that complements our discussion on their impact in clinical practice.

It is pertinent to note that all NOACs underwent evaluation in large-scale, Phase III clinical trials against warfarin as the control arm. Table 2 succinctly outlines the clinical trials under discussion, highlighting the unique aspects and key distinctions of each study. A 2014 meta-analysis by Ruff et al., involving 71,683 participants across the RE-LY, ROCKET AF, ARISTOTLE, and ENGAGE AF–TIMI 48 trials, highlighted the superior efficacy of NOACs over warfarin in reducing stroke or systemic embolic events by 19%, with this benefit primarily driven by a significant decrease in hemorrhagic stroke [25]. Notably, NOACs also demonstrated a marked reduction in all-cause mortality and intracranial hemorrhage, albeit with an observed increase in gastrointestinal bleeding. This pivotal analysis offers a comprehensive evaluation of rivaroxaban, apixaban, edoxaban, and dabigatran, emphasizing their favorable risk–benefit profile, consistent efficacy across diverse patient subgroups, and potential to significantly mitigate the risk of stroke in patients with atrial fibrillation. Additionally, the analysis underscores the importance of NOACs in providing a safer alternative to warfarin, with similar major bleeding rates but reduced instances of intracranial hemorrhage, thus reaffirming their role in the current therapeutic landscape for atrial fibrillation.

### 2.1. Mechanism of Anticoagulation: Inhibition of Factor Xa and Thrombin

*Factor Xa Inhibition*: FXa is a vitamin K-dependent serine protease composed of two amino acid chains connected by a disulfide bridge [26]. The heavy chain contains 303 amino acids, including the active site, while the light chain consists of 139 amino acids [27]. According to Schechter and Berger, the binding sites for FXa inhibitors are located in the S1 and S4 pockets [28]. A schematic representation of FXa protein structural domains showing the location of the binding sites of FXa inhibitors (apixaban, edoxaban, and rivaroxaban) is depicted in Figure 2.

Apixaban is a potent, highly selective, and direct FXa inhibitor [29]. It effectively prevents arterial and venous thrombosis at doses that preserve hemostasis. Enzyme kinetics indicate that apixaban is a direct and competitive inhibitor of free human FXa, independent of antithrombin III. It likely acts as a mixed-type inhibitor of FXa activation of prothrombin in blood [30]. Additionally, apixaban inhibits prothrombinase-bound FXa activity in vitro, reducing prothrombin to thrombin conversion, suggesting its efficacy against both free and prothrombinase-bound FXa [30].Rivaroxaban is a highly potent, competitive, reversible, and direct FXa inhibitor with a Ki of 0.4 nM for free human FXa [31]. In vivo studies show that rivaroxaban effectively prevents arterial and venous thrombosis. Its high affinity for the prothrombinase-bound FXa complex supports its efficacy. Almost complete inhibition of thrombin generation was observed at 80 nM rivaroxaban. Unlike fondaparinux, an antithrombin-dependent FXa inhibitor, rivaroxaban can access the active site within the prothrombinase-bound FXa more effectively. The antithrombotic effect of rivaroxaban primarily results from FXa inhibition, though it may indirectly decrease platelet activation by inhibiting thrombin generation.Edoxaban inhibits FXa with Ki values of 0.561 nM for free human FXa and 2.98 nM for FXa within the prothrombinase complex, showing over 10,000-fold selectivity for FXa [32]. Its inhibition is concentration-dependent and competitive. Edoxaban is a highly selective direct FXa inhibitor, offering significant potency [33]. In vitro studies found edoxaban to be a more potent inhibitor of tissue factor-induced platelet aggregation and clot-bound FXa than fondaparinux, suggesting that direct FXa inhibition provides additional benefits over indirect inhibition [34].

*Thrombin Inhibition*: As discussed previously, thrombin plays a central role in thrombus formation and can be inhibited directly or indirectly by targeting one or two of its three domains: the active site and exosites 1 and 2 [35]. Direct thrombin inhibitors (DTIs) like dabigatran bind directly to thrombin without requiring a cofactor like antithrombin [36]. Dabigatran, a reversible inhibitor, binds to thrombin’s active enzymatic site and inhibits both soluble and fibrin-bound thrombin [37]. Dabigatran is orally available as the prodrug dabigatran etexilate and dose-dependently inhibits human thrombin with a Ki of 4.5 nM. Real-time binding kinetics demonstrate rapid and reversible direct binding of dabigatran to thrombin [38].

### 2.2. Pharmacodynamic and Pharmacokinetic Characteristics of NOACs

As noted previously, in contrast to traditional oral anticoagulants such as warfarin, NOACs exhibit a more stable pharmacokinetic profile, enabling standardized dosing without the need for required routine monitoring. Due to this stability, coupled with demonstrated efficacy, NOACs have established their position as a feasible treatment option. However, there are several significant and clinically pertinent distinctions in the pharmacokinetic and pharmacodynamic profiles of each of the NOACs, which are elaborated upon below and detailed further in Table 1.

Among NOACs, absorption patterns and bioavailability vary. Apixaban is primarily absorbed in the upper small intestine, with absorption progressively decreasing throughout the gastrointestinal tract. Following oral administration, its absolute bioavailability is approximately 50% [39]. Edoxaban, also absorbed primarily in the proximal small intestine, demonstrates high solubility in acidic environments but becomes practically insoluble in basic solutions as it travels through the digestive tract [40]. This characteristic translates to an estimated absolute oral bioavailability of around 60%, with peak concentration (Cmax) occurring within 1 to 2 h of ingestion [41]. Rivaroxaban stands out for its exceptional oral bioavailability, reaching around 80–100% under ideal conditions [42]. Unlike the others, it is primarily absorbed in the stomach and achieves peak concentration (Cmax) 2–4 h after oral administration. Dabigatran is administered as the inactive prodrug, dabigatran etexilate. The acidic environment of the stomach and small intestine aids its dissolution and absorption [43]. However, dabigatran etexilate has a low oral bioavailability of around 6% [44].

Apixaban undergoes O-demethylation, hydroxylation, and sulfation, primarily by CYP3A4 and CYP3A5 enzymes, with minor contributions from CYP1A2, CYP2C8, CYP2C9, CYP2C19, and CYP2J2 [45]. It is also affected by P-glycoprotein (P-gp) and breast cancer resistance protein (BRCP) [46]. Elimination happens through multiple pathways including metabolism, biliary and renal excretion, and direct intestinal excretion [45]. Around 27% is cleared by the kidneys, with the rest eliminated through other routes [45].

Edoxaban’s primary metabolism involves hydrolysis by carboxylesterase 1, with additional contributions from conjugation and oxidation by CYP3A4 and CYP3A5 [47]. Less than 10% of the active drug is cleared through metabolism. Edoxaban interacts with P-gp, and coadministration with strong inhibitors can increase exposure [48]. Elimination occurs via multiple routes, with renal clearance of the unchanged drug accounting for roughly half, and the remainder eliminated through metabolism and bile excretion [47].

Rivaroxaban is metabolized by CYP3A4 and CYP2J2, and undergoes non-CYP mediated hydrolysis. Around 60% of the dose is metabolized [42]. It is a substrate for P-gp and BRCP [42]. Elimination happens through metabolism and renal clearance, with 36% excreted unchanged in urine and another 30% eliminated as metabolites [42]. Importantly, renal elimination of the active metabolite relies heavily on active renal secretion involving both P-gp and BRCP [49].

Dabigatran is unique in that it is not metabolized by CYP450 enzymes [44]. Instead, nonspecific esterase enzymes rapidly and almost completely convert it into its active form via two intermediates [44]. While the liver metabolizes about 20% of dabigatran, the majority is eliminated through the kidneys [50]. Dabigatran etexilate interacts with P-gp, but dabigatran itself does not, suggesting these interactions are limited to absorption [51].

The presence of food does not impact the bioavailability of apixaban in any clinically significant way [52]. For edoxaban, food increases both Cmax and total exposure modestly but does not have clinical significance [53]. With rivaroxaban, the presence or absence of food affects bioavailability only at higher doses (15 mg and 20 mg), where food increases the absorption rate and bioavailability [54]. The presence of food does not impact the extent of dabigatran absorption but can increase the time to achieve Cmax [55].

Pharmacokinetic studies have shown that exposure to apixaban is modestly raised in elderly subjects, likely due to declining renal function, but no dose adjustments are recommended based on age alone [56]. In the case of rivaroxaban, although a study examining the impact of age did demonstrate an increase in total exposure and factor Xa inhibition in elderly patients, this was the result of reduced renal clearance rather than age itself [57]. For dabigatran, exposure is significantly higher in elderly subjects, most likely due to reduced renal function associated with advancing age [44].

The impact of renal impairment on apixaban is less significant compared to other NOACs, and it does not affect Cmax or the relationship between plasma concentration and anti-factor Xa activity, though exposure increases in renal impairment [58]. For edoxaban, exposure increases as renal function declines, and dose reduction is recommended for creatinine clearance of 15–50 mL/min [59]. Rivaroxaban clearance is reduced in renal impairment, leading to increased exposure and higher levels of factor Xa inhibition, with dosing recommendations varying based on renal function and indication [60]. Dabigatran clearance is highly dependent on renal function, and exposure increases proportionally with renal impairment [61].

Mild or moderate hepatic impairment has little or no impact on apixaban pharmacokinetics [45]. For edoxaban, mild to moderate hepatic impairment does not significantly affect its pharmacokinetics or pharmacodynamics [62]. The presence of mild hepatic impairment has limited impact on rivaroxaban pharmacokinetics, but moderate hepatic impairment increases Cmax and exposure, contraindicating its use in patients with moderate to severe hepatic impairment [63]. Moderate hepatic impairment does not significantly impact the pharmacokinetics or pharmacodynamics of dabigatran [64].

### 2.3. Dabigatran’s Clinical Efficacy and Application

Dabigatran etexilate was the inaugural thrombin-targeting NOAC to receive regulatory approval. Table 1 compiles the essential pharmacokinetic attributes and clinical profiles of dabigatran, offering an in-depth examination of its therapeutic properties, efficacy, and safety parameters. A pooled analysis of two trials published (RE-NOVATE and RE-NOVATE II) demonstrated that dabigatran was at least as effective as enoxaparin for thromboprophylaxis after knee or hip replacement [65]. The RE-COVER I and II trials showed that dabigatran was non-inferior to warfarin for the treatment of acute venous thromboembolism (VTE) [66]. In patients with atrial fibrillation at risk of stroke, the 2009 RE-LY study reported that dabigatran was non-inferior to warfarin with respect to the primary efficacy outcome of stroke and systemic embolism in patients with atrial fibrillation (Table 2). Dabigatran received FDA approval for this indication in the subsequent year. More recent studies have compared dabigatran with other anticoagulants. A 2023 multi-center cross-sectional study in Spain assessed the usage of NOACs among elderly patients (≥75 years) with atrial fibrillation, focusing on dabigatran and rivaroxaban, among others [67]. The study, encompassing 500 patients, revealed a nuanced landscape where dabigatran and rivaroxaban not only provided effective stroke prevention but also navigated the complex balance of dosing appropriateness in a demographic challenged by frailty and varied comorbidities. It highlighted that while 76% of NOAC prescriptions adhered to recommended dosing guidelines, a significant portion of the elderly population was still either underdosed or overdosed, underscoring the critical need for meticulous patient assessment to optimize therapeutic outcomes. Particularly, the study illuminated how dabigatran and rivaroxaban, despite their higher costs, offer a compelling blend of efficacy and safety for heart protection in this vulnerable cohort, thereby enriching the therapeutic arsenal against atrial fibrillation-induced cardiovascular events in the elderly. Additionally, a 2018 meta-analysis found no significant difference between 110 mg dabigatran and warfarin on the rate of stroke and systemic embolism, but the 110 mg dabigatran was associated with a lower incidence of bleeding compared to warfarin [68]. However, in a comprehensive 2023 nationwide cohort study, a detailed analysis revealed that dabigatran was associated with more than a twofold increase in the incidence of myocardial infarction compared with the factor Xa inhibitors apixaban and rivaroxaban [69]. Specifically, dabigatran demonstrated a rate of 1.4 myocardial infarction events per 100 person-years, which starkly contrasts with the lower rates of 0.7 events per 100 person-years observed for both apixaban and rivaroxaban. Despite this significant difference in myocardial infarction rates, the all-cause mortality remained comparable across all three drugs, suggesting that the elevated myocardial infarction risk associated with dabigatran does not translate into increased mortality within the study period. The observed higher myocardial infarction rates with dabigatran may stem from its unique pharmacological action. Unlike apixaban, edoxaban, and rivaroxaban, which inhibit FXa, dabigatran directly inhibits thrombin. This distinction in mechanism could influence the coagulation cascade and platelet activation differently, potentially contributing to the increased myocardial infarction risk. Previous research has suggested that dabigatran may enhance platelet activation, a pivotal process in the pathogenesis of myocardial infarction, thereby providing a plausible explanation for the observed elevated risk [70]. This finding is particularly relevant for clinical practice, highlighting the necessity for clinicians to consider the differential risk profiles of NOACs when selecting the most appropriate anticoagulant therapy, especially for patients at higher risk of myocardial infarction. The study underscores the importance of personalized medicine, where the benefits and risks of each anticoagulant are carefully balanced against the individual patient’s risk factors and clinical history to optimize therapeutic outcomes.

Hence, dabigatran emerges as a potent anticoagulant, demonstrating efficacy that matches or surpasses conventional anticoagulants such as warfarin and enoxaparin across diverse clinical settings. Nevertheless, akin to all anticoagulants, dabigatran is associated with a bleeding risk. Its application necessitates a tailored approach, taking into consideration the unique risk profile and clinical attributes of each patient to ensure optimal safety and effectiveness.

### 2.4. Rivaroxaban’s Clinical Efficacy and Application

Rivaroxaban, a direct FXa inhibitor, has been extensively studied and utilized in the prevention and treatment of thromboembolic disorders. The key pharmacokinetic and clinical profiles of rivaroxaban are summarized in Table 1. Following the RECORD I–IV trials conducted between 2008 and 2009, rivaroxaban emerged as a favorable alternative to enoxaparin for VTE prophylaxis after total hip or knee arthroplasty [71]. The EINSTEIN-DVT and EINSTEIN-PE trials, published in 2010 and 2012, further established rivaroxaban’s noninferiority to enoxaparin or warfarin in the treatment of VTE [72]. In the realm of AF, the ROCKET-AF trial of 2011 demonstrated rivaroxaban’s noninferiority to warfarin for stroke prevention in patients with moderate-to-high stroke risk, leading to FDA approval for both stroke prevention in AF and VTE prophylaxis that same year, summarized in Table 2 [73]. Recent studies have continued to evaluate rivaroxaban’s efficacy and safety profile. The PREVENT-HD trial, a multi-center, randomized, double-blind, placebo-controlled study, assessed the efficacy and safety of prophylactic rivaroxaban, although specific results from this trial are not detailed in the search results [74]. A 2021 study examining the comparative risks associated with rivaroxaban and apixaban in patients with atrial fibrillation unveiled significant findings regarding their safety profiles [75]. Conducted on a large cohort of 581,451 Medicare beneficiaries aged 65 years or older, the research revealed that rivaroxaban treatment was associated with a higher incidence of major ischemic events, including ischemic and hemorrhagic strokes, compared to apixaban. Specifically, the adjusted incidence rates for major ischemic events were 8.6 per 1000 person-years for rivaroxaban versus 7.6 per 1000 person-years for apixaban. Additionally, hemorrhagic stroke rates were notably higher for rivaroxaban at 2.5 per 1000 person-years, compared to 1.7 per 1000 person-years for apixaban. A plausible explanation for this increased risk lies in the distinct pharmacokinetic characteristics of rivaroxaban, particularly its once-daily dosing schedule, which results in pronounced peak–trough fluctuations in drug concentration. These fluctuations could lead to periods of under-anticoagulation, heightening the risk of thromboembolic events, and periods of over-anticoagulation, increasing the risk of bleeding events. On the other hand, apixaban’s twice-daily regimen achieves more stable plasma concentrations, potentially reducing the risks of both ischemic and hemorrhagic complications. Furthermore, a comprehensive 2023 randomized controlled trial aimed to determine the optimal duration of rivaroxaban treatment for patients with symptomatic isolated distal deep vein thrombosis (DVT) [76]. This study randomized participants who had already completed an initial six weeks of standard rivaroxaban therapy to either continue with the medication or switch to a placebo for an additional six weeks, with outcomes observed over a 24-month follow-up. The primary efficacy outcome focused on the recurrence of VTE, including progression or recurrence of isolated distal DVT, proximal DVT, symptomatic pulmonary embolism, or fatal pulmonary embolism. The findings highlighted that extending rivaroxaban treatment beyond the initial six weeks significantly reduced the risk of recurrent VTE, mainly through the prevention of recurrent isolated distal DVT, without increasing hemorrhage risks. This significant contribution to our understanding of rivaroxaban’s therapeutic efficacy underscores the value of tailored anticoagulant regimens in preventing VTE recurrence, especially in isolated distal DVT cases. However, the study did not address variations in stroke risk associated with rivaroxaban, as seen in other research. Differences in stroke risk may be attributed to rivaroxaban’s unique pharmacokinetic and pharmacodynamic properties, including its once-daily dosing, which could influence its safety profile. This suggests a need for further investigation to elucidate the mechanisms driving these differences and to refine treatment approaches that balance efficacy with minimizing adverse effects such as stroke. The FDA has also expanded the approved indications for rivaroxaban over the years, including its use in reducing the continued risk of VTE, secondary prevention after acute coronary syndrome, and hospitalized adult patients with an acute medical illness at risk of thromboembolic complications [77]. Additionally, in 2021, rivaroxaban was approved for use in certain pediatric populations to treat and help prevent types of blood clots [78]. Furthermore, in a 2024 OSCAR-UK study, rivaroxaban was found to be a reasonable alternative compared to low-molecular-weight heparins (LMWH) in patients with cancer-associated venous thromboembolism [79]. A 2024 meta-analysis found rivaroxaban to be a better option as compared to warfarin, due to its association with significantly lower risks of stroke and bleeding outcomes in obese patients with atrial fibrillation [80].

Therefore, rivaroxaban remains a key player in the management of thromboembolic diseases, with ongoing research contributing to a nuanced understanding of its role in various clinical scenarios. Its approval for a range of indications reflects its versatility and effectiveness as an anticoagulant, although recent findings highlight the importance of individualized patient assessment when considering its use.

### 2.5. Apixaban’s Clinical Efficacy and Application

Apixaban, the second direct FXa inhibitor to reach the market, boasts a broad clinical profile, as summarized in Table 1. Numerous trials have established its efficacy and safety across various thromboembolic and cardiovascular conditions. In the realm of VTE prevention, the ADVACE-2 trial (2010) demonstrated apixaban’s superiority over enoxaparin for preventing VTE after total knee replacement surgery without increasing bleeding risk [81]. The AMPLIFY trial (2013) confirmed alone a fixed dose of apixaban to be non-inferior to conventional therapy (subcutaneous enoxaparin, followed by warfarin) for treating acute VTE [82]; a 2015 analysis further showed significant reductions in all-cause hospitalization and length of hospital stay [83]. For atrial fibrillation management, the AVERROES trial (2011) established apixaban’s superiority over aspirin in preventing strokes for patients unsuitable for VKA therapy, without significant increased risk of major bleeding [84]. The AVERROES-MRI assessment study (2016) showed a nonsignificant trend towards reduced ischemic stroke and covert embolic-pattern infarction with apixaban compared to aspirin, with no increase in microbleeds [85]. The ARISTOTLE trial (2012) further expounded apixaban’s position by demonstrating its superiority over warfarin in AF patients, translating to a lower risk of stroke and systemic embolism with a better bleeding profile (details in Table 2) [86]. Apixaban’s efficacy extends to cancer-associated thrombosis. The AVERT trial (2019) demonstrated that apixaban significantly reduced VTE for thromboprophylaxis compared to placebo in ambulatory cancer patients who were at intermediate-to-high risk for VTE [(Khorana score, ≥2); *the Khorana score, ranging from 0 to 6, is a risk assessment tool used to predict cancer-associated VTE in chemotherapy patients, with a higher score indicating a greater risk and the potential need for preventative anticoagulation*], and were initiating chemotherapy [87]. The CARAVAGGIO trial (2020) established oral apixaban’s non-inferiority to subcutaneous dalteparin (a LMWH) for treating cancer-associated VTE, again without increased major bleeding [88]. Additionally, a 2020 randomized trial demonstrated oral apixaban’s non-inferiority to subcutaneous enoxaparin in terms of safety and efficacy for thromboprophylaxis after gynecologic cancer surgery, suggesting oral apixaban as a potentially safe, less painful, and more convenient alternative [89]. Recent studies have explored apixaban’s use in special populations. The AXADIA-AFNET 8 (2022), an investigator-initiated PROBE (prospective randomized open blinded end point) outcome assessment trial, showed no apparent differences in safety or efficacy between oral apixaban and VKA therapy in atrial fibrillation patients on chronic hemodialysis [90]. A 2023 Phase 2, open-label trial in children with heart disease requiring thromboprophylaxis found infrequent bleeding and thromboembolic events with both oral apixaban and current standard care (VKA or LMWH). These findings support the use of oral apixaban as an alternative to Standard-of-Care (SOC) thromboprophylaxis in pediatric heart disease [91]. The PREVAPIX-ALL study, a pioneering phase 3, open-label, randomized, controlled trial conducted across 74 pediatric hospitals in nine countries, evaluated the efficacy and safety of oral apixaban for primary thromboprophylaxis in pediatric patients with acute lymphoblastic leukemia or lymphoma [92]. This trial, unique in its assessment of a direct oral anticoagulant for preventing VTE in this high-risk group, randomly assigned participants to receive either SOC without systemic anticoagulation or weight-adjusted twice-daily apixaban during induction chemotherapy. Despite no statistically significant reduction in VTE with apixaban, major and clinically relevant non-major (CRNM) bleeding events were uncommon, though CRNM bleeding, mainly mild epistaxis, was more frequent in the apixaban group. With a total of 512 participants analyzed, the study’s findings suggest that for pediatric patients considered at particularly high risk of thrombosis, apixaban presents an encouraging safety profile, indicating its potential utility in clinical scenarios where the benefits may outweigh the risks of bleeding.

Overall, apixaban emerges as a versatile and effective anticoagulant, with a robust body of evidence supporting its use across a spectrum of thromboembolic and cardiovascular conditions, demonstrating not only its efficacy and safety in adults but also its potential in pediatric populations, marking it as a pivotal option in the evolving landscape of anticoagulation therapy.

### 2.6. Edoxaban’s Clinical Efficacy and Application

Edoxaban, the fourth NOAC approved by the FDA on January 9, 2015 (details on its pharmacokinetics and clinical properties are provided in Table 1), has established itself as a valuable addition to the treatment armamentarium. Numerous trials have confirmed its efficacy and safety across various thromboembolic and cardiovascular conditions. The landmark trial of the ENGAGE AF-TIMI 48 study (2013) evaluated edoxaban’s long-term effectiveness and safety compared to warfarin in patients with AF. Edoxaban demonstrated non-inferiority to warfarin in preventing stroke or systemic embolism, with the added benefit of significantly lower bleeding and cardiovascular death rates (details in Table 2) [93]. Edoxaban’s utility extends to VTE treatment. The Hokusai-VTE trial (2013) assessed its efficacy as an alternative to warfarin in patients with acute VTE following initial heparin therapy. Edoxaban, administered once daily after initial heparin administration, proved non-inferior to warfarin while causing significantly less bleeding across a broad spectrum of VTE patients, including those with severe pulmonary embolism [94]. A 2016 subgroup analysis of this trial suggested edoxaban might be as effective as warfarin for treating VTE in cancer patients, with a reduced risk of clinically relevant bleeding [95]. This was further confirmed by the Hokusai VTE Cancer trial (2018), which demonstrated the non-inferiority of oral edoxaban to subcutaneous dalteparin for preventing recurrent VTE or major bleeding in patients with cancer [96]. Edoxaban’s effectiveness in preventing thromboembolic complications following surgery has also been established. In 2014, a Japanese study compared edoxaban to subcutaneous enoxaparin for preventing thromboembolic events after hip fracture surgery, demonstrating similar safety and efficacy [97]. Another 2014 trial, STARS E-3, showed oral edoxaban to be more effective than subcutaneous enoxaparin for thromboprophylaxis following total knee arthroplasty, with a similar bleeding incidence [98]. Studies have explored edoxaban’s safety and efficacy in special populations. A 2015 study in Japanese patients with non-valvular atrial fibrillation (NVAF) and severe renal impairment found similar safety, plasma concentration, and biomarker profiles for edoxaban compared to doses used in patients with normal or mild renal function [99]. The ENSURE-AF trial (2016) compared edoxaban to enoxaparin–warfarin therapy in patients with AF undergoing cardioversion. Edoxaban displayed similar rates of major bleeding and thromboembolism compared with patients on well-managed, optimized enoxaparin–warfarin therapy [100]. Recent years have witnessed continued investigation into edoxaban’s potential. The ENNOBLE-ATE in Children at Risk Because of Cardiac Disease trial (2022) suggests edoxaban as a promising alternative thromboprophylaxis option for children with cardiac disease, with low bleeding rates and advantages like once-daily dosing and less frequent monitoring compared to SOC (VKA or LMWH) [101]. The ENAVLE trial (2023) compared edoxaban with warfarin in patients following surgical bioprosthetic valve implantation or valve repair. Edoxaban showed non-inferiority to warfarin in preventing thromboembolism and a potentially comparable risk of major bleeding during the initial 3 months [102]. Finally, the KABUKI trial (2024) explored edoxaban in patients with chronic thromboembolic pulmonary hypertension (CTEPH). The study demonstrated edoxaban’s non-inferiority to warfarin in preventing worsening pulmonary vascular resistance with a similar incidence of clinically relevant bleeding [103].

Through a diverse range of pivotal trials, edoxaban has solidified its status as a key player in the NOAC landscape, demonstrating broad efficacy and safety in preventing and treating thromboembolic and cardiovascular conditions,

Having delineated the individual NOACs and their pivotal roles in managing thromboembolic and cardiovascular conditions, we now transition to examine the PARs, exploring their intricate involvement in inflammation and the broader implications for therapy.

## 3. Exploring Protease-Activated Receptors (PARs): Unveiling Their Role in Inflammation and Disease

PARs are a unique class of G protein-coupled receptors activated by proteolytic cleavage. This cleavage unveils a tethered ligand that binds to a specific region within the receptor, triggering a conformational shift and subsequent cellular signaling. Interestingly, synthetic peptides mimicking the tethered ligand can directly stimulate PARs, even in the absence of protease activity [104]. There are four known PAR subtypes: PAR1, PAR2, PAR3, and PAR4, each with distinct characteristics like the number of amino acid residues, tethered ligands, and preferred activating proteases/synthetic ligands (refer to Table 3 for details). Notably, the cellular responses elicited by PAR activation are highly variable, depending on the specific PAR subtype and the activating protease involved. Even with the same PAR, different proteases can trigger diverse cellular responses. This intricate interplay between proteases, PARs, and resulting cellular responses paints a complex and evolving picture in the niche of cell signaling. The complex signal transduction mechanisms in PAR-mediated inflammation involve factors like Ca^2+^, extracellular signal-regulated kinase (ERK)1/2, and cyclic adenosine monophosphate (cAMP) signaling, ultimately leading to a cascade of cellular changes (as illustrated in Figure 3 and Figure 4). Interestingly, activation of PAR3 requires the involvement of PAR1, PAR2, or PAR4 [105,106].

Recent evidence suggests FXa exerts non-hemostatic effects, primarily mediated through PAR-1 and PAR-2 [107,108]. These effects contribute to pathophysiological conditions like atherosclerosis, inflammation, and fibrosis. FXa initiates diverse non-hemostatic effects across different cell types, bridging coagulation with inflammatory conditions. While thrombin production primarily occurs on platelets and subendothelial vascular walls, its presence has also been detected extraluminally, in areas like synovial fluid [109] and around tumors [110]. Thrombin plays a central role in platelet activation [111]. PAR1, PAR3, and PAR4 can all be activated by thrombin, but each through distinct mechanisms [112].

Notably, it has been observed that PARs play a dynamic role in modulating cellular signaling across different inflammatory zones in the human body. They can both promote and suppress inflammatory reactions, indicating a dualistic function in the regulation of inflammation. This principle and its significance in diseases linked to inflammation are detailed in Table 4 and explored in the subsequent discussion.

### 3.1. Exploring the Role of PARs in Neuroinflammation and Hyperalgesia

PARs are extensively expressed in central and peripheral neurons, implicating them not only in physiological and somatosensory functions, but also in progression of neuroinflammatory conditions, with the capacity to both induce neurodegenerative and neuroprotective effects [113]. Among the PAR family, PAR1 and PAR2 emerge as key contributors in nerve-related inflammation. Granzyme B (a serine protease released by cytotoxic T-lymphocytes and natural killer cells, instrumental in inducing apoptosis in target cells as part of the immune response) has been observed to mediate neurodegeneration through PAR1 activation, triggering diverse downstream signaling cascades. Given its ability to induce neural apoptosis, inhibiting PAR1 is hypothesized to offer a neuroprotective effect in the context of neuroinflammatory diseases [114]. PAR2, found on glia, neurons, and mast cells, is noted for enhancing neuroinflammation in the CNS by activating pathways like ERK1/2, c-Jun N-terminal kinase (JNK), and p38 mitogen-activated protein kinases (MAPKs). Moreover, its interaction with glial maturation factor and mast cell proteases, including mouse mast cell protease (MMCP)-6 and MMCP-7, results in heightened PAR2 expression in primary astrocytes and neurons, further intensifying the neuroinflammatory condition [115]. However, it is crucial to note that PAR1 activation through low concentrations of thrombin has exhibited therapeutic potential, showcasing neuroprotective effects that enhance cell viability and microarchitecture in primary cortical neurons, safeguarding them against excitotoxic damage caused by glutamate exposure [113].

The PARs play a role in hyperalgesia by directly activating nociceptors and/or sensitizing them. This process involves the activation of receptors such as transient receptor potential vanilloid subfamily 1 (TRPV1), TRPA1, TRPV4, and P2X3, which are crucial in mediating neurogenic inflammation and inflammatory pain [116]. Additionally, PAR2 coupling with TRPV4 has been reported to amplify the pro-inflammatory state via calcium release, leading to a chronic hyperexcitability of nociceptors, especially implicated in irritable bowel syndrome (IBS) [117,118]. Furthermore, it is observed that the PAR2 agonist signals primary spinal efferent neurons, meditated by central activation of neurokinin (NK) 1-receptors and release of prostaglandins to cause a pronounced and chronic hyperalgesia [119]. In contrast, it was demonstrated that topical administration of selective PAR1 agonists resulted in attenuation of the nociceptive response elicited by noxious mechanical and thermal stimuli. Furthermore, a significant reduction in inflammatory mechanical and thermal hyperalgesia was observed [120]. 

To conclude, PAR1 and PAR2 contribute to neurodegeneration and neuroprotection through neuroinflammation and hyperalgesia. PAR1 activation by low thrombin concentrations may improve cell viability and protect against excitotoxic damage, making it an attractive therapeutic approach. However, PAR2’s various signaling pathways contribute to neuroinflammatory diseases and persistent pain. These findings suggest that PARs, particularly PAR2, may cure neuroinflammation and hyperalgesia. This could lead to new treatments.

### 3.2. Chronic Gastrointestinal Inflammation

Chronic inflammation in the gastrointestinal tract is closely linked to the activity of both human and bacterial proteases within the gut’s lumen. These proteases, through PARs, notably PAR1 and PAR2, regulate the intestinal barrier’s permeability, mirroring their role in endothelial barriers. The presence of all four PARs in the gastrointestinal lining underscores their significant role in driving the complex mechanisms behind chronic gastrointestinal conditions via PAR-mediated inflammation.

Inflammatory bowel disease (IBD), encompassing Crohn’s disease (CD) and ulcerative colitis (UC), is a chronic and recurrent inflammatory condition of the gastrointestinal (GI) tract. PARs have been implicated in IBD pathogenesis through a biphasic regulatory mechanism depending on the cell type and the pathogen in question. For instance, PAR1 exhibits a pro-inflammatory role in bacterial-induced colitis, as evidenced by *Citrobacter rodentium* infection and the enhancement of T helper 17 cell (Th17)-type immune responses in Crohn’s disease patients [121]. Conversely, PAR1 displays an anti-inflammatory effect in oxazolone-induced colitis, which involves a T helper 2 cell (Th2)-mediated immune response, showcasing its multifaceted role in modulating inflammation within the gastrointestinal tract [122]. Furthermore, the function of PAR1 in colitis is found to vary based on cell type. Boucher et al. demonstrated that PAR1 expression on specific effector cells within bacterial-derived colitis differentially modulates pro-inflammatory cytokine expression profiles [123], highlighting the nuanced role of PAR1 in influencing the inflammatory response.

PAR2 exhibits widespread distribution within the GI cells, particularly at apical and basolateral membranes of epithelial cells [124]. PAR2 deficiency has been reported to significantly attenuate *C. rodentium*-induced inflammation, suggesting a proinflammatory effect similar to PAR1 [125]. Furthermore, increased mast cell tryptase secretion, which can activate the PAR2/Protein kinase B (AKT)/mammalian target of rapamycin (mTOR) signaling pathway, can contribute to intestinal fibrosis, a hallmark of IBD [126]. Alternatively, within a high-fat diet-fed murine colitis model, PAR2 seems to play a protective role in inflammatory processes. This protective effect appears to be mediated by promoting autophagy and inhibiting apoptosis in intestinal cells. This recent study underscores the potential of PAR2 agonism as a novel therapeutic avenue for patients suffering from IBD who also present with comorbid metabolic syndrome. This interventional strategy could address the complex pathophysiological nexus between inflammatory and metabolic pathways in such cohorts. Notably, the therapeutic implication draws from the multifaceted role that PARs play in modulating gastrointestinal inflammation and metabolic homeostasis, further bolstering the case for PAR2-targeted treatments in dual-pathology contexts such as IBD with concurrent metabolic syndrome [127]. 

Overall, human and bacterial proteases significantly affect chronic inflammation in the gastrointestinal tract through PARs, especially PAR1 and PAR2, which regulate the permeability of the intestinal barrier. IBD features a complex interplay of these receptors: PAR1 reduces inflammation in Th2-mediated colitis but increases Th17 immune responses in CD and promotes inflammation in colitis caused by bacteria. PAR2 has also shown pro-inflammatory properties, by reducing inflammation in PAR2-deficient individuals and its involvement via the PAR2/AKT/mTOR pathway. However, PAR2 offers protective benefits in colitis induced by a high-fat diet, indicating therapeutic potential for IBD. This highlights the dual role of PARs in GI inflammation, suggesting their exploration as therapeutic targets for IBD.

### 3.3. Inflammatory Disorders of Respiratory System

The role of PARs in the pathophysiology of respiratory disorders, including asthma and pneumonia, represents a focal point of medical research due to the critical insights it provides into pulmonary inflammatory mechanisms. The heightened research interest is propelled by clinical observations that demonstrate significantly increased concentrations of proteases, such as thrombin and tryptase, which activate PARs in the bronchoalveolar lavage fluid of patients suffering from pulmonary inflammation [128,129]. These proteases are implicated in the exacerbation of airway remodeling and hyperresponsiveness, which are pivotal factors in the progression of these respiratory conditions [130,131,132]. The upregulation of key inflammatory mediators, including matrix metalloproteinase (MMP)-9, interleukin (IL)-1β, IL-6, and substance P (SP), through PAR activation, delineates a pathogenic pathway that contributes to the severity and chronicity of respiratory diseases [130]. This evidence highlights the potential of targeting PARs in the therapeutic management of pathological airway responses.

Direct activation of PAR1 is implicated in the initiation of asthma symptoms through mechanisms involving transforming growth factor- β (TGF-β), which in turn fosters a Th2-skewed immune response characteristic of allergic asthma [133]. Conversely, the inhibition of PAR2 has shown efficacy in reducing allergen-induced airway hyperresponsiveness and the infiltration of inflammatory cells [134]. This highlights the mechanistic role of PAR2 in mediating the pathophysiology of allergic airway inflammation, offering a clearer understanding of its potential as a therapeutic target in asthma management. Furthermore, non-mammalian PAR-activating proteases, found in sources like house dust mites and cockroaches, suggest the significance of PARs in respiratory health [135]. In fact, selective PAR2 antagonists have shown efficacy in reducing allergen-induced asthma symptoms in animal models [136]. The influence of PAR2 on bronchoconstriction or dilation presents a complex picture, colored by interspecies differences and the nuanced responses of diverse tissues [137,138]. Despite these complexities, human studies suggest that both PAR1 and PAR2 agonists—activated by thrombin, trypsin, and tryptase—have the capacity to induce bronchoconstriction by catalyzing Ca^2+^ signaling pathways within airway smooth muscle cells [139,140]. Additionally, persistent activation of PAR1 and PAR2 is associated with the development of pulmonary fibrosis, suggesting a deleterious role in prolonged inflammatory responses [141,142,143]. Intriguingly, a low thrombin dose has been observed to mitigate inflammatory signaling in the context of allergic bronchial asthma, indicating a potential therapeutic window where careful modulation of PAR1 may exert anti-inflammatory effects [144].

Paralleling their role in asthma, accumulating evidence suggests pro-fibrotic and pro-inflammatory effects of PARs in the pathogenesis of pneumonia. Streptococcal pneumonia appears to exploit PAR1 activation to exacerbate neutrophil infiltration and viral co-infections through downstream mediators such as IL-1β, chemokine (C-X-C motif) ligand (CXCL) 1, and chemokine (C-C motif) ligand (CCL) 7 and CCL2 [145]. Conversely, PAR1 inhibition has been shown to ameliorate excessive neutrophilic inflammation and alveolar barrier disruption [146]. PAR2 activation by proteases facilitates Streptococcus pneumoniae colonization in the respiratory tract followed by systemic dissemination, most likely due to impairment of the lung–blood barrier integrity [147]. Conversely, evidence suggests that PAR2 deletion has the capacity to attenuate TRPV4-dependent Ca^2+^ signaling, which primarily promotes persistent lung inflammation [148]. Interestingly, PAR4 appears to contribute to host defense by enhancing the systemic cytokine response during late-stage pneumococcal infection in murine models [149]. Despite PAR4’s immunomodulatory role, PAR1 and PAR2 remain critical modulators in pneumococcal pneumonia. 

In conclusion, elevated protease activity drives pulmonary inflammation via PAR activation, impacting respiratory diseases. PAR1 activation promotes Th2-driven asthma and bronchoconstriction, while PAR2 inhibition reduces airway hyperresponsiveness. Low-dose thrombin may exhibit anti-inflammatory effects. However, both PAR1 and PAR2 activation contribute to bronchoconstriction and lung fibrosis. Combined PAR1 and PAR2 antagonism might offer superior therapeutic benefit compared to targeting them individually.

### 3.4. Cardiovascular Inflammatory Disorders

The multifaceted expression profile, enzymatic activity, and availability of selective antagonists position PARs as attractive molecular targets for therapeutic intervention in cardiovascular diseases. Furthermore, evidence suggests that PAR signaling disrupts the normal homeostatic functions of cardiac fibroblasts and cardiomyocytes, contributing to the structural remodeling of myocardium [150,151,152]. Elucidating the specific roles of PARs in these processes holds significant promise for improving our understanding and treatment of cardiovascular system disorders.

Mirroring the observed functional dichotomy of PAR1 and PAR2 discussed previously, these receptors appear to mediate opposing effects on inflammatory responses within viral myocarditis, specifically in ssRNA viruses such as coxsackievirus group B (CVB) 3 and influenza A virus [153]. Studies have demonstrated that upregulation of PAR1- and PAR4-protected cardiomyocytes by attenuating viral myocarditis [154,155]. This protective effect appears to be mediated by enhanced toll like receptor (TLR)3 signaling, and subsequent increased expression of Interferon (IFN) β and CXCL10, which was associated with reduced viral load and decreased expression of pro-inflammatory mediators, such as tumor necrosis factor (TNF)-α, monocyte chemoattractant protein (MCP) 1, and CXCL1. Notably, PAR1 is primarily expressed by non-hematopoietic cells, which can potentially infiltrate the heart and further limit viral replication by augmenting the cytotoxic activity of natural killer (NK) cells [156]. Conversely, PAR2 activation was observed to reduce the antiviral response mediated by TLR3-dependent expression of IFN-β, promoting viral load and replication, potentially leading to severe myocarditis [157,158]. Interestingly, however, both PAR1 and PAR2 activation have been implicated in the pathogenesis of cardiac remodeling, heart dysfunction, and myocardial infarction [150,151].

To conclude, PAR signaling negatively affects the functions of cardiac fibroblasts and cardiomyocytes, contributing to changes in myocardial remodeling. In the context of viral myocarditis from ssRNA viruses, PAR1 and PAR4 provide protective effects by enhancing TLR3 signaling and increasing IFN-β levels, which help lower viral load and inflammatory markers. Conversely, PAR2 activation can weaken the antiviral response, potentially increasing viral replication and severity of myocarditis; furthermore, both PAR1 and PAR2 are implicated in cardiac dysfunction. These findings collectively suggest that broad-spectrum PAR inhibition might not be an optimal therapeutic strategy for myocarditis. Instead, a more nuanced approach utilizing PAR-specific ligands tailored to their individual effects may be a more promising avenue for therapeutic development.

### 3.5. Inflammatory Disorders of Urinary System

PARs emerge as key regulators of inflammatory processes within the urinary system, encompassing the kidneys, bladder, and urethra, under both physiological and pathological conditions. In cystitis, a condition frequently associated with urothelial dysfunction and persistent bladder inflammation, PAR signaling appears to play a pivotal role. While PAR activation, particularly by PAR1 and PAR4, is not directly causative of cystitis in the urothelium, it contributes to the disease progression through the mediation of macrophage migration inhibitory factor (MIF), at least in part through activation of CXCR4 receptors [159]. Furthermore, PAR4 activation in urothelial cells has been shown to induce the release of high-mobility group box 1 protein (HMGB1) via a MIF-dependent mechanism, potentially contributing to bladder pain [160]. Notably, PAR1 expression appears to be upregulated in the bladder following cystitis induced by classical mediators such as lipopolysaccharide (LPS), SP, and cyclophosphamide (CYP), potentially exacerbating the bladder inflammatory response, thereby impacting bladder sensation and function during cystitis [161,162]. Similarly, PAR2 activation in primary cultures of human urothelial cells and the mouse urothelium/suburothelium is associated with increased (cyclooxygenase) COX-2 expression, augmenting the inflammatory processes in cystitis primarily mediated by the ERK1/2 MAPK pathway [163].

Within the renal compartment, thrombin-mediated PAR activation has been implicated in both autocrine and paracrine signaling pathways, thereby linking PARs and proteases to nephritis [164]. Renal inflammation, a key driver of chronic kidney disease (CKD) and often accompanied by renal tubulointerstitial fibrosis (RTIF), exhibits a direct correlation between FXa activity and the severity of fibrosis. FXa-meditated PAR1 signaling has been demonstrated to enhance the production and proliferation of procollagen by fibroblasts [165]. Similarly, PAR2 activation by FXa in these cells appears to stimulate both fibrotic and pro-inflammatory responses [166]. Additionally, PAR2 plays a critical role in the formation of glomerular crescents (a hallmark of rapidly progressive forms of glomerulonephritis) by promoting macrophage accumulation, activation of parietal epithelial cells, and glomerular thrombosis [167]. 

Collectively, these findings suggest PARs, particularly PAR1 and PAR2, are crucial regulators of inflammatory processes in the urinary system, significantly influencing conditions like cystitis and renal inflammation through mechanisms involving macrophage migration, fibrotic responses, and inflammatory mediator activation.

### 3.6. Rheumatoid Arthritis and Osteoarthritis

PAR1 and PAR2, have emerged as intriguing players in the intricate world of arthritis. While arthritis encompasses various joint disorders, this review focuses on their involvement in rheumatoid arthritis (RA), an autoimmune systemic inflammatory disorder, and OA, a degenerative joint disease with inflammation as its hallmark.

Initial evidence for PAR2’s direct role in rheumatic diseases came from a 2003 study by Ferrell et al. [168]. Their work demonstrated that activating PAR2 with a specific peptide triggered robust inflammation in wild-type mice, including joint swelling and increased blood flow to the synovium. In addition, studies since have shown that spontaneous release of TNF-α and IL-1β from the RA synovium is significantly inhibited by a PAR2 antagonist in a dose-dependent manner [169]. Further solidifying PAR2’s role as a key pro-inflammatory mediator, a study by Russell et al. [170] demonstrated that PAR2 plays an acute inflammatory role in the knee joint. This effect is mediated by TRPV1 and NK1 receptors, suggesting involvement of both PAR2-mediated neuronal sensitization and leukocyte trafficking. Supporting this notion, another study examined the differential expression of mast cell tryptase and PAR-2 in the synovium and synovial fluid of patients with RA and OA [171]. The study revealed that mast cell tryptase in synovial fluid stimulates the proliferation of synovial fibroblasts (SFCs) and the release of pro-inflammatory cytokines via PAR-2. This suggests a potential role for PAR-2 in the exacerbation of synovitis in arthritis. Further substantiating this link, murine studies assessed the capacity of mast cell tryptase to mediate synovial pro-inflammatory responses via PAR-2 and investigated whether degranulating mast cells induced synovial hyperemia by PAR-2 activation [172].

Clinical relevance was established by studies revealing elevated PAR2 expression in circulating leukocytes from RA patients compared to healthy individuals [173].

Further studies using murine models solidified PAR2’s pro-inflammatory function. Mice deficient in PAR2 exhibited reduced inflammatory response during acute arthritis. Additionally, inflamed joints showed increased PAR2 protein expression, and silencing PAR2 using siRNA effectively dampened inflammation [174].

PAR1, unlike PAR2, appears to have opposing effects in RA. While urokinase plasminogen activator (uPA)-activated PAR1 demonstrably attenuates inflammatory osteoclastogenesis, a process essential for bone destruction in RA [175], the picture with thrombin–PAR1 interaction is more complex. Thrombin can activate PAR1 on human osteoblasts, leading to increased CCL2 expression, which promotes macrophage aggregation, potentially contributing to inflammation [176]. Similarly, thrombin–PAR1 signaling induces RANTES mRNA expression, possibly facilitating the migration of inflammatory cells to the synovium in RA patients [177].

However, thrombin–PAR1 signaling in synovial fibroblasts appears to have a dual effect. While it stimulates their proliferation [178], it also promotes the secretion of MMP-2 and inhibits the invasion of these fibroblasts, potentially limiting tissue damage. Additionally, thrombin–PAR1 activation reduces the production of pro-inflammatory cytokines IL-17 and TNF-α by synovial fibroblasts. In contrast, PAR2 activation in synovial fibroblasts appears to be predominantly detrimental, promoting their growth, invasion, and TNF-α production [179]. This suggests that while PAR1 has some anti-inflammatory properties in synovial fibroblasts, PAR2 seems to be a key driver of their aggressive behavior, contributing to the pathological progression of RA. Within the context of OA, an in vitro model using LPS-induced inflammation in chondrocytes revealed upregulation of PAR2 expression [180]. Interestingly, this study also showed that subsequent downregulation of PAR2 expression was associated with a decrease in inflammatory markers, including TNF-α, IL-6, IL-8, and IFN-γ.

In summary, PAR1 and PAR2 are significant modulators in arthritis, particularly in RA and OA. PAR2 activation has been shown to induce inflammation, joint swelling, and pro-inflammatory cytokine release, while its antagonism reduces inflammatory responses in RA. In contrast, PAR1 exhibits a dual role: it can attenuate osteoclastogenesis but may also promote inflammation through thrombin interactions. Studies indicate elevated PAR2 expression in RA patients, reinforcing its pro-inflammatory role, while PAR1’s effects suggest a complex interplay in synovial fibroblast behavior, and PAR3 and PAR4 seem to have minimal involvement. These findings suggest that PAR2 antagonists may hold promise as therapeutic strategies for the treatment of arthritis.

Having established the critical role of PARs in inflammatory processes, we now shift our focus back to individual NOACs. Now we will delve into the complex interplay between NOACs and inflammatory pathways, exploring their potential anti-inflammatory properties and the broader therapeutic implications.

## 4. Anti-Inflammatory Potentials of NOACs

### 4.1. Apixaban

#### Analysis of the Anti-Inflammatory and Risk-Benefit Profile and Clinical Implications of Apixaban

In a review of the anti-inflammatory effects of apixaban (Table 5), the data present a multifaceted picture. Studies stemming from the ARISTOTLE trial in AF patients revealed limited impact on conventional inflammatory markers. Specifically, apixaban did not significantly influence Growth Differentiation Factor 15 (GDF-15), high-sensitivity C-reactive protein (hs-CRP), or IL-6 levels in these patients [181,182]. However, intriguingly, obesity, a condition often associated with chronic inflammation, was linked to better survival outcomes in apixaban-treated AF patients, hinting at a potential, yet indirect, anti-inflammatory role for the drug [183]. In contrast to these neutral to ambiguous findings in AF, more targeted research has offered promising results. An in vitro and in vivo study demonstrated that apixaban, especially when used in combination therapy, increased the specificity of activated protein C (APC), a molecule with known anti-inflammatory effects [184]. Similarly, in a clinical setting of coagulopathy associated with extracorporeal membrane oxygenation (ECMO), apixaban impaired plasmatic coagulation and platelet aggregation, though the direct anti-inflammatory impact was not assessed [185]. Furthermore, in a study related to CKD, apixaban significantly reduced markers like vascular cell adhesion molecule-1 (VCAM-1), intercellular adhesion molecule-1 (ICAM-1), and von Willebrand Factor (VWF) while normalizing reactive oxygen species (ROS) levels in endothelial cells exposed to uremic serum [186]. Collectively, while the anti-inflammatory effects of apixaban appear to be condition- and context-dependent, these findings open doors for further investigations into its role beyond anticoagulation.

The heterogenous data concerning apixaban’s anti-inflammatory effects point toward an exigent need for more granular future studies. In vitro cell line studies employing endothelial and glial cells could be particularly informative, with assays such as Western blot and RT-PCR focusing on key signaling pathways like NF-κB or MAPK. Moreover, transgenic mouse models could offer crucial in vivo insights. These mice, engineered through advanced genetic manipulation techniques such as CRISPR to either amplify or suppress inflammatory responses, can serve as valuable models to assess apixaban’s systemic and localized impact. One particularly promising model is the L2-IL-1β transgenic mice, which exhibit chronic inflammation along with a stepwise development of esophageal conditions including Barrett’s esophagus-like metaplasia, dysplasia, and adenocarcinoma [187]. These mice also display squamous cell carcinoma (SCC) in the esophagus and tongue, with T cell-predominant inflammatory profiles and significantly increased levels of various pro-inflammatory markers. Importantly, this model could serve as an excellent system for studying the anti-inflammatory potential of apixaban in inflammation-driven esophageal and oral cancers. Researchers could administer apixaban to these mice at different developmental stages and then employ immunohistochemistry and cytokine profiling to assess whether apixaban mitigates the inflammation and related carcinoma progression. This specific model, validated for both germ-free and specific pathogen-free conditions, could provide nuanced insights into how apixaban might interact with IL-1β-mediated inflammation pathways and whether it has a role in halting or reversing the progression of SCC and adenocarcinoma. In addition to classical cell lines and animal models, specialized in vitro systems can offer invaluable insights. For instance, the recent development of a 3D culture system to generate chondrocytes from bone marrow-derived mesenchymal stem cells (BMSCs) has opened new avenues in cartilage biology and inflammation research [188]. Given the key role of chondrocytes in maintaining cartilage integrity and their susceptibility to inflammation, this model could serve as an ideal platform for investigating FXa’s role in inflammation associated with OA. In this system, a pro-inflammatory environment is simulated by treating chondrocytes with lipopolysaccharide, creating a setup that can be leveraged to evaluate the anti-inflammatory effects of apixaban. Specifically, the impact of apixaban on tumor necrosis factor-α levels and other inflammatory markers could be quantified using enzyme-linked immunosorbent assay (ELISA). Moreover, the 3D architecture of this model more closely mimics the in vivo cellular environment compared to 2D monolayers, offering a more physiologically relevant assessment of apixaban’s potential anti-inflammatory effects. Such an approach not only expands the toolkit for studying cartilage-related disorders but also offers a more nuanced understanding of how apixaban could mitigate inflammation in a highly tissue-specific context.

**Table 5 ijms-25-08727-t005:** Anti-inflammatory profile of apixaban.

Model or Population	Mono/Combination Therapy	Condition Investigated	Inflammatory Markers Examined	Effect Observed	Reference
Inflammatory marker was measured at study entry, and at 2 months in 4830 patients in the ARISTOTLE trial with 1.8 years median follow-up.	Monotherapy	AF	IL-6	Repeated measurements of IL-6 suggest that persistent systemic inflammation is independently associated with increased mortality in apixaban treated AF patients, even after considering established clinical risk factors and other strong cardiovascular biomarkers.	*Circulation. 2014 Nov 18;130(21):1847–58.* [182]
18,201 patients with AF from the ARISTOTLE trial, out of which biomarkers were measured for 14,798 patients.	Monotherapy	AF	GDF-15	The effects of apixaban in reducing stroke, mortality, and bleeding were consistent irrespective of GDF-15 levels, but no data is shown if apixaban had any influence on the levels of GDF-15 to mitigate inflammation.	*Heart. 2016 Apr;102(7):508–17.(***)* [181]
18,201 patients with AF who were randomized to receive either apixaban or warfarin in the ARISTOTLE trial.	Monotherapy	AF	hs-CRP and IL-6	No significant interactions were observed with respect to apixaban and IL-6 or CRP levels and their outcomes.	*Heart. 2016 Apr;102(7):508–17. (***)* [181]
14,753 patients from the Apixaban for Reduction In STroke and Other ThromboemboLic Events in Atrial Fibrillation (ARISTOTLE) trial.	Monotherapy	AF	Hs-CRP and IL-6	Obesity was found to be linked with better survival outcomes in apixaban treated patients. Given that obesity is often associated with chronic inflammation, it is possible that apixaban may have a modulatory role in dampening inflammation in obese, anticoagulated patients with AF.	*Open Heart. 2018 Nov 1;5(2):e000908.* [183]
**In vitro *model*:** Glia cell lines (specifically microglial and astrocytic cell lines). **In vivo *animal model*:** Adult male mice, of similar genetic lineage as C57BL/6, developed by Institute of Cancer Research. ***Human model*:** Human plasma and cerebrospinal fluid (CSF), specifically, the CSF taken from viral meningoencephalitis patients and controls was analyzed.	Combination therapy where apixaban individually and in combination with alpha-naphthylsulphonylglycyl-4-amidinophenylalanine piperidine (NAPAP) was used.	Measurement of activated protein C (APC) activity in the context of neural inflammation: ▪ Mild traumatic brain injury (mTBI) in mouse brain slices. ▪ Systemic LPS injection, which is commonly used to induce an inflammatory response. ▪ Viral meningoencephalitis in patients, as evidenced by elevated APC activity in the CSF.	Conventional inflammatory markers were not assessed but APC titers were measured. APC modulates inflammation by: ▪ Reducing the release of pro-inflammatory cytokines. ▪ Protecting endothelial cell barriers. ▪ Decreasing leukocyte adhesion and migration.	Apixaban increases the specificity of APC activity, which in turn alludes to the possibility that APC’s anti-inflammatory effects are advantageously affected in the presence of apixaban.	*Int J Mol Sci. 2020 Mar 31;21(7):2422.* [184]
Venovenous extracorporeal membrane oxygenation (n = 10) due to acute respiratory distress syndrome; and patients treated with venoarterial extracorporeal membrane oxygenation (n = 8) due to cardiocirculatory failure in the ICU of a university hospital.	Monotherapy	Coagulopathy associated with extracorporeal membrane oxygenation (ECMO) in patients with severe cardiocirculatory and/or respiratory failure.	Plasmatic coagulation and platelet aggregation.	Plasmatic coagulation and platelet aggregation were impaired before ECMO due to apixaban.	*Crit Care Med. 2020 May;48(5):e400–e408.* [185]
***Macrovascular endothelial cells*:** HUVEC (Human Umbilical Vein Endothelial Cells) ***Microvascular endothelial cells*:** HMEC (Human Microvascular Endothelial Cells)	Monotherapy	Protective role of apixaban in uremia caused by CKD.	VCAM-1, ICAM-1, p38MAPK and p42/44 (also known as ERK1/2)	Apixaban reduced VCAM-1, ICAM-1, and VWF expression, normalized ROS levels and eNOS, and inhibited p38MAPK and p42/44 activation in endothelial cells exposed to uremic serum.	*Cardiovasc Drugs Ther. 2021 Jun;35(3):521–532.* [186]

The studies marked with (***) indicate no significant impact of NOAC on specific inflammatory markers, suggesting limited influence despite research efforts in this area.

In the same vein, 3D organoid cultures [189], representing different human organs, offer a more physiologically relevant platform to study apixaban’s effects on specific tissues. These organoids could be treated with various concentrations of apixaban, followed by immunohistochemical and transcriptomic analyses to evaluate anti-inflammatory mechanisms. For example, multicellular 3D neurovascular unit organoids present a promising platform for studying apixaban’s anti-inflammatory mechanisms in the context of blood brain barrier (BBB) dysfunction. These organoids, comprising human brain microvascular endothelial cells, pericytes, astrocytes, microglia, oligodendrocytes, and neurons, mimic the complex cellular interactions and environment of the native BBB [190]. Dysfunction of the BBB is evident in many neurological disorders, such as ischemic stroke and chronic neurodegenerative diseases, and is often associated with hypoxia and neuroinflammation. Utilizing this organoid model to simulate hypoxic conditions—by culturing the organoids in a hypoxic chamber with 0.1% O2 for 24 h—could offer an invaluable system for investigating the anti-inflammatory effects of apixaban. Specifically, the model enables quantification of changes in permeability, pro-inflammatory cytokine production, and oxidative stress markers in response to apixaban treatment. Preliminary experiments with other anti-inflammatory agents like secoisolariciresinol diglucoside and 2-arachidonoyl glycerol have already shown promising results in reducing inflammatory cytokine levels under hypoxic conditions. Therefore, this 3D organoid model not only recapitulates key characteristics of BBB dysfunction, but also offers a platform for assessing how apixaban may counteract these changes at the cellular and molecular levels. It is an ideal system for high-throughput screening of drug candidates and elucidating the mechanistic pathways through which apixaban exerts its anti-inflammatory effects.

Regarding human research, future cohort studies should focus on specialized sub-populations, such as those with chronic inflammatory conditions like chronic kidney disease [191] or autoimmune disorders [192], to investigate real-world anti-inflammatory efficacy. Statistical comparisons can then be made to measure apixaban’s impact on specific markers and outcomes. These multi-pronged, detailed experimental approaches would provide a comprehensive understanding of apixaban’s anti-inflammatory potential and could pave the way for its broader clinical applications beyond anticoagulation. Also, in the section on specific human populations, the utility of mass spectrometry for deeply understanding the anti-inflammatory effects of apixaban can be highlighted by drawing upon insights from studies investigating metabolic changes in coronary artery disease (CAD) patients. By leveraging widely targeted plasma metabolomic and lipidomic profiling via mass spectrometry in CAD patients, researchers have successfully mapped the metabolic disturbances that accompany systemic immune inflammation [193]. A comprehensive panel of 860 metabolites was assessed, offering a robust framework for defining general and low-grade inflammatory states. Mass spectrometry enabled precise quantification of associations between multiple metabolites and seven established inflammatory markers, including high-sensitivity C-reactive protein and the neutrophil–lymphocyte ratio. Techniques like least absolute shrinkage and selection operator (LASSO) logistic-based classifiers were deployed to further hone in on the significant biomarkers and metabolic pathways involved, such as glycerophospholipid metabolism and arginine and proline metabolism. In particular, β-pseudouridine emerged as an inflammation-associated metabolite linked to arteriosclerosis indicators. Importantly, mass spectrometry facilitated the development of predictive models with high discriminatory power (area under the curve: 0.81 to 0.88) for various inflammatory states. Adding mass spectrometry to the investigation of apixaban’s effects in specific populations, such as CAD patients, could offer unparalleled depth of insight. Not only does it allow for the precise quantification of a wide array of metabolites and lipids, but it also enables the elucidation of the intricate metabolic pathways that may be impacted by apixaban’s anti-inflammatory mechanisms. The detailed metabolic profiles generated could serve as robust markers for assessing drug efficacy and understanding the broader metabolic context in which apixaban operates. This adds a layer of sophistication to therapeutic monitoring, paving the way for individualized treatment approaches that are tailored to the patient’s unique inflammatory and metabolic landscape.

### 4.2. Edoxaban’s Anti-Inflammatory Profile

Edoxaban has showcased diverse anti-inflammatory effects across various pathophysiological conditions (a comprehensive review of these effects is depicted in Table 6). In models of diabetic nephropathy, edoxaban has been shown to reduce the expression of proinflammatory and profibrotic genes such as TGF-β, Plasminogen activator inhibitor (PAI)-1, Collagen Type I/IV, TNF-α, MCP-1, IL-8, and COX-2. It also decreased the expression of PAR1 and PAR2, which are implicated in the pathogenesis of diabetic nephropathy, suggesting a potential therapeutic role for edoxaban in this condition [194]. Similarly, in a model of renal tubulointerstitial injury, edoxaban suppressed elevated levels of FX, PAR-1, and PAR-2, reduced fibrosis and extracellular matrix expression, and attenuated the upregulation of inflammatory molecules and macrophages [166]. Furthermore, the anti-inflammatory effects of edoxaban were also observed in an immortalized proximal tubule epithelial cell line, where it alleviated oxidative stress induced by FXa through its FXa inhibitory and direct radical scavenging activity. This is particularly relevant as oxidative stress is closely interrelated with inflammation. In a transient middle cerebral artery occlusion (tMCAO) model, edoxaban reduced infarct volumes, improved neurological outcomes, enhanced blood–brain barrier function, and attenuated brain tissue inflammation, suggesting potential protective effects in ischemic stroke [195]. However, not all studies have shown significant anti-inflammatory effects. For instance, in a murine model of vascular remodeling and arteriogenesis, edoxaban did not exhibit a significant effect on the mRNA expression of inflammatory cytokines such as IL6, MCP-1, IL1b, and TNF-α [196]. Similarly, in an LPS-induced microvascular thrombosis model, edoxaban did not affect the levels of inflammatory cytokines [197]. Furthermore, its efficacy in vascular remodeling, thrombosis during infection, and systemic inflammation, especially in HIV-positive populations, has been met with equivocal outcomes and did not manifest significant reductions in specific inflammatory markers [196,197,198].

The potential pleiotropic effects of edoxaban in inflammation, atherosclerosis, and stroke have been experimentally tested in various models, with key findings suggesting that edoxaban is a promising target for inhibition of the PAR-2 pathway [199]. Despite these promising observations, there are gaps that need to be addressed through further research. The exact molecular mechanisms by which edoxaban exerts its anti-inflammatory effects are not fully understood. Clinical trials are needed to confirm the anti-inflammatory benefits of edoxaban observed in preclinical models, and the long-term safety and efficacy of edoxaban in the context of inflammation need to be evaluated. It is also important to explore the potential differential effects of edoxaban on various types of inflammation and in different patient populations. In conclusion, edoxaban has demonstrated significant anti-inflammatory effects in various models, suggesting its potential utility in treating inflammatory conditions. However, further research is necessary to fully elucidate its mechanisms of action and to validate these effects in clinical settings. The studies summarized in Table 6 provide a strong foundation for future investigations into the anti-inflammatory potential of edoxaban.

**Table 6 ijms-25-08727-t006:** Anti-inflammatory profile of edoxaban.

Model or Population	Mono/Combination Therapy	Condition Investigated	Inflammatory Markers Examined	Effect Observed	Reference
Male diabetic mice models, specifically eNOS+/+DM, eNOS+/-DM, and eNOS^−/−^DM, to investigate the effects of FXa inhibition. Effects of PAR2 absence on DN were assessed using F2rl1^−/−^; Ins2Akita/+; eNOS+/- mice and compared to F2rl1+/+; Ins2Akita/+; eNOS+/- mice.	Monotherapy	Diabetic nephropathy	TGF β, PAI-1, Collagen Type I, Collagen Type IV, TNF-α, MCP-1, IL- 8 and Prostaglandin-endoperoxide Synthase 2 or COX-2	Edoxaban ameliorated diabetic nephropathy by reducing the expression of key proinflammatory and profibrotic genes, as well as the expression of PAR1 and PAR2.	*Arterioscler Thromb Vasc Biol. 2016;36(8):1525–1533.* [194]
Unilateral ureteral obstruction (UUO) mice (a renal tubulointerstitial fibrosis model) and data from the Food and Drug Administration Adverse Events Reporting System (FAERS) database	Monotherapy	Renal tubulointerstitial injury, which is associated with inflammation and is a major cause of CKD.	PAR1, PAR2, Collagen type 1 and 3, Fibronectin, F4/80 (macrophage marker), TNF-α, IL-10, MCP1, IL1 β, TGFβ, Alpha-smooth muscle actin.	Edoxaban suppressed the elevated levels of FX, PAR-1, and PAR-2, reduced fibrosis and extracellular matrix expression, and attenuated the upregulation of inflammatory molecules and macrophage infiltration in the UUO mouse model of renal tubulointerstitial injury.	*Sci Rep. 2018;8(1):10858.* [166]
Immortalized proximal tubule epithelial cell line (HK-2) derived from normal adult human kidneys	Monotherapy	CKD	The study investigated markers of oxidative stress which is closely interrelated with inflammation. (Intracellular ROS, superoxide anion, peroxynitrite, radical scavenging activity for hydroxyl radicals and hydrogen peroxide.	Edoxaban through its FXa inhibitory and direct radical scavenging activity alleviates oxidative stress induced by FXa, thereby breaking the cycle between oxidative stress and inflammation.	*Int J Mol Sci. 2019;20(17):4140.* [200]
Apolipoprotein E knockout (ApoE^−/−^) mice fed with cholesterol-rich diet categorized into three groups: a control (Co) group on the diet alone, a Warf group treated with warfarin and vitamin K1, and an Edo group treated with edoxaban.	Monotherapy and combination therapy approaches. Co group received cholesterol-rich diet only, Warf group received warfarin and vitamin K1, and the Edo group received edoxaban.	Vascular remodeling, atherosclerosis and arteriogenesis.	Histological assessment of smooth muscle cells per collateral artery in both hind limbs; Frequency of perivascular macrophages in the ligated and non-ligated hind limb; Expression of inflammatory cytokines—IL6, MCP-1, IL1b and TNF-α.	Edoxaban did not exhibit significant effect on inflammation, specifically the mRNA expression of IL6, MCP-1, IL1b and TNF-α in the murine hindlimbs or the spleen remained unaltered.	*Vascul Pharmacol. 2020;127:106661. (***)* [196]
Individuals aged ≥ 18 years, on continuous ART with maintained HIV RNA < 200 copies/mL for 2+ years, having plasma D-dimer level ≥ 100 ng/mL, creatinine clearance ≥ 50 mL/min, and weight ≥ 60 kg, excluding those with recent VTE, contraindications to anticoagulant therapy, certain health conditions like prior stroke, and invasive cancer within the past year, among other criteria.	Monotherapy	Effects of edoxaban in relation to elevated risk of thrombotic events and pro-inflammatory state in the HIV-positive population.	Systemic inflammation—IL6 (high-sensitivity), IL1b and TNFr-1; Monocyte activation—soluble cluster of differentiation (sCD)14 and sCD163; Vascular injury—sVCAM; Thrombin generation—D-dimer, thrombin antithrombin complex (TAT); Circulating microparticle procoagulant activity (MPTF), whole blood tissue factor (WBTF), functional phospholipid surface assay; immunophenotyping of cryopreserved peripheral blood mononuclear cells (PBMC).	Edoxaban did not significantly impact most inflammatory or immune activation markers, but it reduced specific coagulation markers like D-dimer and TAT and was associated with a decrease in effector memory T cells.	*Open Forum Infect Dis. 2020;7(2):ofaa026. (***)* [198]
Male C57Bl/6 N mice aged 6–8 weeks subjected to a transient middle cerebral artery occlusion (tMCAO) procedure. Edoxaban was administered in two different doses across three dosing regimens, with a specific dose of 3.3 mg/kg used for tMCAO experiments. Additionally, the vitamin K antagonist (VKA) phenprocoumon was given at 0.3 mg/kg, three times prior to tMCAO, aiming to achieve INR between 2 and 3 during the tMCAO procedure.	Monotherapy where both drugs (edoxaban and phenprocoumon) were given via oral gavage using a gastric tube and were dissolved in 0.5% (*w*/*v*) methyl cellulose.	Effect of edoxaban on thrombin mediated inflammatory processes in ischemic stroke.	The inflammatory markers and pathways assessed included IL-1b, IL-6, TNF-α gene expression, invasion of immune cells (T cells, neutrophils, macrophages/microglia), and the stabilization of the blood-brain barrier (BBB) through tight junction protein expression and reduced Evans Blue extravasation.	Edoxaban reduced infarct volumes, improved neurological outcomes, enhanced blood-brain barrier function, and attenuated brain tissue inflammation, suggesting its potential protective effects in ischemic stroke.	*Int J Mol Sci. 2021;22(18):9893* [195]
Male Slc:Wistar rats were used in an LPS-induced microvascular thrombosis model.	Monotherapy	Effect of edoxaban on risk of thrombosis during infection, specifically focusing on LPS-induced coagulopathy in rats, where the effect focused on microvascular thrombus formation in the context of this coagulopathy.	The study investigated microvascular thrombus formation in the liver and kidneys, focusing on inflammatory markers IL-6, TNF-α and MCP-1.	Edoxaban did not affect the levels of inflammatory cytokines,	*J Thromb Thrombolysis. 2021;52(1):9–17. (***)* [197]
Patients with AF from the ENGAGE AF-TIMI 48 trial. This randomized trial compared edoxaban versus warfarin in these patients and followed them for a median of 2.8 years.	Monotherapy with either edoxaban or warfarin	Relationship between neutrophil-to-lymphocyte ratio (NLR) and clinical outcomes in AF patients.	NLR	Systemic inflammation (as reflected by NLR) is associated with adverse outcomes in AF patients, and that edoxaban offers protection against some of these outcomes regardless of the inflammatory state	*Int J Cardiol. 2023;386:118–124.* [201]

The studies marked with (***) indicate no significant impact of NOAC on specific inflammatory markers, suggesting limited influence despite research efforts in this area.

Building on the aforementioned findings, it is pivotal to acknowledge the limitations that accompany them. A substantial proportion of the investigations are rooted in animal models. While these models are undeniably instrumental, they might not mirror human pathophysiological conditions in their entirety. This becomes especially pertinent when considering the effects of edoxaban on certain populations, such as HIV-positive individuals, where the anticipated anti-inflammatory outcomes were not significantly realized [198], thereby raising questions regarding the drug’s broad-spectrum applicability. Moreover, a noteworthy segment of the research, specifically those marked with (***) [196,197,198], was unable to demonstrate a pronounced impact of edoxaban on particular inflammatory markers. This underscores the imperative for more refined and nuanced analyses, to truly decipher the drug’s therapeutic breadth and limitations in diverse clinical scenarios.

Understanding the intricacies of edoxaban’s mechanisms remains a continuously unfolding domain. A salient aspect of this domain is the drug’s potential impact on oxidative stress markers, especially in the context of CKD. The alleviation of oxidative stress induced by FXa in human kidney (HK)-2 cells has emphasized the pivotal relationship between oxidative stress and inflammation [200]. Future endeavors could delve deeper into this aspect by leveraging patient-derived kidney organoids [202] or advanced cellular models like induced pluripotent stem cell (iPSC)-derived renal cells. Contrastingly, in vascular remodeling studies with ApoE^−/−^ mice, edoxaban did not display a significant influence on specific inflammatory markers [196]. This deviation from the general positive anti-inflammatory narrative suggests that further research is warranted. Explorations into alternative animal models, such as LDLr^−/−^ mice or Zebrafish models—renowned for their insights into vascular development—might prove enlightening.

Another discernible research gap pertains to the population diversity in existing studies. Though preliminary research on HIV-positive individuals [198] and atrial fibrillation patients from the ENGAGE AF-TIMI 48 trial [201] has been undertaken, the ambit of these studies needs broadening. Future research should factor in a wider range of populations, spanning diverse age brackets, ethnicities, and other comorbid conditions. Such an approach can ensure a comprehensive grasp of edoxaban’s therapeutic spectrum across varying clinical scenarios

### 4.3. Rivaroxaban’s Anti-Inflammatory Profile

Rivaroxaban demonstrates a significant anti-inflammatory effect, as outlined in various studies summarized in the attached Table 7. For instance, in Japanese subjects with NVAF, rivaroxaban was associated with an increase in thrombomodulin levels and a decrease in MMP-9 compared to warfarin, indicating its potent anti-inflammatory action [203]. Further examination of ex vivo abdominal aortic aneurysm samples revealed that rivaroxaban notably reduced markers of inflammation and oxidative stress, such as IL-6 and MMP-9, underscoring its potential in mitigating vascular inflammation [204]. An ancillary analysis of the X-TRA study also demonstrated rivaroxaban’s efficacy in reducing hsCRP, D-dimer, vWF, and TAT complex levels in AF patients, further supporting its anti-inflammatory capabilities [205]. Moreover, in a direct comparison between rivaroxaban and warfarin users with NVAF, rivaroxaban users showed decreased levels of inflammatory cytokines, highlighting a less evident inflammatory profile in these patients [206].

These findings are complemented by studies across various models, including ex vivo AAA models, clinical trials, and animal studies, consistently showing rivaroxaban’s role in modulating key inflammatory markers and pathways. The table provides a comprehensive overview of rivaroxaban’s anti-inflammatory effects (Table 7), showcasing its therapeutic potential beyond its conventional anticoagulant role.

### 4.4. Dabigatran’s Anti-Inflammatory Profile

Dabigatran exhibits diverse anti-inflammatory effects across different models and conditions, as delineated in Table 8. For instance, in patients undergoing elective orthopedic surgical procedures, dabigatran demonstrated comparable efficacy to LMWH in managing post-operative inflammation, with no significant differences in the incidence of local or systemic inflammatory complications [241]. In vitro analyses using human peripheral blood mononuclear cells (PBMCs) and whole blood revealed that dabigatran significantly reduced levels of pro-inflammatory cytokines, growth factors, and chemokines, suggesting potential anti-inflammatory effects by dampening pro-inflammatory stimuli [242].

Furthermore, in experimental models, such as the bleomycin-induced interstitial lung disease in female C57BL/6 mice, dabigatran markedly decreased thrombin activity, TGF-β1 levels, and the number of inflammatory cells in bronchoalveolar lavage fluid, alongside a reduction in histologically evident lung inflammation and fibrosis [243]. Similar anti-inflammatory and antifibrotic effects were observed in models of atherosclerosis and nonalcoholic fatty liver disease (NAFLD), where dabigatran attenuated hepatic fibrin deposition, inflammation, hepatocellular injury, and steatosis in mice fed a high-fat diet, suggesting that thrombin may drive NAFLD pathogenesis by altering expression of genes associated with lipid metabolism and bile acid synthesis [244].

Interestingly, recent studies have established PAR1, the primary thrombin receptor, is a pivotal connection between coagulation and inflammation in the pathophysiology of COVID-19 [245]. Activation of PAR1 contributes to the hyperinflammatory state observed in severe cases by excessively stimulating PAR receptors through elevated levels of activated proteases, resulting in increased inflammation and coagulopathy [246]. Moreover, higher levels of autoantibodies against PAR1 in severe COVID-19 patients may act as allosteric agonists on endothelial cells and platelets, potentially causing microthrombosis [247]. PAR1 activation on endothelial cells can also elevate IL-6 expression via the PAR1/p70S6K/ERK pathway, though this might not significantly impact systemic IL-6 levels. Additionally, PAR1-dependent platelet activation leads to increased aggregation of circulating cells and collagen, contributing to a prothrombotic state [247]. Dysregulated PAR signaling, particularly PAR1, is linked to the development of acute respiratory distress syndrome (ARDS) and poor prognosis in COVID-19 patients [248]. Thus, targeting thrombin, and consequently PAR1, through dabigatran, a direct thrombin inhibitor, can be proposed as a potential therapeutic strategy to mitigate coagulopathy and inflammatory dysfunctions in COVID-19, offering perspectives on novel therapeutic approaches [248]. Furthermore, understanding the role of PAR receptors in COVID-19 inflammation elucidates the complex interplay between coagulation and inflammation in the disease, potentially leading to new therapeutic approaches.

These observations are supported by studies across various models, including ex vivo and in vivo models of pulmonary fibrosis, atherosclerosis, and liver injury, consistently demonstrating dabigatran’s role in modulating key inflammatory markers and pathways. Table 8 provides a detailed overview of dabigatran’s anti-inflammatory effects, underscoring its therapeutic potential beyond anticoagulation by reducing inflammation in cardiovascular diseases, interstitial lung disease, NAFLD, COVID-19 and potentially other inflammatory conditions.

**Table 8 ijms-25-08727-t008:** Anti-inflammatory profile of dabigatran.

**Human Model or Population**	**Mono/Combination Therapy**	**Condition Investigated**	**Inflammatory Markers Examined**	**Effect Observed**	**Reference**
Patients who had elective orthopedic surgical procedures, specifically total hip or knee arthroplasty.	Monotherapy (dabigatran vs. low-molecular weight heparin)	Thromboembolic complications post-total hip or knee arthroplasty	▪ State of local inflammation, C-reactive protein levels	No significant difference in the incidence of local or systemic inflammatory complications between dabigatran and low-molecular weight heparin	*Pol Orthop Traumatol. 2012 (***)* [241]
▪ Human peripheral blood mononuclear cells (PBMCs) and ▪ human whole blood in vitro model	Monotherapy	Impact of dabigatran on pro-inflammatory cytokines, growth factors, and chemokines; exploring its potential anti-inflammatory effects.	Angiogenin, CXCL1/GRO alpha, IL-1β/IL-1F2, human IL-6, IL-8/CXCL8, CCL2/MCP-1, TGFβ 1, TNF-α, Vascular endothelial growth factor (VEGF).	▪ Dabigatran significantly lowered supernatant protein levels in PBMCs. ▪ Notably decreased the growth factor and chemokine levels in PBMCs that were treated with 200 ng/mL thrombin in a dose-dependent manner. ▪ Showed potential to reduce proinflammatory stimuli via reduced expression of cytokines and chemokines. In summary, the study suggests that dabigatran could have anti-inflammatory effects by reducing the expression of pro-inflammatory cytokines, growth factors, and chemokines	*Life Sci. 2020 Dec 1;262:118474.* [242]
**Cell, Mice, or Combined Models**	**Mono/Combination Therapy**	**Condition Investigated**	**Inflammatory Markers Examined**	**Effect Observed**	**Reference**
Female C57BL/6 mice	Mono therapy	▪ Interstitial lung disease induced by bleomycin	▪ Thrombin activity, levels of TGF-β1, number of inflammatory cells, protein concentrations in bronchoalveolar lavage (BAL) fluid. ▪ Additionally, collagen, connective tissue growth factor, and α-smooth muscle actin expression were examined.	▪ Reduced thrombin activity ▪ Reduced levels of transforming growth factor β1 in BAL fluid ▪ Reduced number of inflammatory cells and protein concentrations in BAL fluid ▪ Histologically evident lung inflammation and fibrosis were significantly decreased ▪ Reduced expression of collagen, connective tissue growth factor, and α-smooth muscle actin ▪ Marked anti-inflammatory and antifibrotic effects observed in a bleomycin model of pulmonary fibrosis	*Arthritis Rheum. 2011 May;63(5):1416–25.* [243]
Transgenic atherosclerosis-prone mice with diminished coagulant or hypercoagulable phenotype (FII(-/WT):ApoE(-/-) and TM(Pro/Pro):ApoE(-/-))	Monotherapy (dabigatran etexilate or recombinant APC)	▪ Onset and progression of atherosclerosis ▪ Plaque phenotype determination	▪ Leukocyte infiltration ▪ Collagen and vascular smooth muscle cell content ▪ Neutrophilia, neutrophil hyper-reactivity, oxidative stress, neutrophil intraplaque infiltration, and apoptosis	▪ Counteracted the pro-inflammatory and pro-atherogenic phenotype of pro-thrombotic TM(Pro/Pro):ApoE(-/-) mice	*PLoS One. 2013;8(2):e55784.* [249]
▪ Cerebrovasculature of transgenic Alzheimer’s disease (AD) mice ▪ Cultured brain endothelial cells	Monotherapy (dabigatran)	▪ Inflammatory protein expression in the cerebrovasculature of AD mice ▪ Role of thrombin in cerebrovascular inflammation and oxidative stress in AD ▪ Hypoxia-induced changes in brain endothelial cells	▪ Thrombin ▪ Hypoxia-inducible factor 1α (HIF-1α) ▪ IL-6 ▪ MCP-1 ▪ MMPs ▪ Reactive oxygen species (ROS)	▪ Significantly decreased expression of inflammatory proteins and ROS in AD mice ▪ Reduced ROS generation and inflammatory protein expression in endothelial cells exposed to hypoxia	*Front Aging Neurosci. 2013 May 9;5:19.* [250]
▪ C57BL/6J mice	Monotherapy	▪ Nonalcoholic fatty liver disease (NAFLD) induced by a high-fat diet (HFD)	▪ Hepatic fibrin deposition. ▪ Serum alanine aminotransferase levels. ▪ Hepatic inflammation ▪ Hepatocellular injury	▪ Dabigatran significantly reduced hepatic fibrin deposition, hepatic inflammation, hepatocellular injury, and steatosis in mice fed an HFD. ▪ Dabigatran also attenuated HFD-induced body weight gain. ▪ Gene expression analysis suggested that thrombin may drive NAFLD pathogenesis by altering the expression of genes associated with lipid metabolism and bile acid synthesis.	*J Pharmacol Exp Ther. 2014 Nov;351(2):288–97.* [244]
▪ ApoE(-/-) mice that are prone to atherosclerosis were used as the experimental model.	Monotherapy with the thrombin inhibitor dabigatran at a dosage of 1.2 g/kg/day.	▪ The study is aimed at understanding the role of thrombin in atherosclerosis and whether its inhibition with dabigatran could provide therapeutic benefits.	▪ Thrombin time via HEMOCLOT^®^. ▪ Vascular oxidative stress assessed by L012 chemiluminescence. ▪ Endothelium-derived vasorelaxation. ▪ Histological and immuno-histological analyses of atherosclerotic lesions and macrophage infiltration.	▪ Thrombin time was significantly extended in the dabigatran-treated group. ▪ Vascular oxidative stress was significantly reduced as indicated by L012 chemiluminescence in aortic segments. ▪ Improved endothelium-derived vasorelaxation was observed in dabigatran-treated mice. ▪ Treated mice developed fewer atherosclerotic lesions and showed less macrophage infiltration in these lesions. ▪ Blood pressure, body weight, and food intake were unaffected by the treatment. **Conclusion:** The study concludes that dabigatran, a thrombin inhibitor, effectively reduces vascular oxidative stress and inflammation.	*Arch Med Sci. 2014 Feb 24;10(1):154–60.* [251]
▪ ApoE-/- mice of different ages as the experimental model.	Monotherapy	▪ Initiation and progression of atherosclerosis and its impact on the expression of the proinflammatory cytokine oncostatin M.	The mean lesion area in the aortic sinus and the innominate artery. ▪ Presence of CD45-positive cells in aortic tissue measured by flow cytometry. ▪ Expression of oncostatin M measured by immunocytochemistry.	▪ Significant reduction in the mean area of atherosclerotic lesions in both young and older mice treated with dabigatran etexilate compared to controls. ▪ Fewer CD45-positive cells were observed in the aortas of dabigatran-treated mice. ▪ Enhanced Nitric Oxide (NO) production in endothelial cells pretreated with dabigatran. ▪ Reduced expression of oncostatin M in the lesions of dabigatran etexilate-treated mice. The study concluded that inhibition of thrombin with dabigatran etexilate retards the initiation and progression of atherosclerotic lesions in ApoE-/- mice. The treatment also improved endothelial function and reduced the accumulation of macrophages within the vascular wall.	*Drug Des Devel Ther. 2015 Sep 10;9:5203–11.* [252]
▪ HDM-based (house dust mite) murine asthma model.	Monotherapy using the oral thrombin inhibitor dabigatran (10 mg/g)	▪ Chronic allergic lung inflammation in asthma triggered by house dust mite (HDM) allergens.	▪ TAT complexes (markers of coagulation activation). ▪ D-dimer levels (markers of fibrin degradation). ▪ Eosinophils and neutrophils influx into the lungs. ▪ Mucus production in the airways. ▪ T helper 2 response as measured by bronchoalveolar IL-4 and IL-5 levels. ▪ Systemic rise in IgE and HDM-IgG1.	▪ Dabigatran modestly improved HDM-induced lung pathology (*p* < 0.05). ▪ Dabigatran significantly decreased IL-4 levels (*p* < 0.01). ▪ Dabigatran did not inhibit HDM-evoked coagulation activation in the lung. ▪ No impact on eosinophils and neutrophils influx, mucus production, IL-5 levels, or systemic IgE and HDM-IgG1 levels. The study found that while dabigatran was successful in achieving systemic thrombin inhibitory activity similar to human trials, it did not significantly mitigate allergic lung inflammation triggered by HDM in a murine asthma model. ***Given the limited effects of dabigatran on the evaluated markers of inflammation and asthma pathology, the study suggests that dabigatran might not be a suitable candidate for clinical evaluation in asthma treatment.***	*Am J Physiol Lung Cell Mol Physiol. 2015 Oct 15;309(8):L768–75. (***)* [253]
▪ Adult healthy mice ▪ Mouse model of AD	Monotherapy (dabigatran etexilate or warfarin)	▪ Thrombin-dependent neuroinflammation in a healthy CNS ▪ Glial activation in Alzheimer’s disease	▪ Astrocyte GFAP (Glial Fibrillary Acidic Protein) ▪ Microglia Iba-1 ▪ Sulfatide levels ▪ Cerebellar gene expression	▪ Reduced expression of astrocyte GFAP and microglia Iba-1 in healthy CNS. ▪ Reduced glial activation in Alzheimer’s disease model, but no change in amyloid plaque burden	*J Neuroimmunol. 2016 Aug 15;297:159–68.* [254]
▪ Wistar rats	Monotherapy with dabigatran etexilate at a dosage of 15 mg/kg.	▪ The study aimed to investigate the effects of pre-treatment with dabigatran on outcomes following ischemic stroke.	▪ Infarct volume, neurologic outcome, and intracranial hemorrhage (ICH). ▪ Thrombin generation indirectly assessed by measuring the thrombin/antithrombin III complex. ▪ Microvascular patency evaluated histologically. ▪ Cytokine expression and CD68-immunoreactivity. ▪ BBB integrity examined by quantifying brain oedema.	▪ Rats treated with Dabigatran had a significant reduction in infarct size and recovery of neurologic deficits compared to controls, without an increase in ICH. ▪ Dabigatran administration led to decreased thrombin generation and thrombus formation. ▪ There was dampened CD68-immunoreactivity and attenuated pro-inflammatory cytokine expression in the cerebral parenchyma ipsilateral to the ischemic lesion. ▪ BBB permeability remained unaltered after treatment with dabigatran. **Conclusion:** The study concluded that prophylactic anticoagulation with dabigatran etexilate improves outcomes following ischemic stroke in rats by reducing thrombin-induced inflammation and thrombus formation.	*Curr Neurovasc Res. 2016;13(3):199–206.* [255]
Vascular leak-dependent mice model for pulmonary fibrosis	Monotherapy	▪ Fibrotic lung disease, specifically idiopathic pulmonary fibrosis (IPF), in the context of vascular leak after lung injury	▪ Thrombin activity ▪ PAR1 activation ▪ Integrin αvβ6 induction ▪ TGF-β activation ▪ Fibrin accumulation visualized using ultrashort echo time (UTE) lung MRI	▪ Significant inhibition PAR1 activation ▪ Reduced integrin αvβ6 induction ▪ Lowered TGF-β activation ▪ Attenuation of fibrin deposition ▪ Inhibition of the development of pulmonary fibrosis In summary, dabigatran exhibited significant anti-inflammatory effects in a vascular leak-dependent mouse model for fibrotic lung disease.	*JCI Insight. 2017 May 4;2(9):e86608.* [256]
▪ Fibγ390-396A mice and Homozygous thrombomodulin-mutant ThbdPro mice ▪ Obese patients (for immunohistochemical studies)	Monotherapy	▪ High-fat diet-induced obesity, fatty liver disease, and related sequelae	▪ Extravascular fibrin deposits within white adipose tissue and liver ▪ Systemic, adipose, and hepatic inflammation ▪ Macrophage counts within white adipose tissue ▪ Weight gain and elevated adiposity	▪ Limited high-fat diet-induced obesity development ▪ Suppressed progression of obesity-related sequelae ▪ Diminished systemic, adipose, and hepatic inflammation ▪ Reduced macrophage counts within white adipose tissue In summary, dabigatran treatment showed anti-inflammatory effects in a high-fat diet-induced obesity model in mice. It limited weight gain, reduced systemic inflammation, and decreased macrophage counts in adipose tissue, thereby potentially suppressing the progression of obesity-associated diseases.	*J Clin Invest. 2017 Aug 1;127(8):3152–3166.* [257]
▪ Mouse model immunized with topoisomerase I dendritic cells (TOPOIA DCs), serving as an experimental model for Systemic Sclerosis (SSc).	Mono therapy (dabigatran administered either during the onset of fibrotic phase (late treatment) or inflammatory phase (early treatment)).	▪ Effect of dabigatran on TOPOIA DCs-induced lung and skin fibrosis, aimed at understanding its potential role in Systemic Sclerosis (SSc).	▪ Thrombin levels in lungs. ▪ Gene expression markers including Col5a1, Timp1, Tweakr, Vwf, Il6, Il33, Il4, and Ifng.	▪ Early administration of dabigatran led to aggravation of pulmonary fibrosis with signs of severe perivascular inflammation. ▪ Late administration of dabigatran was not protective against fibrosis. ▪ Thrombin levels in lungs were increased in TOPOIA DCs immunized group and were further augmented by dabigatran. ▪ Early dabigatran treatment exacerbated skin fibrosis and induced a profibrotic and inflammatory skin gene expression signature. In summary, dabigatran had a detrimental effect, exacerbating lung and skin fibrosis in a TOPOIA DCs-induced model of SSc. The study concludes that dabigatran may not be a suitable treatment for patients with Systemic Sclerosis based on these experimental findings.	*Clin Exp Rheumatol. 2017 Sep-Oct;35 Suppl 106(4):35–39. (***)* [258]
▪ ARPE-19 cells ▪ Vitreous fluids (with and without thrombin activity)	Monotherapy (dabigatran)	▪ Proliferative vitreoretinopathy (PVR) and its associated fibrosis and inflammation due to thrombin activity	▪ CCL2, CXCL8, GM-CSF, IL-6, PDGF-BB mRNA expression (RQ-PCR) ▪ Protein levels of 27 cytokines, chemokines, and growth factors in culture supernatants	▪ Inhibited thrombin and vitreous-induced expression of CCL2, CXCL8, GM-CSF, IL-6, and PDGF-BB in ARPE-19 cells ▪ Inhibited thrombin activity in vitreous fluids after oral intake	*Acta Ophthalmol. 2018 Aug;96(5):452–458.* [259]
▪ 24 Sprague-Dawley rats	Monotherapy Group 1: Control (No treatment) Group 2: Dabigatran etexilate (10 mg/kg orally for 7 days) Group 3: Bemiparin sodium (250 IU/kg subcutaneously for 7 days)	▪ Revascularization effects post carotid artery anastomosis	▪ Lumen diameter, lumen area, tunica media thickness, edema, vessel wall injury, intimal hyperplasia, thrombus, inflammation	▪ INCREASED edema and inflammation ▪ Did not prevent intimal hyperplasia ▪ Anticoagulation effect was more pronounced than antithrombotic effect	*World Neurosurg. 2019 Jun;126:e731-e735. (***)* [260]
▪ Female Low density lipoprotein receptor knockout (Lldr-/-) mice	Monotherapy	▪ Diet-induced obesity and atherosclerosis	▪ Adipocyte hypertrophy ▪ Pro-inflammatory M1-polarized macrophages in adipose tissue ▪ Abundance of pro-inflammatory M1 macrophages in the aortic wall ▪ Multiple circulating cytokines TNF-α, IL-1β, and IL-6	▪ Reduced numbers of pro-inflammatory M1-polarized macrophages in adipose tissue ▪ Decreased abundance of pro-inflammatory M1 macrophages in the aortic wall ▪ Reduction in multiple circulating cytokines ▪ Increased adipocyte hypertrophy (although not necessarily anti-inflammatory, worth noting) In summary, dabigatran treatment showed anti-inflammatory effects by reducing the numbers of pro-inflammatory M1 macrophages both in adipose tissue and in the aortic wall, contributing to plaque stabilization.	*Atherosclerosis. 2019 Aug;287:81–88.* [261]
▪ Rat model of severe aortic valve *Staphylococcus aureus* infective endocarditis (IE)	Combination therapy (dabigatran with gentamicin)	▪ *S. aureus*-induced IE on the aortic valve	▪ Keratinocyte-derived chemokine (Human IL-8 analogue) ▪ IL-6 ▪ IL-1β ▪ ICAM-1 ▪ Tissue inhibitor of metalloproteinase (TIMP)-1 ▪ L-selectin	▪ Reduced valve vegetation size ▪ Reduced bacterial load in aortic valves ▪ Lowered expression of pro-inflammatory markers: keratinocyte-derived chemokine, IL-6, ICAM-1, TIMP-1, L-selectin ▪ 2.5-fold increase in circulating platelets ▪ Elevated expression of functional and activated platelets not bound to neutrophil.	*PLoS One. 2019 Apr 19;14(4):e0215333.* [262]
▪ Rat model (both diabetic and control rats)	Monotherapy (dabigatran)	▪ Impact of long-term thrombin inhibition in diabetes and its relation to atherosclerosis and atherothrombotic risk	▪ High-sensitivity CRP ▪ VWF ▪ VEGF ▪ Fibronectin ▪ PAR4 ▪ Agonist-induced platelet aggregation ▪ Aortic and coronary lipid deposits	▪ Increased levels of high-sensitivity C-reactive protein, VWF, VEGF, and fibronectin in dabigatran-treated control and diabetic rats ▪ Lower PAR4 agonist-induced aggregation in DE-treated control rats ▪ Higher ADP-induced aggregation in dabigatran-treated diabetic rats ▪ Aortic lipid deposits in dabigtran-treated control rats ▪ Coronary lipid deposits in 75% of dabigatran-treated diabetic rats ▪ Increased vascular PAR4 expression in both dabigtran-treated control and diabetic rats In summary, the study indicates that long-term treatment with dabigatran etexilate may increase vascular PAR4 expression and promote mechanisms related to atherosclerosis. Particularly in diabetic rats, it was found to increase platelet aggregation and favor the occurrence of coronary lipid deposits. These findings suggest that long-term thrombin inhibition could increase the risk of atherosclerotic and atherothrombotic events, especially in the presence of diabetes.	*J Thromb Haemost. 2019 Mar;17(3):538–550.* [263]
▪ Tau-based mouse model (Tg4510)	Monotherapy (dabigatran)	▪ Effects of inhibiting thrombin on oxidative stress, inflammation, and AD-related proteins in the Tg4510 mouse model	▪ iNOS ▪ NOX4 ▪ Phosphorylation of tau (S396, S416)	▪ Reduced iNOS, NOX4, and phosphorylation of tau (S396, S416) ▪ Increased expression of several signaling proteins related to cell survival and synaptic function	*Biochem Biophys Rep. 2020 Nov 19;24:100862.* [264]
▪ Primary astrocytes isolated from mice ▪ Mouse model of Experimental autoimmune encephalomyelitis (EAE)	Monotherapy (dabigatran)	▪ Correlation between coagulation cascade and CNS immune diseases. ▪ CNS demyelination	▪ Thrombin-induced activation of astrocytes ▪ PAR-1 ▪ Sphingosine-1-phosphate (S1P) ▪ Sphingosine kinases (SphKs) ▪ IL-1β	▪ Suppressed thrombin-induced activation of astrocytes ▪ Recovered neurological function ▪ Reduced inflammation in the spinal cord ▪ Prevented spinal cord demyelination caused by EAE ▪ Limited activation of PAR-1, downregulated SphK1, and disrupted S1P receptor signaling	*Front Mol Neurosci. 2020 Jun 30;13:114.* [265]
▪ STZ-induced diabetic mice ▪ Human umbilical endothelial cells (HUVEC)	Monotherapy (dabigatran etexilate)	▪ Endothelial dysfunction in diabetic mice	▪ PAR1, PAR3, and PAR4 in the aorta ▪ MCP-1 ▪ ICAM-1 ▪ VCAM ▪ Phosphorylation of IκBα ▪ Phosphorylation of eNOSSer1177	▪ Attenuated endothelial dysfunction in diabetic mice with no change in metabolic parameters including blood glucose ▪ Decreased expression of MCP-1 and ICAM-1 in the aorta of diabetic mice ▪ In HUVEC, reduced thrombin-induced inflammatory molecule expression and decreased phosphorylation of IκBα	*Vascul Pharmacol. 2020* [266]
▪ Unilateral ureteral obstruction or UUO-induced renal injury mice model	Monotherapy	▪ CKD with focus on renal injury and tubulointerstitial fibrosis (TIF)	▪ Collagen-1 ▪ PAR-1 ▪ TGF-β ▪ SNAI2 ▪ Histological examination for epithelial-mesenchymal transition (EMT) program	▪ Significant inhibition of UUO-induced collagen-1 and TIF ▪ Reduction of thrombin-activated PAR-1 expression in the fibrotic kidney ▪ Improved histoarchitecture of the obstructed kidney ▪ Inhibited TGF-β and SNAI2-induced epithelial-mesenchymal transition (EMT) In summary, dabigatran was found to be effective in inhibiting renal injury and fibrosis in a mouse model. It significantly reduced markers associated with inflammation and fibrosis, suggesting its potential as a therapeutic agent for chronic kidney disease, particularly in conditions involving tubulointerstitial fibrosis.	*Eur J Pharmacol. 2021 Feb 15;893:173838.* [267]
▪ **Clinical patients who underwent hepatectomy.** ▪ In vivo mice model undergoing 60-min hepatic partial-warm ischemia/reperfusion injury (IRI). ▪ In vitro model with hepatic sinusoidal endothelial cells (SECs) and hepatocytes.	Monotherapy	▪ Hepatic ischemia/reperfusion injury (IRI), specifically focusing on sinusoidal endothelial cells (SECs).	▪ TAT for thrombin activity. ▪ Lactate dehydrogenase levels for cytotoxicity. ▪ Liver histological damage. ▪ Endogenous thrombomodulin (TM) levels. ▪ Serum high-mobility group box-1 (HMGB-1) levels.	▪ Dabigatran significantly improved liver histological damage and provided antiapoptotic and anti-inflammatory effects. ▪ Enhanced expression of endogenous TM. ▪ Reduced levels of serum HMGB-1, a marker of inflammation. ▪ In vitro, dabigatran attenuated hypoxia-reoxygenation (H-R) damage in SECs and enhanced TM expression while reducing extracellular HMGB-1. The study concluded that dabigatran treatment can mitigate the damage caused by hepatic IRI by protecting sinusoidal endothelial cells (SECs). The treatment not only improved liver histological damage but also provided antiapoptotic and anti-inflammatory effects. The drug increased the expression of endogenous thrombomodulin (TM) and reduced the levels of serum high-mobility group box-1 (HMGB-1), which are both linked to inflammation and tissue damage.	*Liver Transpl. 2021 Feb;27(3):363–384.* [268]
▪ **Human peripheral blood collected from clinical sepsis patients and healthy controls.** ▪ Mouse sepsis models using zymosan-induced peritonitis and cecal ligation and puncture (CLP).	Monotherapy (warfarin, dabigatran, and heparin are each used independently).	▪ Investigating the role of commonly used anticoagulants (warfarin, dabigatran, and heparin) in the resolution of inflammation in sepsis.	▪ Transmembrane Fgl2 (mFgl2) ▪ Soluble Fgl2 (sFgl2) ▪ Peripheral blood mononuclear cells ▪ Prostaglandin E2 ▪ Leukotriene B4 ▪ Lipid metabolites ▪ Inflammation- and coagulation-related indexes	▪ Dabigatran and warfarin attenuated zymosan-induced peritonitis, but heparin did not. ▪ Dabigatran improved sepsis survival in the CLP mouse model. ▪ Dabigatran increased levels of sFgl2 at both the initiation and resolution phases of inflammation. ▪ Mechanistically, dabigatran promotes the shedding of sFgl2, enhancing the biosynthesis of a specialized pro-resolving mediator (RvD5n-3 DPA) via the STAT6-ALOX15 axis. In summary, dabigatran shows dual anti-inflammatory and pro-resolving actions in sepsis through promoting sFgl2-triggered RvD5n-3 DPA production, implying it could be beneficial for promoting tissue homeostasis in sepsis.	*Theranostics. 2021 Feb 20;11(9):4251–4261* [269]
▪ Female Apoe-/- mice (age 12 weeks)	Monotherapy Control (Western-type diet only) Dabigatran etexilate (duration: 6 or 18 weeks) Warfarin (duration: 6 or 18 weeks)	▪ Atherosclerosis progression and calcification	▪ Vascular calcification (measured with µCT and [18 F]-NaF) ▪ Atherosclerotic burden (assessed by (immuno)histochemistry) ▪ Uncarboxylated matrix Gla protein ▪ VSMC oxidative stress and extracellular vesicle release	▪ Reduced plaque progression compared to control ▪ Prevented VSMC oxidative stress and extracellular vesicle release (in vitro) ▪ Beneficial effects on atherosclerosis progression and calcification	*J Thromb Haemost. 2021 May;19(5):1348–1363.* [23]
▪ Female Apoe-/- mice (age 12 weeks) ▪ Primary vascular smooth muscle cells (VSMC) in vitro	Monotherapy (dabigatran etexilate or Warfarin)	▪ Atherosclerosis progression ▪ Vascular calcification	▪ Pro-inflammatory phenotype in atherosclerotic lesions ▪ Uncarboxylated matrix Gla protein ▪ Extracellular vesicle release ▪ Cytokines—IFNγ, IL-10, IL-4, IL-5, IL-6, TNF-α and KC/GRO (Keratinocyte-derived Cytokine/Growth-Regulated Oncogene) refers to a group of chemokines in mice that are analogous to the human CXCL1, CXCL2, and CXCL3	▪ Significantly reduced plaque progression compared to control. ▪ Did not exhibit the pro-inflammatory or calcification effects observed with warfarin. ▪ Prevented VSMC oxidative stress and extracellular vesicle release induced by warfarin and thrombin in vitro	*J Thromb Haemost. 2021 May;19(5):1348–1363.* [23]
Retinoic acid (RA)-differentiated human neuroblastoma cell line SH-SY5Y	Monotherapy (250 nM dabigatran with/without 10–100 nM thrombin)	▪ Neuro-inflammation and AD related pathology due to vascular and non-vascular risk factors	▪ DNA binding of NFκB ▪ Aβ protein expression, p38 MAPK phosphorylation, caspase 3, APP, total Tau, phosphorylated Tau ▪ BACE1 activity, GSK3β expression, Amyloid Precursor Protein (APP), Beta-Secretase 1 (BACE1), Tau, and Glycogen Synthase Kinase 3 Beta (GSK3β) mRNA levels	▪ Reduced thrombin-induced DNA binding of NFκB by 175% (50 nM thrombin) and by 77% (100 nM thrombin) ▪ Attenuated thrombin-induced changes in protein, mRNA, and activities of the aforesaid markers by −31% to −283%	*Cereb Circ Cogn Behav. 2021 May 6;2:100014* [270]
▪ **Human tissues from Crohn’s disease patients and healthy controls.** ▪ Wild-type and protease-activated receptor-deficient mice. ▪ Rat and mouse models induced with colitis via intracolonic administration of trinitrobenzene sulphonic acid.	Monotherapy	▪ The role of elevated thrombin activity in local tissue dysfunction in Crohn’s disease and experimental colitis.	▪ Thrombin levels. ▪ Tissue damage indicators. ▪ PAR-1 and -4.	▪ Elevated thrombin levels were found in the tissues of Crohn’s disease patients and were linked to intestinal epithelial cells. ▪ Experimental colitis in rats also showed increased thrombin activity and expression. ▪ Thrombin activity led to mucosal damage and tissue dysfunction in mouse models through protease-activated receptors −1 and −4. ▪ Inhibition of thrombin with dabigatran, as well as inhibition of protease-activated receptor-1, prevented induced colitis in rodent models. In summary, elevated thrombin activity appears to contribute to tissue damage and dysfunction in Crohn’s disease. The study highlights the potential therapeutic benefits of inhibiting thrombin and protease-activated receptor-1 as a strategy for treating inflammatory bowel disease.	*J Crohns Colitis. 2021 May 4;15(5):787–799.* [271]
Pig coronary artery stenting model	Combination therapy (dabigatran, aspirin, and clopidogrel)	▪ Effects of stenting on coronary arteries (vasomotion, endothelialisation, neointimal formation)	▪ Thrombin generation, tissue burden (degree of peri-strut structure-thrombus and/or fibrin). ▪ Endothelium-dependent vasodilation, endothelialisation speed.	▪ Reduced thrombin generation (*p* < 0.001) ▪ Lower tissue burden (degree of peri-strut structure-thrombus and/or fibrin) (*p* = 0.031) ▪ Improved endothelium-dependent vasodilation after 3 days post-PCI (77 ± 40% vs. 41 ± 31%, *p* = 0.02) In summary, the short-term peri-interventional triple therapy with dabigatran, aspirin, and clopidogrel enhanced endothelium-dependent vasodilation and reduced thrombin generation and tissue burden, showing anti-inflammatory effects in a porcine model of coronary artery stenting.	*Front Cardiovasc Med. 2021 Jul 2;8:690476.* [272]
▪ Murine model of breast cancer metastasis (BALB/c mice) and ▪ in vitro human lung microvascular endothelial cell (HLMVEC) cultures.	Monotherapy	▪ The effect of dabigatran on pulmonary endothelial barrier integrity and metastatic spread in a murine model of breast cancer metastasis.	▪ Pulmonary endothelium permeability ▪ Platelet reactivity ▪ Lung fibrin deposition ▪ INFγ (Interferon γ) ▪ Complement activation ▪ Pulmonary metastasis	▪ Dabigatran-treated mice had more metastases in their lungs and showed increased pulmonary endothelium permeability. ▪ The treatment was not associated with altered lung fibrin deposition, changes in INFγ, or complement activation. ▪ In vitro, dabigatran inhibited platelet-mediated protection of pulmonary endothelium. In summary, in this murine model of breast cancer metastasis, dabigatran treatment appears to have a negative effect by promoting pulmonary metastasis. This is attributed to its inhibition of platelet-dependent protection of pulmonary endothelial barrier integrity. ***No significant anti-inflammatory effects were observed.***	*Front Pharmacol. 2022 Feb 28;13:834472(***)* [273]
Rat model of complete Freund’s adjuvant (CFA)-induced arthritis.	Monotherapy	▪ Rheumatoid arthritis characterized by joint inflammation, swelling, and dysfunction.	▪ Articular surface deformities. ▪ Reduced cartilage thickness. ▪ Loss of intercellular matrix. ▪ Inflammatory cell infiltration. ▪ Increased levels of anti-cyclic citrullinated peptide antibody (ACPA). ▪ Elevated oxidative stress levels. ▪ Tissue Receptor activator of nuclear factor-kappa B ligand (RANKL) levels. ▪ Elevated proteins of the kallikrein-kinin system (KKS). ▪ Toll-like receptor 4 (TLR4) expression.	▪ Dabigatran inhibited thrombin activity, which led to the inhibition of the kallikrein-kinin system (KKS). ▪ This inhibition reduced TLR4 expression and subsequently decreased RANKL levels. ▪ Observed anti-inflammatory and antioxidant effects. The study concluded that dabigatran effectively mitigated the inflammation and oxidative stress involved in CFA-induced arthritis in rats. By inhibiting thrombin, dabigatran indirectly inhibited the kallikrein-kinin system and reduced TLR4 expression, which had downstream effects on reducing RANKL levels and inflammation. Therefore, dabigatran may present a novel therapeutic strategy for the treatment of rheumatoid arthritis.	*Int J Mol Sci. 2022 Sep 7;23(18):10297.* [274]
Mouse model (T7K24R trypsinogen mutant mouse and T7D23A mouse)	Monotherapy	▪ Trypsin-dependent pancreatitis	▪ Histological normalization in the pancreas as a sign of reduced inflammation.	▪ Dabigatran etexilate showed therapeutic efficacy in reducing pancreatitis in T7K24R mice. However, it was not effective in the T7D23A mice, which have a more aggressive form of the disease.	*JCI Insight. 2022 Nov 8;7(21):e161145.* [275]
Pig stenting coronary artery model	Combination therapy: dabigatran with dual antiplatelet therapy (DAPT) (clopidogrel 75 mg + aspirin 100 mg)	▪ Local inflammation, disturbed vasomotion, slowed endothelialisation post-coronary artery stenting	▪ Vasoconstriction post-PCI ▪ Optical coherence tomography (OCT) ▪ Angiography ▪ In vitro myometry ▪ Histomorphometry	▪ Significantly increased vasoconstriction at 3 days post-PCI ▪ No significant impact on endothelium-dependent and -independent vasodilatation, OCT, angiography, or histomorphometry results	*J Pers Med. 2023 Jan 31;13(2):280* [276]

The studies marked with (***) indicate no significant impact of NOAC on specific inflammatory markers, suggesting limited influence despite research efforts in this area.

### 4.5. Comparative Anti-Inflammatory Potential of NOACs

The comparative analysis, as detailed in the Table 9, reveals the nuanced anti-inflammatories—dabigatran, rivaroxaban, apixaban, and edoxaban—across various models, including sickle cell disease (SCD), endothelial cell inflammation, acute ischemic stroke, and more. For example, in the Berkeley mouse model of SCD, rivaroxaban attenuated systemic inflammation by decreasing plasma levels of IL-6, whereas dabigatran did not significantly affect IL-6 levels but impacted neutrophil infiltration in the lung, delineating their specific pathways of action [277]. Another study in human umbilical vascular endothelial cells (HUVECs) underscored that both dabigatran and rivaroxaban suppressed inflammatory gene expression post-thrombin stimulation, indicating their potential anti-inflammatory capabilities [278].

A critical observation from the studies is the differential impact of these anticoagulants on specific inflammatory markers, suggesting a complex interplay between coagulation and inflammation pathways. For instance, dabigatran seemed to reduce hs-CRP and increase pentraxin-3 levels more prominently than apixaban in acute ischemic stroke patients, highlighting potential variances in their anti-inflammatory profiles [279]. Moreover, while both dabigatran and rivaroxaban were observed to reduce aortic clamping-induced renal tissue oxidation and inflammation in a Wistar rat model, further research is warranted to delineate their distinct mechanistic pathways and the clinical relevance of these findings [280].

Critically, while Table 9 provides valuable insights into the anti-inflammatory effects of these NOACs, it also underscores the necessity for further investigation. Notably, the variability in the effects observed across different inflammatory markers, conditions, and models calls for more comprehensive studies to fully understand the therapeutic potential and limitations of NOACs in modulating inflammation. Additionally, the potential impact of NOACs on the progression of conditions like colorectal cancer, as seen with edoxaban’s superior ability to suppress tumor growth and mitigate elevated levels of inflammatory markers, including PAR2, signal transducer and activator of transcription-3 (STAT3), cyclin D1, and Ki67, warrants further exploration to elucidate the broader implications of these findings for clinical practice [281].

**Table 9 ijms-25-08727-t009:** Comparative anti-inflammatory profile of NOACs.

Model or Population	NOACs Used	Condition Investigated	Inflammatory Markers Examined	Effect Observed	Reference
The study utilized the Berkeley (BERK) mouse model of Sickle Cell Disease (SCD) which has specific genetic modifications involving human and murine globins. For certain experiments, wild-type (WT) mice and PAR-1 and PAR-2 deficient mice were used. Additionally, bone marrow (BM) transplantation was conducted where irradiated PAR-1 and PAR-2 mice received bone marrow cells from either the BERK or WT control mice.	Dabigatran and rivaroxaban	Sickle Cell Disease (SCD)	IL-6, sVCAM-1 was assessed in relation to the activation of TF in sickle mice, MPO which is an enzyme primarily found in neutrophils and is released during the activation of neutrophils.	▪ FXa inhibition (via rivaroxaban) and deficiency in PAR-2 on nonhematopoietic cells attenuated systemic inflammation, as measured by decreased plasma levels of IL-6. ▪ Thrombin inhibition (via dabigatran) did not affect plasma IL-6 levels in the sickle mice. However, thrombin was found to contribute to neutrophil infiltration in the lung, independently of PAR-1 expressed by nonhematopoietic cells.	*Blood. 2014;123(11):1747–1756.* [277]
Human umbilical vascular endothelial cells (HUVECs)	Dabigatran and rivaroxaban	Transcriptional changes in HUVECs when exposed to thrombin, focusing on the expression levels of preselected pro-inflammatory genes.	Endothelial Leukocyte Adhesion Molecule (ELAM)-1, VCAM-1, ICAM-1, MCP-1, IL-8, CXCL1, CXCL2 and TF	▪ **Rivaroxaban and Dabigatran**: Both drugs concentration-dependently suppressed the inflammatory gene expression after the stimulation of thrombin generation. This indicates that both these drugs have anti-inflammatory effects. ▪ **Dabigatran at Low Concentrations (3–300 nM)**: At these specific low concentrations, dabigatran significantly increased the expression levels of several inflammatory markers: CXCL1, CXCL2, IL-8, ELAM-1, MCP-1, and tissue factor. ▪ **Comparison**: Rivaroxaban downregulated the expression of pro-inflammatory markers and tissue factor to a similar extent as dabigatran, suggesting that both drugs have a comparable anti-inflammatory effect in the context of the study.	*Thromb Res. 2016 Jun;142:44–51* [278]
44 patients with acute ischemic stroke patients who were newly prescribed anti-thrombotic agents.	Dabigatran (n = 12) and apixaban (n = 14) with antiplatelet agents (n = 18) used as control.	Acute ischemic stroke	Hs-CRP, IL-6, pentraxin–3	▪ **Dabigatran** seemed to reduce hs-CRP (indicating reduced inflammation) and increase pentraxin-3 levels more prominently than apixaban. ▪ **Apixaban** had an alteration in inflammatory markers that was more similar to those observed in the group taking antiplatelet agents.	*Clin Transl Med. 2018 Jan 12;7(1):2.* [279]
Polymorphonuclear leukocytes (PMNLs) isolated from heparinized venous blood taken from non-smoking, healthy adults.	Dabigatran and rivaroxaban	Inflammatory response by NOACs	ROS, elastase release, cytosolic calcium flux, neutrophil extracellular trap (NET) formation, cell viability	No significant pro-inflammatory effects of dabigatran or rivaroxaban at concentrations of up to 10 µM was observed on the indicated PMNL markers.	*Activation. Pharmaceuticals (Basel). 2018 May 14;11(2):46.* [282]
187 patients with NVAF	Dabigatran (n = 96) and rivaroxaban (n = 91)	NVAF	Hs-CRP, pentraxin-3, IL-1β, IL-6, IL-18, TNF-α, MCP-1, GDF-15, and soluble thrombomodulin.	▪ Although the interval changes in soluble thrombomodulin levels tended to be greater in the dabigatran group compared to the rivaroxaban group, the difference was not statistically significant ([0.3 (0–0.7) vs. 0.5 (0–1.0) FU/mL, *p* = 0.061]). ▪ There were no significant differences in the interval changes in any of the other evaluated inflammatory markers between the two groups (rivaroxaban and dabigatran). ▪ The increased levels of soluble thrombomodulin after NOACs treatment might be related to bleeding events, as the interval changes in soluble thrombomodulin levels were significantly greater in patients with bleeding compared to those without.	*Heart Vessels. 2019 Jun;34(6):1002–1013* [283]
Male Wistar rats weighing between 250 and 350 g	Apixaban, dabigatran, and rivaroxaban	Ischemia-reperfusion injury in the kidneys	IL-1β and TNF-α.	▪ Apixaban, dabigatran, and rivaroxaban reduced aortic clamping-induced renal tissue oxidation and inflammation. ▪ Dabigatran and rivaroxaban additionally attenuated ischemia-reperfusion-related histological damage in kidneys	*Turk Gogus Kalp Damar Cerrahisi Derg. 2022 Apr 27;30(2):184–191.* [280]
Human umbilical vascular endothelial cells (HUVECs)	Dabigatran and rivaroxaban	Inflammatory activation in endothelial cells caused by 25-hydroxycholesterol (25-OHC), which is associated with atherosclerosis.	Anti-inflammatory cytokines: TGF-β IL-37 IL-35 (specifically its subunits EBI3 and p35) Pro-inflammatory cytokines: IL-18 IL-23	**25-OHC Effects:** ▪ Decreased the mRNA expression of anti-inflammatory cytokines TGF-β and IL-37. ▪ Increased the mRNA expression of pro-inflammatory cytokines/subunits EBI3, p35, IL-18, and IL-23 in comparison to an untreated control. **Effects of Rivaroxaban and Dabigatran Alone:** ▪ Significantly increased the mRNA expression of anti-inflammatory cytokines TGF-β and IL-37 compared to an untreated control. **Effects of Rivaroxaban and Dabigatran on 25-OHC Treated Cells:** ▪ Increased the mRNA expression of anti-inflammatory cytokines TGF-β and IL-37. ▪ Decreased the mRNA expression of pro-inflammatory cytokines/subunits EBI3, p35, IL-23, and IL-18 compared to cells treated with 25-OHC alone.	*Clin Exp Pharmacol Physiol. 2022 Aug;49(8):805–812.* [284]
**Animal model**: 60 C57B/6J mice divided into groups: control (CON) group, AF group, AF + edoxaban group, and AF + rivaroxaban group. **Human Model**: Erythrocytes of patients with atrial fibrillation	Edoxaban and rivaroxaban	Atrial fibrillation	TNF-α, IL-1β, IL-6, and IL-10.	Both edoxaban and rivaroxaban reduced inflammation in atrial fibrillation, where the effect of edoxaban was superior to that of rivaroxaban.	*Front Pharmacol. 2022;13:904317.* [285]
A syngeneic mouse model comprising male BALB/c mice that were inoculated with colon cancer Colon26 cells.	Dabigatran, edoxaban, and rivaroxaban	Effect of NOACs on tumor progression in colorectal cancer.	Appraisal of tissue-factor, plasminogen activator inhibitor–1 (PAI-1), IL-6 and MMP-2 in the plasma of Colon26-inoculated mice.	All NOACs displayed anti-inflammatory effect, edoxaban displayed a superior ability not just to suppress tumor growth but also to mitigate elevated levels of inflammatory markers.	*TH Open. 2023;7(1):e1-e13.* [281]

## 5. Novel Revelations of Apixaban in Osteoarthritis

As discussed previously, PAR2 is a key player in OA pathophysiology, significantly contributing to the characteristic inflammatory cascade of the disease. The intricate interplay between coagulation and immune–inflammatory systems, particularly FXa activating PAR2, underscores a crucial pathway contributing to OA inflammation. NOACs like apixaban, a selective FXa inhibitor, emerge as potential modulators of this pathway. This is further supported by accumulating evidence that suggests apixaban might possess anti-inflammatory properties (*refer to the above for details*).

Building upon the established role of PAR2 in OA inflammation and the critical role of FXa in PAR2 activation, our team has investigated the potential of apixaban to mitigate FXa-PAR2 mediated inflammation in OA. We specifically aimed to expand apixaban’s therapeutic utility beyond anticoagulation to potentially influence OA’s inflammatory processes. This could pave the way for novel OA treatment strategies through targeted modulation of key inflammatory pathways essential for disease progression.

The core of our research involved a meticulously constructed in vitro model to dissect the anti-inflammatory potential of apixaban in a controlled environment mimicking the inflammatory state of OA. The OA model was created via an FXa-induced inflammatory state in BMSC-derived human chondrocytes. Our model successfully establishes a solid foundation for understanding the induced inflammatory response, as evidenced by the increased expression of TNF-α within our in vitro OA system. The model’s precision is further validated by its ability to replicate essential inflammatory markers in chondrocytes, thereby providing robust insights into the inflammatory mechanisms at play.

Significantly, treatment with apixaban led to a notable decrease in PAR-2 levels, suggesting its ability to alter this receptor’s activity and potentially dampen the downstream inflammatory cascade. Further detailed analysis is required to investigate this effect, mapping out the intricate inflammatory pathways within chondrocytes and how they are influenced by FXa and PAR2 interaction. While we are not presenting new data, the hypothetical scheme depicted in Figure 5 illustrates the potential consequences of this downregulation.

The analysis suggests that apixaban’s inhibition of FXa curtails PAR-2 expression, potentially impacting the NF-κB signaling pathway, monocyte recruitment, and osteoclastogenesis. These factors are crucial contributors to ROS generation and chronic pain associated with OA. Additionally, the analysis visualizes the downstream modulation of inflammatory cytokines like TNF-α and IL-1β, and signaling proteins ERK1/2, which contribute to the deterioration of chondrocyte integrity. The involvement of ADAMTS5 and aggrecan in chondrocyte structure, alongside SOX4’s role in integrity loss, underscores the comprehensive nature of the inflammatory pathway. Collectively, these findings encapsulate the disease pathogenesis and the potential therapeutic modulation by apixaban, highlighting its anti-inflammatory capacity within the chondrocyte model.

In summary, our preliminary data allude to apixaban’s effectiveness in modulating the key inflammatory mediator PAR2. This sets a promising stage for further in-depth investigation of the pathway outlined. As we delve deeper into this research, we aim to elucidate apixaban’s pleiotropic effects, which could significantly contribute to advancements in the management of OA, a condition desperately in need of novel therapeutic approaches. Utilizing multi-omics approaches, such as genomics, transcriptomics, proteomics, and metabolomics, can provide a holistic understanding of the systemic effects of NOACs. This integrative approach can help identify novel biomarkers and therapeutic targets, as well as elucidate the complex interactions between NOACs and inflammatory pathways. For instance, a hypothetical study could analyze the transcriptomic and proteomic profiles of patients with rheumatoid arthritis before and after apixaban treatment, identifying changes in gene expression and protein levels associated with inflammation. Metabolomic analysis could further reveal shifts in metabolic pathways, providing insights into how NOACs influence inflammatory processes at a molecular level. By integrating these diverse datasets, researchers can map out comprehensive signaling networks and identify potential points of therapeutic intervention, ultimately leading to more targeted and effective treatments for inflammatory diseases.

## 6. Comparison of Anti-Inflammation Mediated by NOACs vs. Heparins and Fondaparinux

The pharmacological mechanisms of the anticoagulant effects of heparin are well known and described in numerous reviews of the literature. Recent evidence also associates the use of heparin with various anti-inflammatory properties [286]. Studies show that heparin mediates such anti-inflammatory effects through several mechanisms. Heparin binds to and inhibits the formation of complement factors, interfering with both the classical and alternative complement pathways [287]. Additionally, heparin can bind to pro-inflammatory cytokines and chemokines, preventing them from interacting with their specific receptors [288]. For instance, heparin has been shown to interact with IL-8, a potent chemokine involved in the recruitment of neutrophils to sites of inflammation. By binding to IL-8, heparin inhibits its interaction with the CXCR1 and CXCR2 receptors on the surface of neutrophils. This prevents IL-8 from exerting its chemotactic effects, thereby reducing neutrophil migration and subsequent inflammation.

Furthermore, there is evidence to suggest that heparin may impair neutrophil chemotaxis and leukocyte migration to the site of inflammation [289]. Heparin also inhibits leukocyte adhesion to the endothelium through interaction with P/L-selectin [290]. Another mechanism through which heparin and related compounds may exert their anti-inflammatory effects is through the modulation of NF-κB signaling [291]. Heparin inhibits the translocation of the nuclear inflammatory transcription factor NF-κB from the cytosol to the nucleus, potentially reducing the activation of inflammatory molecules and regulating the production of pro-inflammatory mediators and cytokines. Recent studies also indicate that LMWH binds with high affinity to IFNγ and effectively inhibits its interaction with its cellular receptor [292]. LMWH also influences the biological activity of IL-6 by binding to either IL-6 or the IL-6/IL-6Rα complex, thereby preventing the formation of the IL-6/IL-6Rα/gp130 signaling complex [292]. These findings elucidate yet another molecular mechanism underlying the anti-inflammatory action of LMWH and highlight its potential to favorably modulate conditions characterized by the overexpression of these two cytokines. These mechanisms collectively contribute to heparin’s anti-inflammatory effects, making it a potential therapeutic agent for conditions characterized by excessive inflammation.

Much like heparin, fondaparinux has exhibited pleiotropic effects and has been shown to mediate anti-inflammatory effects [293]. Fondaparinux is a pentasaccharide that works by selectively binding to antithrombin III (ATIII) and potentiating its ability to neutralize factor Xa, interrupting the coagulation cascade and inhibiting thrombin formation and thrombus development [294]. Studies have shown that fondaparinux administration leads to a reduction in various inflammatory cytokines, such as IL-8 and, to a lesser extent, in Rantes and TNFα [215]. Prolonged administration of fondaparinux in hospitalized COVID-19 patients has been associated with anti-inflammatory effects beyond its known antithrombotic action. In COVID-19, a severe inflammatory response often contributes to the pathogenesis and progression of the disease, leading to complications such as ARDS and multi-organ failure. Fondaparinux appears to modulate this inflammatory response through several mechanisms. It has been observed to bind to certain pro-inflammatory cytokines and chemokines, such as IL-6 and IL-8, thereby preventing these molecules from interacting with their receptors on immune cells. This interaction can reduce the recruitment and activation of neutrophils and other inflammatory cells, diminishing the overall inflammatory milieu. Additionally, fondaparinux may inhibit the release of extracellular traps (NETs) from neutrophils, which are web-like structures composed of DNA and proteins that can exacerbate inflammation and thrombosis. By reducing NET formation, fondaparinux not only helps to prevent thrombosis, but also alleviates associated inflammatory damage. Moreover, prolonged use of fondaparinux has been linked to decreased levels of inflammatory markers such as CRP and fibrinogen in COVID-19 patients. These effects suggest that fondaparinux can contribute to a reduction in the hyperinflammatory state often seen in severe cases of COVID-19, potentially improving patient outcomes by mitigating the risk of inflammatory complications. [293]. Additionally, fondaparinux can interfere with the complement cascade and formation of complement factors, which play a role in inflammation [295]. It can also bind to certain pro-inflammatory molecules, preventing them from interacting with their receptors, and may interfere with the adhesion of inflammatory cells to the endothelium, reducing their migration to sites of inflammation [296]. While the exact mechanisms are not fully elucidated, these actions collectively contribute to fondaparinux’s anti-inflammatory effects, making it a potential therapeutic agent for conditions characterized by excessive inflammation, such as COVID-19. Furthermore, given its mechanism of action in inhibiting FXa, it is likely that fondaparinux would have some anti-inflammatory effects through reduced PAR activation, similar to NOACs, serving as a primer for future studies.

## 7. Unexplored Aspects of NOACs in Inflammation

Despite the extensive research and numerous clinical trials surrounding NOACs, certain aspects of their roles in inflammation/anti-inflammation remain inadequately explored. This section aims to identify the existing gaps in our understanding and suggest methodologies and future directions to address these gaps effectively. The focus will be on the potential anti-inflammatory mechanisms of NOACs, their broader therapeutic applications, and the experimental models and techniques that could be utilized to further investigate these properties.

### 7.1. Identified Gaps in Current Research

#### 7.1.1. Limited Understanding of Molecular Mechanisms

While it is established that NOACs, such as rivaroxaban and apixaban, can modulate inflammation via PAR signaling pathways, the precise molecular mechanisms remain unclear. For instance, how these drugs interact with different PARs (PAR1, PAR2, etc.) and influence downstream signaling pathways like NF-κB and MAPK is not fully understood. The current studies have predominantly focused on the broad effects without delving into the specific intracellular events triggered by NOACs.

#### 7.1.2. Inconsistencies in Anti-Inflammatory Effects across Models

There are inconsistencies in the anti-inflammatory effects of NOACs across different experimental models and clinical studies. For example, while rivaroxaban has shown promise in reducing markers of inflammation in animal models of atherosclerosis and AAA (abdominal aortic aneurysm), similar effects are not always observed in human studies [297]. This discrepancy suggests a need for more standardized and comparable research protocols.

#### 7.1.3. Lack of Longitudinal Human Studies

Most of the existing research is based on short-term studies. Longitudinal studies examining the long-term effects of NOACs on inflammation and related conditions are scarce. Understanding how chronic administration of NOACs influences inflammation over extended periods is crucial for assessing their potential therapeutic roles in chronic inflammatory diseases.

#### 7.1.4. Population Diversity in Research

The majority of studies have focused on specific patient populations, such as those with atrial fibrillation or undergoing orthopedic surgery. There is a need to explore the anti-inflammatory effects of NOACs in a broader range of conditions, including autoimmune disorders, chronic kidney disease, and metabolic syndromes.

#### 7.1.5. Comparative Effectiveness of Different NOACs

While individual NOACs have been studied, comparative research on their relative effectiveness in reducing inflammation is limited. This is particularly important for understanding which NOAC might be more beneficial for specific inflammatory conditions.

### 7.2. Proposed Methodologies and Future Directions

#### 7.2.1. Advanced Molecular and Cellular Studies

To elucidate the molecular mechanisms of NOACs in inflammation, advanced molecular biology techniques such as CRISPR-Cas9 gene editing can be used to create knockout models for specific PARs. These models can help determine the exact role of each PAR in mediating the anti-inflammatory effects of NOACs. Additionally, utilizing techniques like RNA sequencing and proteomics can provide a comprehensive view of the changes in gene and protein expression in response to NOAC treatment. In fact, currently, we are analyzing the transcriptomic data obtained following apixaban treatment in our pro-inflammatory chondrocyte model. This ongoing research aims to identify specific gene expression changes induced by apixaban, providing insights into its molecular mechanisms of action in inflammation. Preliminary results suggest modulation of key inflammatory pathways ([298] and further analysis will help elucidate the broader impact of apixaban on chondrocyte function and viability.

#### 7.2.2. Standardized Experimental Models

Developing standardized experimental models across different laboratories can help address the inconsistencies observed in the effects of NOACs. For example, using uniform protocols for inducing inflammation and measuring outcomes in animal models and in vitro systems can provide more reliable and comparable data.

#### 7.2.3. Longitudinal Clinical Studies

Conducting longitudinal studies in diverse human populations can provide valuable insights into the long-term anti-inflammatory effects of NOACs. These studies should include regular monitoring of inflammatory biomarkers, clinical outcomes, and potential side effects over extended periods. This approach can help identify the sustained benefits and risks associated with chronic NOAC therapy.

#### 7.2.4. Population-Specific Research

Expanding research to include diverse populations, such as those with chronic inflammatory diseases, autoimmune disorders, and metabolic syndromes, can help determine the broader applicability of NOACs. Stratified analysis based on demographic factors (age, gender, ethnicity) and comorbid conditions can provide a more nuanced understanding of how different populations respond to NOAC therapy.

#### 7.2.5. Comparative Effectiveness of Research

Conducting head-to-head trials comparing the anti-inflammatory effects of different NOACs can help identify the most effective agents for specific conditions. These studies should include a comprehensive assessment of both efficacy and safety, considering various inflammatory markers and clinical endpoints.

#### 7.2.6. Integration of Multi-Omics Approaches

Utilizing multi-omics approaches, such as genomics, transcriptomics, proteomics, and metabolomics, can provide a holistic understanding of the systemic effects of NOACs. This integrative approach can help identify novel biomarkers and therapeutic targets, as well as elucidate the complex interactions between NOACs and inflammatory pathways. For instance, a hypothetical study could analyze the transcriptomic and proteomic profiles of patients with rheumatoid arthritis before and after apixaban treatment, identifying changes in gene expression and protein levels associated with inflammation. Metabolomic analysis could further reveal shifts in metabolic pathways, providing insights into how NOACs influence inflammatory processes at a molecular level. By integrating these diverse datasets, researchers can map out comprehensive signaling networks and identify potential points of therapeutic intervention, ultimately leading to more targeted and effective treatments for inflammatory diseases.

#### 7.2.7. Innovative Experimental Models

Leveraging advanced experimental models, such as 3D organoid cultures and patient-derived organoids, can offer more physiologically relevant platforms for studying the anti-inflammatory effects of NOACs. For example, 3D neurovascular unit organoids can be used to investigate the impact of NOACs on blood–brain barrier function and neuroinflammation in conditions like stroke and neurodegenerative diseases.

#### 7.2.8. Designing Better Thrombin and Factor Xa Using AI

The advancements in artificial intelligence (AI) have revolutionized protein engineering, enabling the design of optimized thrombin and FXa molecules that can be used to develop peptides targeting PARs and conjugated with NOACs for enhanced anti-inflammatory properties.

Utilizing AI-based approaches such as generative adversarial networks (GANs), transformer models, and specifically, tools like AlphaFold [299], ESM2 [300], and ProteinMPNN [301], researchers can predict and design thrombin and FXa inhibitors that mimic natural proteins but with optimized functionality for laboratory use, a strategy as predicted by Johnson et ِal. [302].

AlphaFold plays a crucial role in accurately predicting protein structures, which helps in understanding the potential interactions of these inhibitors with their targets. ESM2 leverages extensive protein sequence datasets to generate potential thrombin and FXa inhibitors optimized for stability and function, while ProteinMPNN refines these sequences to ensure they are structurally feasible and functionally active in laboratory conditions. Once optimized thrombin and FXa molecules are designed, these sequences can be used to derive peptides specifically targeting PARs. This involves optimizing peptide sequences for high binding affinity and specificity to PARs, validating their binding conformations using structural prediction tools like AlphaFold, and conducting in vitro assays to test their binding efficacy.

Subsequently, these AI-derived peptides can be conjugated to NOACs, using a strategy similar to that of Martin et al. [303] to combine their anticoagulant properties with targeted PAR inhibition, potentially enhancing therapeutic efficacy and reducing side effects. The resulting NOAC–peptide conjugates will then have to undergo rigorous pharmacokinetic and pharmacodynamic testing. Finally, the anti-inflammatory activity of these conjugates can be evaluated through comparative studies against NOACs alone, measuring the reduction in pro-inflammatory markers such as TNF-α and MCP-1 using techniques like ELISA, RT-PCR, and Western blotting. This approach aims to demonstrate that the conjugates offer superior anti-inflammatory benefits compared to NOACs alone, showcasing the power of AI in protein engineering and promising more effective and targeted anticoagulant therapies with added anti-inflammatory properties.

#### 7.2.9. Designing NOAC-Antibody Conjugates for Enhanced Anti-Inflammatory Activity

The strategic development of NOAC–antibody conjugates, inspired by the successful design of antibody–peptide conjugates (APICs) for targeting cysteine proteases [304], promises to enhance the anti-inflammatory properties of NOACs. This approach integrates advanced AI-driven design of optimized thrombin and FXa inhibitors and their conjugation with antibodies to ensure targeted delivery and enhanced therapeutic efficacy. Utilizing AI-based techniques such as GANs, transformer models, AlphaFold, ESM2, and ProteinMPNN, we can predict and design thrombin and FXa inhibitors that are both structurally feasible and functionally active. These inhibitors serve as the foundation for deriving peptides that specifically target PARs.

The process begins with the rational design of thrombin and FXa inhibitors using AI to predict sequences that exhibit high binding affinity and specificity to PARs. AlphaFold’s structural predictions and ESM2’s sequence optimization play crucial roles in refining these peptides to ensure their functionality in biological systems. Once the optimized sequences are validated, they are chemically modified to enhance their binding capabilities to the PARs. Following peptide optimization, the next step involves conjugating these peptides to antibodies. This conjugation is designed to leverage the natural targeting and internalization properties of antibodies, thereby ensuring that the NOAC–peptide conjugates accumulate in specific cell types where PAR is expressed. By targeting these specific cells, the conjugates can deliver the NOACs more precisely, reducing systemic side effects and increasing local therapeutic efficacy.

The final evaluation phase involves rigorous testing of the NOAC–antibody conjugates for their anti-inflammatory activity compared to NOACs alone. This includes in vitro and in vivo assays to measure the reduction in pro-inflammatory markers such as TNF-α and MCP-1. Techniques like ELISA, RT-PCR, and Western blotting will quantify these effects, providing comparative data to demonstrate the enhanced efficacy of the conjugates. The targeted delivery mechanism is expected to exhibit superior anti-inflammatory benefits by ensuring higher concentrations of the therapeutic agent at the site of inflammation, thus maximizing efficacy and minimizing off-target effects.

This innovative strategy of designing NOAC–antibody conjugates using AI-driven approaches and targeted delivery systems exemplifies the potential of integrating computational tools with advanced bioconjugation techniques. It holds promise for developing more effective and targeted anti-inflammatory therapies, offering significant improvements over traditional NOAC treatments.

#### 7.2.10. Design of Aptameric NOAC Conjugates to Attenuate PAR-Mediated Inflammation

Aptamers are single-stranded oligonucleotides that fold into specific three-dimensional shapes, allowing them to bind with high affinity and specificity to various target molecules, including proteins, small molecules, and even cells [305]. The unique properties of aptamers make them suitable for therapeutic applications, including the development of aptameric NOAC conjugates designed to attenuate PAR-mediated inflammation.

To design aptameric NOAC conjugates targeting FXa/thrombin and PAR interactions, we can employ a strategy inspired by the creation of EXACT inhibitors, which combine exosite-binding aptamers with active site inhibitors to enhance potency and selectivity [306]. The process involves several key steps:

Selection and Optimization of Aptamers: The first step is to identify and optimize aptamers that specifically bind to FXa/thrombin and PAR. This can be achieved using a selection process such as SELEX (Systematic Evolution of Ligands by EXponential enrichment), where a large library of random oligonucleotide sequences is screened for binding to the target proteins [306]. Selected aptamers are then optimized for binding affinity and specificity through iterative rounds of selection and amplification. [Once optimized aptamers are obtained, they are conjugated to NOACs. This involves chemically linking the aptamer to the active site inhibitor, thereby ensuring that the conjugate retains both the aptamer’s binding affinity and the inhibitory activity of the NOAC. The design must consider the spatial arrangement and flexibility of the linker to allow for effective binding to FXa, thrombin, and PAR simultaneously. The linker length and composition are critical parameters that need to be optimized to prevent steric hindrance and ensure synergistic binding [306].

The aptamer–NOAC conjugates are then to be tested for their ability to bind to FXa or thrombin and PAR and inhibit their activity. This involves in vitro assays such as surface plasmon resonance (SPR) and ELISA to measure binding affinity and specificity. Additionally, functional assays such as coagulation assays and cell-based assays are used to evaluate the inhibitory effects on FXa or thrombin and the attenuation of PAR.

## 8. Addressing Specific Research Gaps

### 8.1. Molecular Mechanism of NOACs

Investigating the interaction of NOACs with PARs at a molecular level using techniques like co-immunoprecipitation and fluorescence resonance energy transfer (FRET) can help map out the specific signaling pathways involved. Additionally, exploring the role of NOACs in modulating epigenetic changes associated with inflammation can provide new insights into their long-term effects.

### 8.2. Comparative Studies in Diverse Models

Conducting parallel studies in both animal models and human cell lines can help bridge the gap between preclinical and clinical findings. Using transgenic animal models that mimic human inflammatory conditions can provide more relevant data on the efficacy of NOACs.

### 8.3. Long-Term Impacts of NOACs

Implementing longitudinal cohort studies with regular follow-up intervals can help assess the cumulative impact of NOACs on inflammation. These studies should include comprehensive health assessments, biomarker analysis, and quality of life evaluations to capture the full spectrum of effects.

### 8.4. Broadening Research Populations

Including under-represented populations in clinical trials and observational studies can ensure that the findings are generalizable. Collaborating with international research centers can facilitate the inclusion of diverse cohorts, enhancing the robustness of the data.

### 8.5. Integration of Multi-Omics Data

Creating integrated databases that combine multi-omics data with clinical information can help identify patterns and correlations that are not apparent from single-layer analyses. Machine learning algorithms can be employed to analyze these complex datasets and generate predictive models for the anti-inflammatory effects of NOACs.

### 8.6. Effects of NOAC Antidotes on Anti-Inflammatory Properties

The use of NOAC antidotes, such as idarucizumab for dabigatran and andexanet alfa for factor Xa inhibitors, may affect their ability to reduce inflammation. These remedies efficiently counteract the blood thinning effects of NOACs, reinstating the production of thrombin and diminishing factor Xa activity. However, the use of antidotes to neutralize NOACs may potentially counteract the possible anti-inflammatory advantages linked to the blockage of PAR signaling pathways caused by NOACs. This has the potential to reduce the overall effectiveness of NOACs in treating inflammatory diseases. Further investigation should prioritize comprehending the enduring consequences of NOAC antidotes for inflammation and their overall influence on patient outcomes. Studies should also examine if reversing anticoagulation also results in a recurrence of inflammatory processes and how this may impact the treatment of disorders that involve both blood clotting and inflammation. In addition, investigating various approaches to achieve a balance between reversing anticoagulation and preserving anti-inflammatory benefits might offer useful insights for optimizing therapy procedures.

## 9. Materials and Methods

The methodology for this review on non-vitamin K oral anticoagulants (NOACs) and their role in inflammation and protease-activated receptor (PAR) signaling involved a systematic literature search in PubMed, using terms such as “Non-Vitamin K Oral Anticoagulants”, “NOACs”, “Direct Oral Anticoagulants”, “inflammation”, “Protease-Activated Receptor”, “PAR signaling”, “rivaroxaban”, “apixaban”, “edoxaban”, and “dabigatran”, limited to articles published from January 1996 to May 2024 in English and peer-reviewed journals. Two independent reviewers (SJ and YB) screened titles and abstracts for relevance, with full texts assessed based on inclusion criteria: studies focusing on NOACs and their effects on inflammation and PAR signaling. Exclusion criteria included non-relevant studies, reviews, and editorials.

Data extraction, performed by two reviewers (SJ and YB), captured study design, sample size, NOACs investigated, outcomes, and key findings. Data synthesis involved narrative synthesis, with grouping by NOAC type and investigated outcomes. Discrepancies between reviewers were resolved through discussion. Ethical considerations were minimal as the review used publicly available data. Limitations included potential language bias and heterogeneity of the studies, which might affect generalizability.

## 10. Conclusions

The potential anti-inflammatory effects of NOACs represent a promising yet underexplored area of research. By addressing the identified gaps through advanced methodologies and comprehensive studies, we can unlock new therapeutic applications for these drugs. The integration of molecular, cellular, and clinical approaches, along with a focus on diverse populations and long-term effects, will provide a deeper understanding of how NOACs can modulate inflammation. This knowledge will pave the way for innovative treatments for chronic inflammatory diseases, enhancing patient outcomes and expanding the therapeutic utility of NOACs beyond anticoagulation.

## Figures and Tables

**Figure 1 ijms-25-08727-f001:**
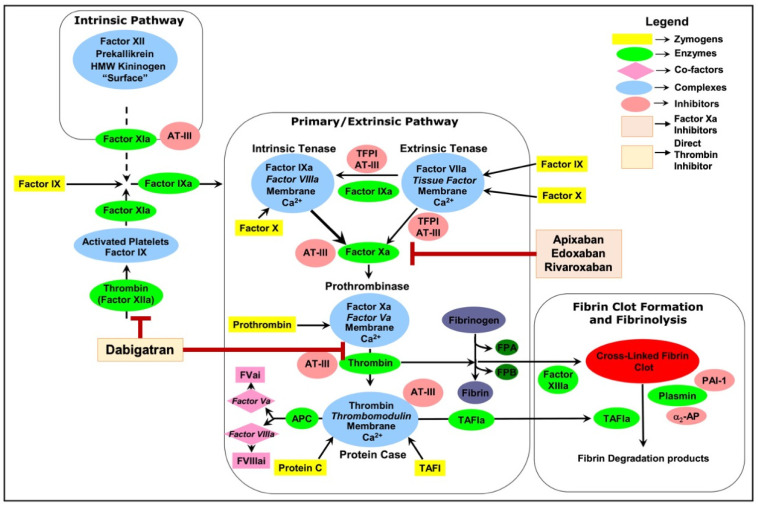
The current model of the blood coagulation cascade, depicting NOACs’ mechanism of action. There are two pathways, the intrinsic pathway and the extrinsic pathway. These multicomponent processes are illustrated as enzymes, inhibitors, zymogens, or complexes. On injury to the vessel wall, tissue factor, the cofactor for the extrinsic tenase complex, is exposed to circulating FVIIa and forms the extrinsic tenase. FIX and FX are converted to their serine proteases FIXa and FXa, which then form the intrinsic tenase and the prothrombinase complexes, respectively. The combined actions of the intrinsic and extrinsic tenase and the prothrombinase complexes lead to an explosive burst of the enzyme thrombin (IIa). In addition to its multiple procoagulant roles, thrombin also acts in an anticoagulant capacity when combined with the cofactor thrombomodulin in the protein Case complex. The product of the protein Case reaction, activated protein C (APC), inactivates the cofactors FVa and FVIIIa. The cleaved species, FVai and FVIIIai, no longer support the respective procoagulant activities. Once thrombin is generated through procoagulant mechanisms, thrombin cleaves fibrinogen (releasing fibrinopeptide A and B [FPA and FPB]), as well as activating FXIII to form a cross-linked fibrin clot. Thrombin–thrombomodulin also activates thrombin activate-able fibrinolysis inhibitor, which slows fibrin degradation by plasmin. The procoagulant response is downregulated by the stoichiometric inhibitor tissue factor pathway inhibitor (TFPI) and antithrombin III (AT-III). TFPI serves to attenuate the activity of the extrinsic tenase trigger of coagulation. AT-III directly inhibits thrombin, FIXa, and FXa. The accessory pathway provides an alternate route for the generation of FIXa. Thrombin has also been shown to activate FXI. The fibrin clot is eventually degraded by plasmin, yielding soluble fibrin peptides. Factor Xa inhibitors (apixaban, edoxaban, and rivaroxaban) act by binding to the active site of factor Xa, inhibiting the conversion of prothrombin to thrombin, the final enzyme in the coagulation cascade. Dabigatran, conversely, functions as a direct thrombin inhibitor. It binds with high affinity to the active site of thrombin, inhibiting its ability to convert fibrinogen to fibrin, thereby preventing clot formation.

**Figure 2 ijms-25-08727-f002:**
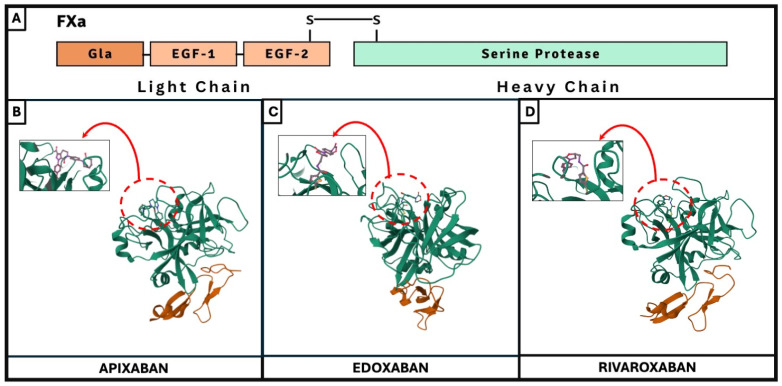
Schematic representation of structural domains of FXa and binding sites of FXa inhibitors. A schematic representation of FXa protein structural domains and the location of the binding sites of FXa inhibitors (apixaban, edoxaban, and rivaroxaban) are depicted. (**A**) illustrates the domain organization of FXa, highlighting the serine protease domain in the heavy chain (indicated by green) where the binding site is located. (**B**–**D**) depict the structures of apixaban, edoxaban, and rivaroxaban, respectively, as obtained from the Protein Data Bank, indicating their binding sites with FXa, which exhibit enzyme kinetics similar to competitive inhibitors.

**Figure 3 ijms-25-08727-f003:**
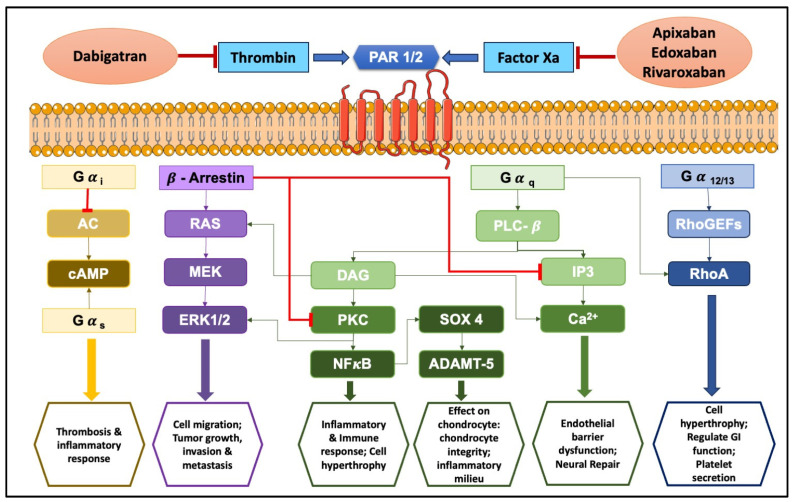
Schematic representation of PAR1- and PAR2-mediated signal transduction. PAR1 and PAR2 are G protein-coupled receptors that can be activated by thrombin and FXa, initiating a cascade of cellular responses. Upon cleavage, PAR1/2 interact with different G proteins like Gα_i_, Gα_12/13_, Gα_s_, and Gα_q_. Gα_12/13_ leads to Ras homolog family member A (RhoA) activation, via Rho guanine nucleotide exchange factors (RhoGEFs) influencing cell hypertrophy. Gα_q_ activates phospholipase C-β, generating second messengers that trigger calcium release and Protein Kinase C (PKC) activation. PKC can further activate the nuclear factor kappa B (NF-κB) signaling pathway to upregulate production of SRY-box transcription factor 4 (SOX4) and A disintegrin and metalloproteinase with thrombospondin motifs 5 (ADAMTS5). Gα_i_ can inhibit adenylate cyclase (AC) to regulate downstream cAMP, whereas Gα_s_ can increase cAMP. β-arrestin can activate the ERK1/2 signaling pathway but exhibits inhibitory effects on PKC and calcium release. Anticoagulants like dabigatran (thrombin inhibitor) and apixaban, edoxaban, and rivaroxaban (FXa inhibitors) can potentially disrupt this signaling by preventing PAR activation. Ultimately, these signal transduction pathways can trigger physiological changes like inflammatory and immune responses, cell hypertrophy, and cell migration.

**Figure 4 ijms-25-08727-f004:**
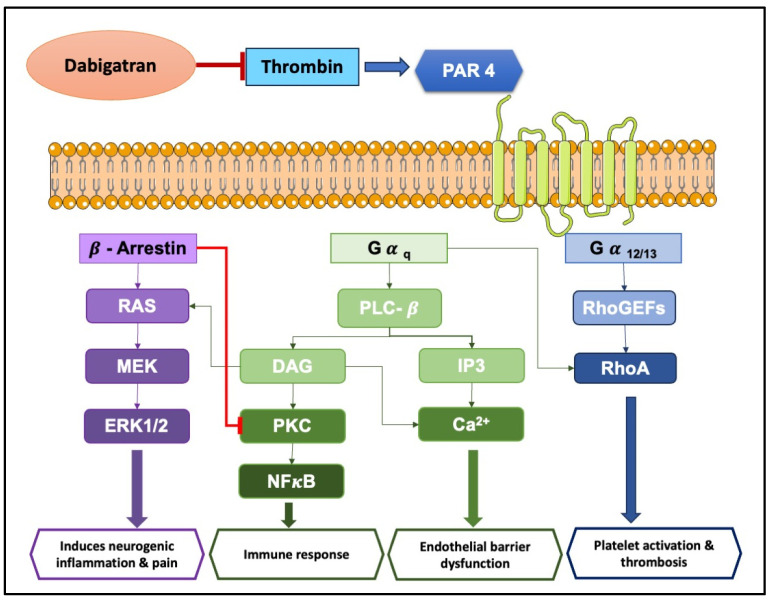
Schematic representation of PAR4-mediated signal transduction. PAR4 can activate signaling pathways involving Gα_12/13_ and Gα_q_. Gα_12/13_ prompts RhoGEFs to activate RhoA, while Gα_q_ -Phospholipase C- β (PLC- β) leads to downstream effects such as upregulation of inositol triphosphate (IP3) and diacylglycerol (DAG), resulting in calcium alterations and PKC upregulation, which ultimately leads to activation of the NF-κB signaling pathway. Additionally, β-arrestin can facilitate ERK1/2 phosphorylation but has inhibitory effects on PKC. Thrombin, known for its ability to cleave PAR4, can influence these pathways, therefore dabigatran (thrombin inhibitor) can modulate signal transduction by attenuating thrombin’s effects. As seen, PAR4 activation can cause physiological alterations such as inflammatory and immune response, endothelial barrier dysfunction, and platelet activation.

**Figure 5 ijms-25-08727-f005:**
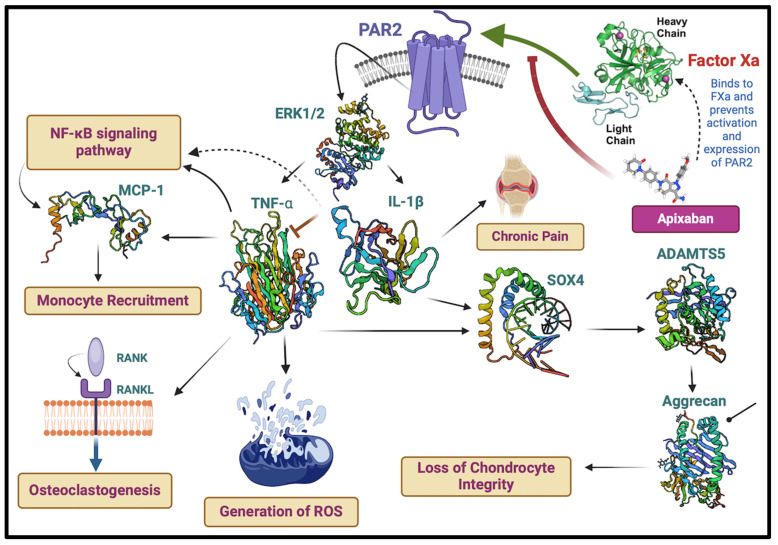
Proposed mechanism of apixaban’s modulatory effects on factor Xa and associated inflammatory signaling pathways in an osteoarthritic chondrocyte model. This illustration delineates the pathways through which apixaban may exert anti-inflammation in the context of osteoarthritis. Apixaban targets FXa, inhibiting its ability to bind and activate PAR2, which is represented by the red inhibitory line. This intervention most likely attenuates the downstream signaling cascades involved in OA pathophysiology: 1. PAR2 inhibition: The blockage of PAR2 activation by apixaban may ameliorate the downstream signaling events mediated by ERK1/2 that lead to the production of pro-inflammatory cytokines, such as TNF-α and IL-1β, potentially alleviating chronic pain associated with OA. 2. Cytokine modulation: The expected reduction in TNF-α and IL-1β due to apixaban’s action on FXa mitigates the upregulation of molecules like MCP-1, which are involved in monocyte recruitment and the NF-κB signaling pathway, both key contributors to inflammation and osteoclastogenesis. 3. Protein expression: The illustration also indicates the potential effects of apixaban on the expression of regulatory proteins, including SOX4 and ADAMTS5, and their impact on critical components like aggrecan, which is essential for cartilage integrity. 4. Chondrocyte integrity and bone health: By modulating these inflammatory and catabolic pathways, apixaban may help preserve chondrocyte integrity, mitigate the generation of reactive oxygen species (ROS), and contribute to maintaining joint health by potentially impacting the RANK/RANKL pathway, which is crucial for osteoclast activity and bone resorption.

**Table 1 ijms-25-08727-t001:** Pharmacokinetic and clinical profiles of NOACs.

Anticoagulant	Apixaban	Edoxaban	Rivaroxaban	Dabigatran
Mechanism	Direct FXa Inhibitor	Direct FXa Inhibitor	Direct FXa Inhibitor	Direct Thrombin Inhibitor
Prodrug/absorption	No/3–4 h	No/Rapid	No/Rapid	Yes/Rapid
Bioavailability/half-life	50%/12 h	62%/9–11 h	66% w/o food up to 100% with food/5–9 h (young) 11–13 h (elderly)	6%/12–17 h
Vd	21 L	107 L	50 L	50–70 L
Time to reach max. Plasma conc./protein binding	1–4 h/87%	1–2 h/55%	2–4 h/92–95%	0.5–2 h/35%
Liver metabolism	Yes	Minimal	Yes	No
Renal excretion	25%	50%	35%	80%
Effect of diet	No effect on exposure	No effect on exposure	Peak levels reached at 3 h on fasting and 4 h with food.	Delayed absorption
Effect of age	Exposure is 32% greater in patients above 65 years	Exposure is 32% greater in patients above 65 years	Bioavailability greater in elderly with no difference in concentration	2 times higher bioavailability in elders
Effect of body weight	Weight < 50 kg have 20–30% increased exposure and weight > 120 kg has 20–30% reduced exposure	Weight < 50 kg have 20–30% increased exposure and weight > 120 kg has 20–30% reduced exposure	Weight < 50 kg have 20–30% increased exposure and weight > 120 kg has 20–30% reduced exposure	None
Effect of renal impairment	Peak concentration unaffected. However, there is a rise in exposure by 16%, 29%, and 44% corresponding to creatinine clearances of 51–80, 30–50, and 15–29 mL/min, respectively.	Peak concentration unaffected. However, there is a rise in exposure by 16%, 29%, and 44% corresponding to creatinine clearances of 51–80, 30–50, and 15–29 mL/min, respectively.	Rise in exposure with moderate or severe renal impairment	6 times higher exposure with severe renal impairment, half-life extended to 28 h
Effect of hepatic impairment	No change in exposure with Child–Pugh classification A/B	No change in exposure with Child–Pugh classification A/B	Increased on exposure with Child–Pugh classification B	No change in exposure with Child–Pugh classification B
Doses	2.5 mg, 5 mg	15 mg, 30 mg, 60 mg	2.5 mg, 10 mg, 15 mg, 20 mg	75 mg, 110 mg, 150 mg
Dosing/dosing form	Two times a day/Tablet	One time a day/Tablet	One time a day/Tablet	Two time a day/Capsule
ADR	>10% hematologic and oncologic hemorrhage; <10% hematuria, epistaxis; <1% hyper-sensitivity reactions	>10% hematologic and oncologic hemorrhage; <10% skin rash, anemia; <1% intra cranial hemorrhage, interstitial pulmonary disease	>10% hematologic and oncologic hemorrhage; <10% pruritus, abdominal pain; <1% angioedema cholestasis	>10% gastro-intestinal symptoms; <10% gastritis, esophagitis; <1% allergic oedema, thrombocytopenia
Pre- and post-operative care for minor surgical procedures	Suspend treatment for 2 days before the surgical procedure (meaning, skip 1 dose), and recommence 24 h after the surgery	Suspend treatment for 2 days before the surgical procedure (meaning, skip 1 dose), and recommence 24 h after the surgery	Suspend treatment for 2 days before the surgical procedure (meaning, skip 1 dose), and recommence 24 h after the surgery	Creatine clearance greater than 50 mL/min suspend treatment for 2 days before the surgical procedure (meaning, skip 1 dose), and recommence 24 h after the surgery
Pre- and post-operative care for major surgical procedures	Suspend treatment for 3 days before the surgical procedure (meaning, skip 4 doses), and recommence 48 h after the surgery	Suspend treatment for 3 days before the surgical procedure (meaning, skip 4 doses), and recommence 48 h after the surgery	Suspend treatment for 3 days before the surgical procedure (meaning, skip 2 doses), and recommence 48 h after the surgery	Creatine clearance greater than 50 mL/min suspend treatment for 3 days before the surgical procedure (meaning, skip 4 doses), and recommence 48 h after the surgery
Laboratory monitoring (optimal method)	Anti-FXa assay	Anti-FXa assay	Anti-FXa assay	Ecarin clotting time; dilute thrombin time
Laboratory monitoring (emergency)	Dilute prothrombin time	Limited substantial data	Prothrombin time (preferably with specific calibrated reagents)	Activated partial thromboplastin time (preferably with specific calibrated reagents)

**Table 2 ijms-25-08727-t002:** Baseline characteristics of the populations studied in atrial fibrillation NOAC trials, adopted from Ruff et al [25].

	ARISTOTLE		ENGAGE AF-TIMI 48			ROCKET-AF		RE-LY		
	Apixaban	Warfarin	Edoxaban 60 mg	Edoxaban 30 mg	Warfarin	Rivaroxaban	Warfarin	Dabigatran 150 mg	Dabigatran 110 mg	Warfarin
Population	9120	9081	7035	7034	7036	7131	7133	6076	6015	6022
Age (years) >75 years	70 31%	70 31%	72 41%	72 40%	72 40%	73 43%	73 43%	71.5 40%	71.4 38%	71.6 39%
Women	36%	35%	39%	39%	38%	40%	40%	37%	38%	39%
Persistent AF	85%	84%	75%	74%	75%	81%	81%	67%	68%	66%
Previous stroke/TIA	19%	18%	28%	29%	28%	55%	55%	20%	20%	20%
Previous VKA use	57%	57%	59%	59%	59%	62%	63%	50%	50%	49%
Previous aspirin use	31%	31%	29%	29%	30%	36%	37%	39%	40%	41%
Median follow-up (years)	1.8	1.8	2.8	2.8	2.8	1.9	1.9	2.0	2.0	2.0

AF = Atrial fibrillation. TIA = transient ischemic attack. VKA = vitamin K antagonist.

**Table 3 ijms-25-08727-t003:** The molecular characteristics of different PAR members.

Receptor	Amino Acids	Tethered Ligands	Cleavage Type	Classical Proteases	Cleavage Site	Activating Synthetic Ligands
PAR1	425	h: SFFLR m:SFLLR	Canonical	Thrombin		
				Factor Xa	^38^LDPR*SFLL^45^	SFLLRN-NH_2_
				Plasmin		
				MMP1	^36^ATLD*PRSF^43^	PRSFLLRN-NH_2_
				MMP13	^39^DPRS*FLLR^46^	
			Non-canonical	Elastase	^42^SFLL*RNPN^49^	RNPNDKYEPF-NH_2_
				Proteinase-3	^33^ATNA*TLDp^40^	TLDPRSF-NH_2_
				Activated Protein C	^43^FLLR*NPND^50^	NPNDKYEPF-NH_2_
PAR2	395	h: SLIGKV m: SLIGRL	Canonical	Trypsin		Isox-Cha-Chg-AR-NH_2_
				Mast cell Tryptase	^33^SKGR*SLIG^40^	SLIGKV-NH_2_
				Factor Xa		
				Thrombin		SLIGRL-NH_2_
				Elastase	^64^FSAS*VLTG^71^	
			Non-Canonical	Proteinase-3	^57^VFSV*DEFS^64^	
				Cathepsin G	^61^VDEF*SASV^68^	
				Cathepsin S	^53^VTVE*TVFS^60^	TVFSVDEFSA-NH_2_
PAR3	483	h: TFRGAP m: SFNGGP	Canonical	Thrombin	^35^LPIK*TFRG^42^	TFRGAP-NH_2_
			Non-Canonical	Activated protein C	^38^KTFR*GAPp^45^	SFNGGP-NH_2_
PAR4	385	h: GYPGQV m: GYPGKF	Canonical	Thrombin		
				Trypsin	^44^PAPRIGYPG^51^	GYPGQV-NH_2_
				Cathepsin G		

**Table 4 ijms-25-08727-t004:** The pro-inflammatory and anti-inflammatory roles of PAR members in different inflammatory conditions.

Inflammatory Condition	Anti-Inflammatory Properties	Pro-Inflammatory Properties
Neuroinflammation	Activation of PAR1 by low-concentration thrombin enhances neurite growth and branching, neuron viability.	Activation of PAR1 by granzyme B through phospholipase-Cb and the PIP2-IP3 pathway mediates neurodegradation. Glia maturation factor and mast cell proteases increased PAR2 expression in neurons and astrocytes. PAR2 activation mediates neuroinflammation through ERK1/2, JNK, MAPK.
Pruritus and pain	PAR1 agonists attenuated nociception and reduced inflammatory hyperalgesia.	PAR2 induced TRPV1 sensitization through activation of PKCɛ and PLC-dependent pathway. PAR2 also activated TRPV4 in PAR2-TRPV4 coupling, contributing to sustained hyperexcitability of nociceptor signaling. PAR2 activation of NK-1 receptors and release of prostaglandins mediating chronic hyperalgesia.
Inflammatory bowel disease (IBS)	PAR1 limited Th2 cytokine production, in oxazolone- induced colitis. PAR2 promoted autophagy and inhibition of apoptosis	PAR1 activated Th17 response and increased pro inflammatory cytokines in bacterial-induced colitis. PAR2 elevated wall thickness, granulocyte infiltration, and bacterial translocation. Additionally, it activated the PAR2/Akt/mammalian target of rapamycin pathway.
Asthma	Low thrombin via PAR1 activation observed to attenuate inflammation in allergic bronchial asthma	PAR1 increased the release of TGF-b1 and promoted the Th2 response PAR2 inhibition linked with reduction in allergen-induced airway hyperresponsiveness PAR1 and PAR2 agonists induced bronchoconstriction by initiating Ca^2+^ signaling in airway smooth muscle cells. Prolonged activation of PAR1 and PAR2 linked to pulmonary fibrosis in mouse models
Pneumonia	PAR4 enhances systemic cytokine response during late-stage pneumococcal infection	PAR1 reduced production of early inflammatory cytokines and chemokines, facilitated neutrophil recruitment, and induced alveolar leak PAR2 activation facilitates Streptococcus pneumoniae colonization and systemic dissemination PAR2 deletion attenuates TRPV4-dependent Ca^2+^ signaling, which primarily promotes persistent lung inflammation
Myocarditis	PAR1 limits viral load by activating NK cells. PAR1 and PAR4 elevated the levels of TLR3-dependent CXCL10 expression while reducing TLR3-dependent NFκB-mediated proinflammatory gene expression in viral myocarditis.	PAR2 reduced the antiviral TLR3-dependent IFNb production, and activated MAPK pathways, leading to severe myocarditis.
Cystitis		PAR1 and PAR4 activation mediates release of MIF through activation of CXCR4 receptors. PAR4 activation induces HGMB1 release through a MIF-mediated mechanism, leading to bladder pain. PAR1 upregulation in cystitis. PAR2 activation increases COX-2 by the ERK1/2 MAP kinase pathway expression, influencing bladder pain in the bladder urothelium.
Nephritis		PAR1-FXa signaling fosters the generation and growth of fibroblast pro-collagen. FXa induces fibrosis and pro-inflammatory reactions in fibroblasts via PAR2 activation. PAR2 significantly contributes to the development of glomerular crescents by facilitating macrophage infiltration and glomerular thrombosis.
Arthritis	uPA-activated PAR1 attenuates inflammatory osteoclastogenesis thrombin–PAR1 promotes the secretion of MMP-2 and inhibits the invasion of fibroblasts, limiting tissue damage. thrombin–PAR1 activation reduces the production of IL-17 and TNF-α by synovial fibroblasts.	PAR2 inhibition reduced spontaneous release of TNF-α and IL-1 β from synovium. PAR2-mediated neuronal sensitization and leukocyte trafficking via TRPV1 and NK1 receptors. PAR-2 activation by mast cell tryptase stimulates the proliferation of SFCs and the release of pro-inflammatory cytokines. PAR1 activation promotes macrophage aggregation via CCL2. PAR2 deletion associated with reduced inflammatory response during acute arthritis. thrombin–PAR1 signaling induces RANTES, mediating migration of inflammatory cells. PAR2 downregulation reduced TNF-α, IL-6, IL-8, and IFN-γ.

**Table 7 ijms-25-08727-t007:** Anti-inflammatory profile of rivaroxaban.

Human Model or Population	Mono/Combination Therapy	Condition Investigated	Inflammatory Markers Examined	Effect Observed	Reference
Japanese subjects with non-valvular chronic AF undergoing anti-coagulation therapy, analyzed using unbiased liquid chromatography/tandem mass spectroscopy and candidate multiplexed protein immunoassays.	Monotherapy either with rivaroxaban or warfarin.	Modulation of biologically-relevant plasma proteins in AF	▪ Nine metaproteins including fibulin-1, vitronectin, hemoglobin α, apolipoproteins C-II and H, complement C5 precursor, coagulation factor XIIIA and XIIIB subunits. ▪ 10 candidate proteins including thrombomodulin, ICAM -3, IL-8, and MMP-3.	Compared with warfarin, rivaroxaban was associated with a greater increase in thrombomodulin and a trend towards a reduction in MMP-9 over 24 weeks.	*Thromb Haemost. 2012;108(6):1180–1191.* [203]
Ex vivo samples of abdominal aortic aneurysms (AAA) with intraluminal thrombus from six patients. These samples were treated with and without rivaroxaban to assess its effects on inflammation and oxidative stress. Abdominal aortic samples from six organ donors were used as controls.	Monotherapy	AAA with intraluminal thrombus	IL-6, IL-10, MMP-9, nitric oxide synthase 2 and NADPH oxidase subunits (gp67-phox, gp91-phox, and gp47-phox)	Rivaroxaban reduced key inflammatory and oxidative stress markers in human AAA sites.	*Br J Clin Pharmacol. 2017;83(12):2661–2670.* [204]
Ancillary analysis of the X-TRA study with AF patients having left atrial/left atrial appendage (LA/LAA) thrombus.	Monotherapy	Relationship between plasma biomarkers and left atrial thrombus resolution in AF patients using rivaroxaban.	▪ Thrombogenesis/fibrinolysis markers: D-dimer, PAI-1, TAT complexes, vWF. ▪ Inflammation markers: IL-6 and hsCRP.	▪ Significant decrease in mean levels of hsCRP, D-dimer, vWF, and TAT with rivaroxaban treatment. ▪ Elevated baseline inflammatory biomarkers (IL-6, hsCRP) were associated with increased chances of thrombus resolution.	*Ann Med. 2018 Sep;50(6):511–518* [205]
127 patients with NVAF	Monotherapy (in comparison with warfarin)	AF and its association with inflammation	IL-2, IL-4, IL-10, TNF-α, IFN-γ, CCL5 (also known as RANTES), CXCL9 (also known as MIG), CCL2 (also known as MCP-1), CXCL10 (also known as IP-10), TGF-β1, ADAMTS13 GDF-15, sICAM-1, p-selectin lipocalin-2 (also known as NGAL), sVCAM-1	▪ Rivaroxaban users had decreased levels of inflammatory cytokines when compared to warfarin users. ▪ Patients using rivaroxaban displayed elevated levels of chemokines MCP-1 compared to warfarin users, and MIG and IP-10 when compared to controls. ▪ AF’s inflammatory profile was less evident in patients taking rivaroxaban than in those using warfarin.	*Front Cardiovasc Med. 2020;7:114.* [206]
Male Wistar rats	Combination therapy of Sunitinib and Rivaroxaban.	Cardiotoxicity induced by Sunitinib.	Serum levels of Ca^2+^, Mg^2+^, Fe^3+^/Fe^2+^, lipid profiles, and cardiac enzymes. Additionally, measurements of oxidant/antioxidant balance gene and protein expressions in cardiac tissues.	▪ Rivaroxaban treatment significantly mitigated the cardiac injuries caused by Sunitinib. This was achieved by restoring antioxidant enzyme levels and attenuating the proinflammatory cascades associated with the cardiac injuries from Sunitinib.	*Cardiovasc Toxicol. 2020;20(3):281–290.* [207]
Patients with AF scheduled for cardioversion without adequate anticoagulation at baseline.	Rivaroxaban (monotherapy) vs. VKA (monotherapy).	Effects of anticoagulation in patients with AF scheduled for cardioversion.	▪ Coagulation: D-dimer, TAT, Prothrombin Fragment (F1.2). ▪ Inflammation: hs-CRP and hs-IL-6.	▪ Reduced levels of D-dimer, TAT, hs-CRP, and hs-IL-6. ▪ F1.2 (levels increased slightly with rivaroxaban treatment, in contrast to VKA which reduced F1.2 levels.	*TH Open. 2020 Jan 23;4(1):e20-e32.* [208]
Multi-center, prospective, randomized, open-label trial involving 179 participants with type 2 diabetes and subclinical inflammation.	Monotherapy	Type 2 diabetes mellitus patients who had subclinical inflammation and were exhibiting stimulated coagulation, activated platelets, and endothelial dysfunction	hsCRP, IL-6, MCP-1, MMP-9 and sCD40L	Rivaroxaban displayed anti-inflammatory effect and improvement of endothelial function.	*Diabetologia. 2021;64(12):2701–2712.* [209]
Observational, multi-center, prospective study involving newly diagnosed AF patients with CKD stages 3b–4.	Monotherapy comparison of Rivaroxaban and Warfarin.	Effects on inflammation, progression of heart valve calcification, and kidney function in AF patients with CKD stages 3b–4.	Plasma inflammatory mediators (measured via ELISA) and cytokine (IL-1b, IL-6, TGF-b, TNF-a) levels.	▪ Long-term treatment with Rivaroxaban resulted in a significant reduction in cytokines, indicating reduced inflammation. Compared to Warfarin, Rivaroxaban was associated with lower serum markers of inflammation.	*Int J Cardiol. 2021 Dec 15;345:90–97.* [210]
NVAF patients treated with rivaroxaban or warfarin.	Monotherapy (rivaroxaban vs. warfarin).	Inflammation and endothelial activation in patients with AF.	Circulating pro-inflammatory extracellular vesicles (EVs) profiles, proteomics of enriched plasma Evs, and levels of soluble P-selectin.	▪ Rivaroxaban-treated NVAF patients showed attenuated inflammation, fundamentally altered circulating EV profiles, a profound decrease in pro-inflammatory protein expression and complement factors, and increased expression of negative regulators of inflammatory pathways. ▪ Additionally, a reduction in circulating levels of soluble P-selectin was observed.	*J Thromb Haemost. 2021;19(10):2583–2595.* [211]
Prospective, randomized study on 228 patients with VTE	*Combination therapy*. Control group received conventional treatment (warfarin or rivaroxaban), whereas the rosuvastatin-intervention group received rosuvastatin 10 mg daily in addition to their conventional treatment.	The impact of rosuvastatin on D-dimer and other inflammatory serum markers in VTE patients.	D-dimer, mean platelet volume (MPV), neutrophil-to-lymphocyte ratio, and platelet-to-lymphocyte ratio.	▪ The study demonstrated that conventional VTE treatment (using warfarin or rivaroxaban) paired with 10 mg rosuvastatin daily for 3 months significantly reduced the levels of D-dimer and MPV compared to patients on anticoagulant therapy alone. ▪ These findings suggest that adding rosuvastatin to standard anticoagulant treatment may be beneficial for VTE patients.	*Clin Cardiol. 2022;45(7):717–722* [212].
Real-world patients with coronary artery disease (CAD) and/or peripheral artery disease (PAD).	Dual pathway inhibition (DPI) using low-dose rivaroxaban and aspirin.	Effect of DPI on plasma inflammation and coagulation markers among patients with CAD and/or PAD.	IL-6, CRP, lipoprotein-associated phospholipase A2, copeptin, and GDF-15.	At the 24-week follow-up, there was a significant reduction in IL-6 and fibrinogen levels and a significant increase in GDF-15.	*J Cardiovasc Pharmacol. 2023 Feb 1;81(2):129–133.* [213]
**Cell, Mice or Combined Models**	**Mono/Combination therapy**	**Condition Investigated**	**Inflammatory Markers Examined**	**Effect Observed**	**Reference**
KK-A(y) mouse model of type 2 diabetes mellitus.	Monotherapy (Rivaroxaban at doses of 5 or 10 mg/kg)	Leukocyte-endothelial interaction and microthrombus formation in the context of type 2 diabetes mellitus.	Leukocyte-endothelial interaction and microthrombus formation in the context of type 2 diabetes mellitus.	▪ Rivaroxaban, especially at a dose of 10 mg/kg, significantly suppressed leukocyte adherence and microthrombus formation. ▪ Rivaroxaban maintained blood fluidity and extended bleeding time.	*Thromb Res. 2014 Feb;133(2):276–80* [214]
LPS-activated monocytes and THP-1 cells (a human monocytic cell line).	Monotherapy (either rivaroxaban or fondaparinux)	Tissue factor (TF) exposure on activated monocytes and macrophages involved in thrombosis through activation of factor X and cytokine release.	TF expression, prothrombinase activity, cytokine release in cell supernatants (with specific focus on IL-8 and TNFα).	▪ Rivaroxaban did not modify TF expression levels on activated cells. ▪ However, procoagulant activity associated with monocytes and macrophages was dose-dependently inhibited by rivaroxaban. ▪ Rivaroxaban suppressed some chemokine secretion produced by activated macrophages.	*Exp Hematol Oncol. 2014 Dec 17;3(1):30.* [215]
Apolipoprotein E-deficient (ApoE-/-) mice	Monotherapy	Atherogenesis	PAR-1 and PAR-2 receptors, MMP-9, MMP-13, COX-2, TNF-a, and in vitro experiments that evaluated mRNA expression of IL-1b and TNF-a in mouse macrophages.	▪ Rivaroxaban treatment (5 mg/kg/day for 20 weeks) reduced atherosclerotic lesion progression in the mice’s aortic arch without altering plasma lipid levels or blood pressure. ▪ Rivaroxaban led to decreased lipid deposition, reduced collagen loss, lowered macrophage accumulation, and a decline in MMP-9 expression in the aortic root’s atherosclerotic plaques. ▪ Rivaroxaban significantly reduced mRNA expression of inflammatory molecules like MMP-9 and TNF-a in the abdominal aorta. ▪ In vitro experiments showed that FXa’s inflammatory effects were countered by Rivaroxaban.	*Atherosclerosis. 2015;242(2):639–646.* [216]
▪ Human umbilical vein endothelial cells (**in vitro**) ▪ An ischemic hind limb mouse model (**in vivo**)	Monotherapy	The role of FXa in inducing cell senescence and its effect on tissue inflammation and regeneration.	Senescence-associated β-galactosidase, Insulin-like Growth Factor Binding Protein (IGFBP)-5, Early Growth Response Protein (EGR)-1, p53, and Cyclin-dependent kinase Inhibitor 2A (p16INK4a) (via RT-qPCR array) and expression of cytokines ((IL-1b, IL-6, MCP-1 and ICAM-1)	▪ Rivaroxaban decreased FXa-induced endothelial cell senescence and restored cell proliferation. ▪ Rivaroxaban significantly mitigated the FXa-induced impaired angiogenesis in the ischemic hind limb mouse model.	*Sci Rep. 2016 Oct 18;6:35580.* [217]
Mouse model with polyurethane catheters placed unilaterally into the external jugular vein (EJV).	Monotherapy (either rivaroxaban or vehicle).	Dysfunction of indwelling central venous catheters (CVC) due to tissue ingrowth or clotting.	Plasma MCP-1 levels, External Jugular Vein(EJV) MMP-9 levels, cell proliferation (anti-Ki67), macrophage infiltration (anti-MAC387).	▪ Rivaroxaban significantly improved CVC patency compared to the vehicle-treated group. Plasma MCP-1 levels were lower in the rivaroxaban-treated mice compared to vehicle-treated at 21 days. Cell proliferation and MMP-9 protein levels were also reduced in the rivaroxaban group at day 7.	*Thromb Res. 2016 Aug;144:106–12* [218]
Rat model of brain ischemia/reperfusion injury using male Wistar rats.	Monotherapy (prestroke anticoagulation with rivaroxaban).	Influence of prestroke anticoagulation with rivaroxaban on stroke severity and associated effects on thrombo-inflammation.	Thrombin/antithrombin complex, intracerebral thrombus formation, CD68-immunoreactivity, expression of cytokines (IL-1b, TNF-α, INF-γ), and adhesion molecules (specifically ICAM-1 and VCAM-1).	▪ Rivaroxaban significantly reduced stroke size and functional deficits. It reduced thrombin-mediated thrombus formation without increasing the risk of intracranial hemorrhage. ▪ Rivaroxaban dampened the postischemic inflammatory response in the brain by downregulating ICAM-1 expression and decreasing activation of CD68+-immune cells.	*Thromb Haemost. 2016 Apr;115(4):835–43.* [219]
Female SJL/J mice immunized with PLP139-151 to induce autoimmune experimental encephalomyelitis (EAE).	Monotherapy (either warfarin or rivaroxaban).	Effects of anticoagulants (warfarin and rivaroxaban) on autoimmune experimental encephalomyelitis (EAE) as a model for multiple sclerosis.	Neurological deficit scores, histopathological analyses of inflammatory lesions in the spinal cord.	▪ Preventive treatment with rivaroxaban reduced the maximum EAE score and decreased inflammatory lesions in the spinal cord compared to the control group. ▪ No beneficial effects were observed with therapeutic treatment using warfarin. ▪ No signs of intraparenchymal hemorrhage at the site of inflammatory lesions were detected.	*J Neuroinflammation. 2017 Jul 28;14(1):152.* [220]
Human umbilical vein endothelial cells that natively express protease-activated receptor-1 and -2	Monotherapy	Function of rivaroxaban in the inactivated coagulation cascade and its role in altering gene transcripts, especially those of pro-inflammatory genes, upon FXa stimulation.	▪ Gene transcripts of pro-inflammatory genes from DNA microarray. ▪ Expressions of MCP-1, ICAM-1, and IL-8.	▪ In unstimulated conditions, high-dose rivaroxaban did not alter gene transcripts of pro-inflammatory genes. ▪ Upon FXa stimulation, rivaroxaban suppressed the expressions of pro-inflammatory genes such as MCP-1, intracellular adhesion molecule-1, and interleukin-8. ▪ Rivaroxaban inhibited MCP-1 expression observed both in quantitative polymerase chain reaction and ELISA. ▪ Rivaroxaban might not influence gene modulation in the inactivated coagulation state but can reduce endothelial damage caused by FXa and pro-inflammatory cytokine genes.	*J Pharmacol Sci. 2017;133(3):156–161.* [221]
In vitro study using human atherosclerotic plaques from carotid endarterectomy and vascular smooth muscle cells (VSMC) for experimentation.	Monotherapy	Progression and mechanisms of atherosclerotic plaques, specifically the role of coagulation FXa in inducing endothelial cell senescence.	PARs, IGFBP-5, p53, and other inflammatory cytokines (IL-1b, IL-6, MCP-1, IGFBP-5).	▪ Rivaroxaban inhibited FXa signaling, which significantly reduced FXa-induced VSMC senescence and the production of inflammatory cytokines.	*Sci Rep. 2017 Dec 7;7(1):17172* [222]
▪ In vivo: C57BL/6 mice with Acute Lung Injury (ALI). ▪ In vitro: LPS-stimulated human umbilical vein endothelial cells.	Monotherapy	Role of coagulation in acute lung injury (ALI) and the effect of rivaroxaban on it.	▪ In mice: TNF-α, IL-1β, IL-6, total protein, and Evans blue in bronchoalveolar lavage fluid. ▪ In vitro: LPS-induced increases in membrane permeability, proinflammatory cytokine levels, and apoptosis.	▪ Reduced neutrophil sequestration and preservation of lung tissue architecture in mice. ▪ Significantly lowered levels of inflammatory markers in mice bronchoalveolar lavage fluid. ▪ Ameliorated LPS-induced PAR-2 increase and NF-κB activation in mice. ▪ In vitro: increased cell viability, attenuation of LPS-induced inflammatory responses, and reduced apoptosis. ▪ Rivaroxaban inhibited the phosphorylation of TAK1 and p65, suggesting that it attenuates ALI and inflammation by inhibiting the PAR-2/NF-κB signaling pathway.	*Am J Transl Res. 2018;10(8):2335–2349.* [223]
Transluminal femoral artery injury in C57BL/6 mice induced by a straight wire.	Monotherapy	The role of pharmacological blockade of FXa in attenuating neointima formation following wire-mediated vascular injury.	IL-1β, IL-6, TNF-α, stromal cell-derived factor (SDF)-1, TGF-β1, granulocyte-macrophage colony stimulating factor (GM-CSF)	▪ Rivaroxaban significantly attenuated neointima formation in injured arteries compared to control. ▪ Rivaroxaban reduced the expression of inflammatory molecules in injured arteries and in mouse peritoneal macrophages. ▪ In vitro experiments further showed that rivaroxaban blocked the FXa-induced proliferation and migration of rat VSMC and inhibited the inflammatory activation of macrophages and VSMC.	*Eur J Pharmacol. 2018;820:222–228.* [224]
Ten-week-old male CL57/B6 mice subjected to transverse aortic constriction (TAC) surgery	Monotherapy	Atrial fibrillation and inflammatory atrial fibrosis	mRNA levels of TNF-α, IL-6, IL-1β, MCP-1, cardiac PAR-2 expression	▪ Rivaroxaban attenuated TAC-induced left atrial thrombus formation, upregulation of cardiac PAR-2, and cardiac inhomogeneous interstitial fibrosis, and infiltration of macrophages. It also suppressed the overexpression of the mentioned inflammatory markers in the left atrium. ▪ In cardiac fibroblasts, rivaroxaban pre-incubation suppressed the upregulation of PAR-2 induced by persistent intermittent stretch.	*J Cardiol. 2018;71(3):310–319.* [225]
Wild-type (WT) and PAR-2-/- mice subjected to left anterior descending artery (LAD) ligation.	Monotherapy	Cardiac injury and heart failure post-LAD ligation.	IL-6, IL-1β, and MPO (marker for neutrophil infiltration)	▪ Rivaroxaban did not significantly affect the expression of inflammatory mediators or the neutrophil marker at day 2 after LAD ligation.	*Thromb Res. 2018;167:128–134. (***)* [226]
Male Albino rats	Monotherapy	Liver fibrosis induced by carbon tetrachloride	TNF-α, IL-1β and hydroxyproline	Rivaroxaban restored the inflammatory markers associated with liver fibrosis.	*J Biochem Mol Toxicol. 2019;33(5):e22287.* [227]
Murine model of ischemic cardiomyopathy (ICM) using SR-BI KO/ApoeR61h/h mice (Hypo E mice).	Monotherapy	Effects of FXa inhibitors on atherosclerosis and cardiac remodeling post-MI in Hypo E mice.	▪ Cardiac PAR2 levels ▪ Pro-inflammatory genes (IL-1β, IL-6, NF-κB, TNF-α, MMP-9, MMP-12, TIMP1, TGFβ 1, COL-1, COL-3, PAR1, PAR2	▪ Rivaroxaban treatment after MI attenuated heart failure, reduced aortic atherosclerosis, coronary occlusion, and significantly decreased cardiac fibrosis and also decreased cardiac PAR2 levels and pro-inflammatory genes. ▪ In vitro, rivaroxaban increased cell viability against hypoxia in cardiac myocytes and reduced hypoxia-induced inflammation and fibrosis-related molecules in cardiac fibroblasts. ▪ The effects of the PAR2 antagonist against hypoxia-induced inflammation were comparable to rivaroxaban in cardiac fibroblasts.	*J Atheroscler Thromb. 2019;26(10):915–930.* [228]
MI (myocardial infarction) induced in wild-type mice through permanent ligation of the left anterior descending coronary artery.	Monotherapy (rivaroxaban added to regular chow diet).	Protective effects of rivaroxaban against cardiac remodeling after MI.	mRNA expression levels of TNF-α, TGF-β, PAR-1, IL-1 β, IL-6, MCP-1, MMP-2 MMP-9, PAR-2, A-type natriuretic peptides, B-type natriuretic peptides, and phosphorylation of extracellular signal-regulated kinase.	▪ Rivaroxaban improved cardiac systolic function, reduced infarct size, and cardiac mass. ▪ Rivaroxaban downregulated the mRNA expression of the mentioned inflammatory markers in the infarcted and non-infarcted areas. ▪ Rivaroxaban decreased cardiomyocyte hypertrophy and phosphorylation of extracellular signal-regulated kinase in the non-infarcted area.	*Circ Rep. 2020 Mar 4;2(3):158–166.* [229]
Wistar rats weighing 200–250 g.	Combination therapy (Rivaroxaban with Sunitinib).	Nephrotoxicity induced by Sunitinib.	TNF-α/NF-kB signaling pathways, Malondialdehyde, Catalase, Glutathione, Glutathione reductase, Caspase-3, IL-17, MCP-1, Inhibitor of KBα.	Rivaroxaban treatment restored altered levels of inflammatory markers and exhibited nephroprotective effects against Sunitinib-induced nephrotoxicity by inhibiting oxidative stress-induced apoptosis and inflammation.	*J Thromb Thrombolysis. 2020 Aug;50(2):361–370.* [230]
Male rats where inflammation was induced post-rivaroxaban therapy using LPS.	Monotherapy	LPS-induced acute vascular inflammatory response.	▪ Circulating levels of IL-6, MCP-1, VCAM-1, and ICAM-1 in plasma. ▪ Gene expression analysis in isolated aorta for iNOS, VCAM-1, and MCP-1.	▪ Rivaroxaban pre-treatment significantly reduced LPS mediated increase in IL-6, MCP-1, and VCAM-1 in plasma both 6 h and 24 h post-LPS injection. ▪ Similar reductions were observed in the aorta for iNOS, VCAM-1, and MCP-1 gene expression. ▪ Rivaroxaban pre-treatment improved LPS-induced contractile dysfunction in aortic rings.	*PloS One. 2020;15(12):e0240669* [231].
In vitro study using the tissue factor-expressing prostate carcinoma cell line, 22Rv1. Whole blood was also stimulated with LPS or phorbol-myristate-acetate (PMA).	Monotherapy comparisons of rivaroxaban, dalteparin, and tinzaparin.	Cancer-associated thrombosis (CAT) and the influence of myeloperoxidase (MPO) on anticoagulant activity.	Tumor cell-induced procoagulant activity, platelet aggregation, and the impact of the cationic leukocyte-derived enzyme, MPO. Thrombin generation in plasma supernatants from LPS- or PMA-stimulated whole blood was also measured.	▪ Rivaroxaban’s anticoagulant activity was not attenuated by MPO, unlike dalteparin and tinzaparin at trough levels. ▪ This suggests that rivaroxaban remains effective in the context of paraneoplastic leukocyte activation where MPO is present.	*J Thromb Haemost. 2020 Dec;18(12):3267–3279.* [232]
Ang II-infused ApoE-/- mice and calcium chloride-induced AAA models, as well as human aortic endothelial cells.	Monotherapy	AAA	IL-6, IL-8, IL-1β, MCP-1, MMP-2 as well as adhesive molecules were examined in relation to FXa stimulation and rivaroxaban treatment.	▪ Rivaroxaban AAA progression by decreasing leukocyte infiltration, reducing the expression of IL-1β, IL-6, IL-8, and MCP-1, reducing the expression of MMP-2 in the aortic wall, and counteracting FXa-induced aortic wall inflammation. ▪ In human aortic endothelial cells, rivaroxaban also reversed the upregulation of inflammatory cytokines and adhesive molecules induced by FXa stimulation.	*Vascul Pharmacol. 2021;136:106818.* [233]
Young adult male Wistar Albino type rats with surgically induced Achilles tendon injury followed by primary repair.	Monotherapy (rivaroxaban vs. nadroparin calcium vs. no medication)	Effects of antithrombotic-adjusted prophylactic doses on Achilles tendon healing	Inflammatory cells, capillary vessels, fibroblasts, degrees of inflammation, neovascularization, fibroblastic activity, and collagen fiber sequencing for histopathological evaluation.	▪ Rivaroxaban reduced the number of inflammatory cells compared to both the control group and the nadroparin calcium group. It also influenced the sequencing of collagen fibers and increased the presence of mature collagen fibers. ▪ Both rivaroxaban and nadroparin calcium had a beneficial effect on Achilles tendon healing.	*J Hand Microsurg. 2021;15(2):133–140.* [234]
**The investigation consisted of two studies:** **Study 1:** PAR2 deficient (PAR2-/-) and wild-type mice infused with angiotensin II (Ang II) or a vehicle. **Study 2:** Spontaneously hypertensive rats (SHRs) treated after 8 h of right atrial rapid pacing.	Monotherapy (either rivaroxaban, warfarin, or vehicle).	Role of PAR2 signaling in AF arrhythmogenesis and the potential ameliorating effect of rivaroxaban on atrial inflammation and AF prevention.	mRNA expression of collagen1 and collagen3, gene expression of inflammatory (TNF-α, MCP-1, TGF-β, Col1a1, Col31, F2R and F2l1) and fibrosis-related biomarkers in the atrium.	▪ In Ang II-treated PAR2-/- mice, there was a lower incidence of AF and reduced atrial mRNA expression of collagen1 and collagen3 compared to wild-type mice. ▪ In SHRs, rivaroxaban significantly reduced AF inducibility and expression levels of genes related to inflammation and fibrosis in the atrium compared to warfarin or vehicle.	*Circ J. 2021;85(8):1383–1391.* [235]
Pilot, single-center, randomized, double-blind, placebo-controlled, crossover study with subjects having sickle cell anemia.	Monotherapy (either rivaroxaban or placebo).	Sickle cell disease (SCD) and its association with coagulation activation.	Thrombin-antithrombin complex, D-dimer, inflammatory (hs-CRP, IL-6, IL-2 and IL-8) and endothelial activation markers, measures of microvascular blood flow.	▪ Rivaroxaban resulted in a decrease from the baseline of thrombin-antithrombin complex versus placebo, but the difference wasn’t statistically significant. ▪ No significant differences were observed in changes from the baseline of D-dimer, inflammatory, endothelial activation markers, or measures of microvascular blood flow.	*Transfusion. 2021 Jun;61(6):1694–1698. (***)* [236]
Male Ldlr-/- mice fed a western-type diet to induce atherosclerosis.	Combination of aspirin (given in water) and rivaroxaban (given in the diet) compared to each agent alone.	Atherosclerosis in Ldlr-/- mice.	Expression of 55 proteins in the aorta and plasma (specific proteins not listed in provided information).	▪ Both aspirin and rivaroxaban reduced atherosclerosis in the mice. ▪ The combination reduced macrophage content and apoptosis in the lesions. ▪ However, the combination did not show a reduction in atherosclerosis beyond what was observed with each agent alone.	*Atherosclerosis. 2022 Mar;345:7–14* [237]
LPS stimulation of PBMCs or citrate-anticoagulated whole blood.	Monotherapy with Protein Disulfide Isomerase Inhibitor III (PACMA-31) or DMSO vehicle.	Regulation of TF in monocytes by protein disulfide isomerase (PDI) and the effect of PACMA-31, a specific PDI inhibitor.	TF expression (antigen, procoagulant activity, mRNA), release of IL-6 and TNFα, and LPS-induced signaling pathways.	▪ Rivaroxaban did not prevent the stimulatory effect of PACMA-31 on inflammatory monocytes.	*Thromb Res. 2022;220:48–59. (***)* [238]
**In vivo**: Various strains of mice (wild-type and NLRP3 knockout) fed with standard chow or a high-fat diet. **In vitro**: examination with mice aortic endothelial cells (MAECs) and smooth muscle cells (MOVASs).	Monotherapy	Therapeutic role of rivaroxaban in attenuating vascular lesions and dysfunction in type 2 diabetes mellitus (T2DM) mice	NLR family pyrin domain containing 3 (NLRP3) inflammasome activation, vascular tension, intima-media thickness, collagen deposition, PAR-1, PAR-2, mitogen-activated protein kinase (MAPK), NF-κB.	▪ Rivaroxaban significantly protected against vascular dysfunction in T2DM mice, ameliorating key indicators of vascular damage. ▪ Rivaroxaban attenuated NLRP3 inflammasome activation and blocked inflammation and cell dysfunction in MAECs and MOVASs through specific pathways.	*J Cell Physiol. 2022 Aug;237(8):3369–3380.* [239]
ICH induced by collagenase injection into the striatum of wild-type (C57BL/6J) anticoagulated mice (warfarin or rivaroxaban) and Mmp10 -/- mice.	Monotherapy using either prothrombin complex concentrate (PCC) or CM-352 (MMP-fibrinolysis inhibitor).	Intracranial hemorrhage (ICH) associated with oral anticoagulants (specifically rivaroxaban).	PAI-1, IL-6, neutrophil infiltration, thrombin-activatable fibrinolysis inhibitor (TAFI) activation.	▪ Rivaroxaban-ICH was ameliorated by both PCC and CM-352 treatments, with CM-352 reducing neutrophil infiltration in the hemorrhage area. CM-352 diminished MMPs and rivaroxaban-associated fibrinolytic effects.	*Thromb Haemost. 2022 Aug;122(8):1314–1325* [240]

The studies marked with (***) indicate no significant impact of NOAC on specific inflammatory markers, suggesting limited influence despite research efforts in this area.

## Data Availability

The datasets generated and/or analyzed during the current study are not publicly available but are accessible from the corresponding author upon reasonable request. Interested researchers may contact the corresponding author for data access inquiries, subject to compliance with any applicable privacy or confidentiality obligations.

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
