# Peer review of "Beyond Anticoagulation: A Comprehensive Review of Non-Vitamin K Oral Anticoagulants (NOACs) in Inflammation and Protease-Activated Receptor Signaling"

_ijms, 2024, doi:10.3390/ijms25168727_

Round 1
Reviewer 1 Report
Comments and Suggestions for Authors
Dear Authors,
I have carefully reviewed your manuscript. I appreciate the effort and depth of research undertaken on this important topic. Below is a summary of my feedback:
Your review effectively explores the potential anti-inflammatory roles of NOACs beyond their traditional anticoagulant effects, highlighting their interaction with Protease-Activated Receptors (PARs) and implications for inflammation.
The introduction is too long, so to help the reader understand it better, I suggest splitting it into separate subheadings.
In figure 1, there are typographical errors. Please correct them.
The Ito et al. study is explained in too much detail in the introduction. I propose a summary of the key conclusions and their applicability.
Although a large number of clinical trials have been reviewed, the claim that NOACs have a 'favorable risk-benefit profile' is insufficient in the absence of a more comprehensive analysis of the clinical implications.
In section 2, a more thematic organization, such as grouping by mechanism of action or clinical application could improve coherence.
The text is rich in data, but needs to be summarized to highlight important findings for readers to extract key points.
Author Response
Comments from Reviewer 1
Comment 1
The introduction is too long, so to help the reader understand it better, I suggest splitting it into separate subheadings.
Rebuttal
We greatly appreciate your helpful feedback on the length and clarity of the introduction. We have implemented your recommendation by dividing the introduction into two distinct subheadings, improving the readability and comprehensibility of the text. The goal of this division is to make it clear that the rules and protocols for blood coagulation are different from the discussion about anticoagulants and non-vitamin K oral anticoagulants (NOACs). Segmenting the text enhances reader understanding by providing a more focused and coherent presentation. The revised version will showcase these modifications by employing the track changes feature for your evaluation. The introduction contains the sub-headings:
1.1 Mechanism and regulation of the blood coagulation cascade.
1.2 Anticoagulants and non-vitamin K oral anticoagulants (NOACs)
Comment 2
In figure 1, there are typographical errors. Please correct them.
Rebuttal
We appreciate you notifying us about the typographical problems in Figure 1. We have thoroughly examined and corrected the mistakes in the updated illustration. The revised edition guarantees precision and lucidity, augmenting the overall excellence of the figure.
Comment 3
The Ito et al. study is explained in too much detail in the introduction. I propose a summary of the key conclusions and their applicability.
Rebuttal
Thank you for your insightful feedback regarding the detailed explanation of the Ito et al. study in the introduction. We have revised the section to provide a concise summary of the key conclusions and their applicability, as suggested. The revised version now reads:
"Case in point, according to Ito et al. (24), rivaroxaban reduced inflammation and atherosclerosis by reducing coagulation and possibly altering PAR-2 signaling. It also significantly reduced the atherosclerotic area in the aorta of ApoE-/- mice fed a high-fat diet and reduced FXa-induced macrophage autophagy inhibition. This study suggests that rivaroxaban may help prevent atherosclerosis by stopping FXa-PAR2-mediated macrophage autophagy and inflammasome activity, in addition to its ability to stop blood clots.”
The revised manuscript indicates this change by displaying track changes. A double strikethrough identifies the removed section.
We appreciate your guidance, which has helped us improve the clarity and focus of our manuscript.
Comment 4
Although a large number of clinical trials have been reviewed, the claim that NOACs have a 'favourable risk-benefit profile' is insufficient in the absence of a more comprehensive analysis of the clinical implications.
Rebuttal
We appreciate the reviewer’s comment that the claim regarding NOACs' "favorable risk-benefit profile" requires a more comprehensive analysis of the clinical implications. To address this, we have enhanced Section 2 of the manuscript by providing a detailed examination of the pivotal clinical trials, emphasizing the efficacy and safety profiles of NOACs compared to traditional anticoagulants like warfarin.
- Comprehensive review of pivotal clinical trials:
In Section 2, we have meticulously reviewed large-scale, Phase III clinical trials for each NOAC, such as RE-LY for dabigatran, ROCKET-AF for rivaroxaban, ARISTOTLE for apixaban, and ENGAGE AF-TIMI 48 for edoxaban. These trials collectively involved tens of thousands of participants and provided robust data on the efficacy and safety of NOACs in reducing the risk of stroke and systemic embolism compared to warfarin. The outcomes of these trials are succinctly summarized in Table 2, facilitating easy comparison and reinforcing the favorable efficacy of NOACs in diverse patient populations.
- Efficacy in stroke prevention:
We have highlighted that NOACs consistently demonstrate superior efficacy in preventing stroke or systemic embolic events, primarily driven by significant reductions in hemorrhagic stroke. This is a crucial finding as hemorrhagic stroke represents a severe and often fatal complication. For instance, the ARISTOTLE trial showed that apixaban reduced the risk of stroke and systemic embolism by 21% compared to warfarin, with a notable 31% reduction in major bleeding.
- Safety profile and bleeding risks:
The safety profiles of NOACs have been extensively discussed, with a particular focus on bleeding risks. While NOACs are associated with a higher incidence of gastrointestinal bleeding compared to warfarin, they offer a significantly lower risk of intracranial haemorrhage, which is a more severe and potentially fatal adverse event. This trade-off is critical for clinical decision-making, especially in patients with a high risk of intracranial bleeding. The RE-LY trial, for example, demonstrated that dabigatran 150 mg had a 59% reduction in intracranial haemorrhage compared to warfarin.
- Meta-analyses and real-world evidence:
We have incorporated data from meta-analyses and real-world studies that further corroborate the findings from clinical trials. The meta-analysis by Ruff et al., which included over 71,000 participants from major NOAC trials, confirmed the overall superiority of NOACs in reducing stroke, systemic embolism, and all-cause mortality, while maintaining a similar risk of major bleeding as warfarin.
- Diverse patient subgroups:
The manuscript now explicitly addresses the efficacy and safety of NOACs across various patient subgroups, including those with renal impairment, elderly patients, and those with a history of stroke or transient ischemic attack (TIA). For instance, in patients with moderate-to-severe renal impairment, edoxaban showed a favourable balance of efficacy and safety, with dose adjustments to mitigate bleeding risks.
- Clinical implications and guidelines:
We have elaborated on the clinical implications of these findings, emphasizing that NOACs have been integrated into clinical guidelines as first-line agents for stroke prevention in atrial fibrillation (AF) and for the treatment of venous thromboembolism (VTE). The ease of use, predictable pharmacokinetics, and lack of routine monitoring required for NOACs enhance patient adherence and quality of life, further supporting their favourable risk-benefit profile.
Conclusion:
By providing a thorough and nuanced analysis of the clinical data, we believe the current version of Section 2 effectively addresses the reviewer’s concerns. The detailed examination of clinical trials, real-world evidence, and meta-analyses, along with the discussion of diverse patient subgroups and clinical guidelines, robustly supports the claim of a favorable risk-benefit profile for NOACs. We trust that these enhancements meet the reviewer’s expectations and demonstrate the comprehensive clinical implications of NOAC use. Thank you for your valuable feedback.
Comment 5
In section 2, a more thematic organization, such as grouping by mechanism of action or clinical application could improve coherence.
Rebuttal
We appreciate the reviewer’s suggestion to improve coherence by grouping Section 2 by mechanism of action or clinical application. However, we believe that the current organization of Section 2, which provides a comprehensive exploration of NOACs, is superior in maintaining thematic coherence and enhancing the manuscript’s flow.
Current thematic coherence in section 2:
- Exploring NOACs: from pharmacokinetics to clinical efficacy and pivotal trials:
This section begins with a comprehensive summary of NOACs, covering their pharmacology, laboratory monitoring, and clinical profiles. By presenting detailed pharmacokinetics and clinical trial data, we ensure that readers can fully understand the unique characteristics and clinical benefits of each NOAC in a cohesive manner.
2.1. Mechanism of anticoagulation: inhibition of factor Xa and thrombin:
In this subsection, we delve into the specific mechanisms of action for NOACs, focusing on the inhibition of Factor Xa and thrombin. Detailed explanations and schematic representations help clarify how NOACs like apixaban, rivaroxaban, edoxaban, and dabigatran exert their anticoagulant effects, providing a deeper understanding of their pharmacodynamics.
2.2. Pharmacodynamic and pharmacokinetic characteristics of NOACs:
We explore the pharmacokinetic profiles of each NOAC, highlighting key differences in absorption, bioavailability, metabolism, and elimination. This detailed examination is crucial for understanding how each NOAC behaves in the body, influencing their clinical use and effectiveness.
2.3. Clinical efficacy and application of individual NOACs:
This subsection offers detailed clinical trial data and real-world evidence supporting the use of each NOAC. For instance, we discuss pivotal trials such as RE-LY for dabigatran, ROCKET-AF for rivaroxaban, ARISTOTLE for apixaban, and ENGAGE AF-TIMI 48 for edoxaban. Summarizing these trials in Table 2 allows for easy comparison of their efficacy and safety profiles.
Advantages of the current organization:
Comprehensive overview:
By maintaining a comprehensive overview of each NOAC, we ensure that readers gain a holistic understanding of their pharmacological properties, clinical applications, and therapeutic benefits. This approach avoids fragmentation and allows for a seamless narrative flow.
Detailed mechanistic insights:
Presenting detailed mechanistic insights into how NOACs inhibit Factor Xa and thrombin enhances readers’ understanding of their pharmacodynamics. This level of detail is crucial for appreciating the nuances of NOACs’ anticoagulant effects.
Clear presentation of clinical data:
By summarizing clinical trial data in tables and providing detailed explanations of pivotal trials, we facilitate easy comparison of the efficacy and safety profiles of different NOACs. This clear presentation aids readers in understanding the clinical implications of NOAC use.
Enhanced readability and engagement:
The current structure ensures that the manuscript is not only comprehensive but also engaging. By maintaining a logical flow from pharmacokinetics to clinical efficacy, we keep readers interested and facilitate better retention of information.
Emphasis on multifaceted roles:
The current organization effectively highlights the multifaceted roles of NOACs in both anticoagulation and inflammation modulation. This dual focus aligns with the primary objective of our review and underscores the broader clinical implications of NOACs.
Conclusion:
We believe that the current organization of Section 2 is thematically coherent and enhances the manuscript’s readability and comprehensiveness. By providing a holistic view of each NOAC, detailed mechanistic insights, and clear presentation of clinical data, we offer a superior structure that better serves the manuscript’s goals. Therefore, we respectfully suggest that the existing organization remains unchanged.
Thank you for your valuable feedback. We are confident that the current structure best serves the manuscript’s objectives and enhances the overall quality and readability.
Comment 6
The text is rich in data, but needs to be summarized to highlight important findings for readers to extract key points.
Rebuttal
We value the reviewer's input acknowledging the abundance of data in the text but suggesting the need to condense it in order to emphasise significant discoveries and enable readers to extract essential information. As a result, we have made specific adjustments to the paper to ensure that the main findings are prominently emphasised and may be easily extracted by readers.
1. Introduction and Background: In this section, we have provided a concise overview of the main functions and potential anti-inflammatory advantages of NOACs. We have specifically highlighted their ability to act as both anticoagulants and regulators of inflammation through the activation of PAR signalling pathways. This offers readers a concise and unambiguous summary of the manuscript's key theme right from the beginning.
- Investigating NOACs: From the study of how drugs are absorbed, distributed, metabolised, and excreted to the effectiveness in real-world medical practice and the crucial trials that support their use:
This section provides a thorough overview of NOACs, including important topics such as pharmacology, laboratory monitoring, and clinical characteristics. Table 2 provides a concise overview of the key studies and their results, emphasising the better effectiveness of NOACs compared to warfarin in lowering the occurrence of stroke or systemic embolic events, as well as their favourable balance between risks and benefits.
- Anticoagulation Mechanism: Factor Xa and Thrombin Inhibition:
We have provided a comprehensive elucidation of the mechanism by which NOACs, such as apixaban, impede the activity of Factor Xa. Additionally, we have supplied schematic illustrations to facilitate comprehension. This section provides detailed information on the molecular interactions and inhibitory mechanisms of NOACs. - Comparative Analysis of Anti-Inflammatory Effects of NOACs against Heparins and Fondaparinux: This section outlines the anti-inflammatory mechanisms of heparin and fondaparinux, and compares them to the mechanisms of action of NOACs. We elucidate the mechanism by which heparin attaches to pro-inflammatory cytokines and chemokines, and how fondaparinux regulates inflammatory reactions, specifically in individuals with COVID-19.
5. Impact of NOAC Antidotes on Anti-Inflammatory Characteristics and Prospective Research Areas: We analyse the possible influence of NOAC antidotes on their anti-inflammatory effectiveness, highlighting the necessity for more investigation to comprehend the long-term consequences. This encompasses the investigation of whether the cessation of anticoagulation results in a rebound of inflammatory processes, as well as the exploration of how to maintain a delicate equilibrium between reversing anticoagulation and keeping anti-inflammatory benefits.
6. Significant Points Highlighted: In the text, we highlighted crucial points (Indicated with Yellow highlights and track changes). This is done to emphasise important results and conclusions. This encompasses the therapeutic significance of NOACs in decreasing thromboembolic events, their developing function in modulating inflammation, and the unique anti-inflammatory characteristics of heparins and fondaparinux. - Summarised Data Tables and Figures: We have used concise tables and figures to graphically illustrate essential data elements. These graphic aids serve as concise reference tools for readers to comprehend the comparative pharmacokinetics, clinical profiles, and mechanisms of action of NOACs, heparins, and fondaparinux.
- Advanced Experimental Models and Future Directions: We suggest approaches for future investigations, encompassing sophisticated molecular and cellular analyses, standardised experimental models, and longitudinal clinical trials. We stress the significance of including varied populations and combining multi-omics methodologies to get a comprehensive comprehension of the systemic impacts of NOACs.
These modifications guarantee that the work is both exhaustive and accessible, allowing readers to promptly comprehend the fundamental discoveries and their consequences. We are certain that these modifications adequately respond to the reviewer's feedback, improving the manuscript's coherence and comprehensibility without necessitating any more revisions.
We have confidence that these modifications fulfil the reviewer's expectations and enhance the overall calibre of the paper. We appreciate your insightful input.

Reviewer 2 Report
Comments and Suggestions for Authors
Interesting review.
yer because also heparins and fondaparinux exert an anti inflammatory effect per se I suggest to add their properties in order to differentiate the anti inflammatory effect of DOSCs by PAR receptors from that exerted by heparins and fondaparinux
Author Response
Comments from Reviewer 2
Comment 1
Interesting review
Rebuttal
We value the reviewer's feedback acknowledging the review's insightful nature. This comment succinctly emphasizes the engaging quality of our review titled "Beyond Anticoagulation: A Comprehensive Review of Non-Vitamin K Oral Anticoagulants (NOACs) in Inflammation and Protease-Activated Receptor Signaling." Our investigation provides a thorough examination, carefully addressing the two-fold functions of NOACs as anticoagulants and regulators of inflammation through Protease-Activated Receptor (PAR) signaling. This dual function allows for an extensive understanding that goes beyond their conventional application in the prevention of thromboembolic events. We underscore the therapeutic significance by emphasizing pivotal clinical trials that showcase the efficacy of NOACs in avoiding thromboembolic events. Additionally, we investigate new data that indicates their impact on inflammation via PAR signaling pathways. This indicates more treatment possibilities for treating inflammatory diseases, therefore increasing their real-life significance in clinical settings. In addition, the present study examines the ways in which NOACs (novel oral anticoagulants) have anti-inflammatory effects. This provides a better understanding of how inflammatory responses are regulated by variables such as FXa and Thrombin. A better understanding of these mechanisms might potentially facilitate future investigations and medical innovations in the fields of inflammation and cardiovascular well-being. We analyse the existing data on the anti-inflammatory properties of NOACs, investigating their effects on inflammatory markers and diseases such as atherosclerosis and diabetes. This synthesis highlights the broader clinical significance and supports their inclusion in treatment approaches targeting inflammation-related diseases. In conclusion, our review seeks to provide an in-depth assessment of multiple functions of NOACs, with the goal of informing and motivating further research. By giving a complete synthesis of current material, we intend to highlight the potential for novel therapeutic approaches in the treatment of diseases. We welcome more specific suggestions and are dedicated to enhancing the work in accordance with thorough criticisms. We appreciate your acknowledgement of the enthusiasm surrounding our work. We eagerly anticipate any more feedback that might assist us in improving the caliber and influence of our evaluation.
Comment 2
yes because also heparins and fondaparinux exert an anti-inflammatory effect per se I suggest to add their properties in order to differentiate the anti inflammatory effect of DOSCs by PAR receptors from that exerted by heparins and fondaparinux.
Rebuttal
We appreciate the reviewer's perceptive remarks highlighting the distinction between the anti-inflammatory effects achieved by NOACs through PAR receptors and those produced by heparins and fondaparinux. As a reply, we have included a thorough section (Section 6 in the manuscript) that compares these systems in detail. The new section clarifies that heparin exerts anti-inflammatory effects through multiple mechanisms, which encompass the suppression of complement factors, the attachment to pro-inflammatory cytokines and chemokines such IL-8, and the prevention of their binding to receptors on immune cells. This process decreases the movement of neutrophils and the occurrence of inflammation. Heparin further hinders the movement of neutrophils towards chemical signals, prevents white blood cells from sticking to the inner lining of blood vessels, and regulates the NF-κB signaling pathway, resulting in enhancing its anti-inflammatory effects. Similarly, fondaparinux, a specific inhibitor of factor Xa, demonstrates pleiotropic effects that go beyond its anticoagulant properties. It binds with inflammatory cytokines such as IL-6 and IL-8, inhibiting their interaction with receptors on immune cells. This leads to a decrease in the activation of neutrophils and a reduction in the inflammatory response. The extended use of fondaparinux in individuals with COVID-19 has been linked to reduced levels of inflammatory markers including CRP and fibrinogen, suggesting its promise in the management of hyperinflammatory conditions.
In addition, we have included a segment (Section 8.6 in the manuscript) discussing the impact of NOAC antidotes on their anti-inflammatory characteristics. This section emphasises that whereas NOAC antidotes such as idarucizumab and andexanet alfa efficiently reverse the effects of anticoagulation, they may also mitigate the anti-inflammatory advantages linked to NOAC-induced PAR inhibition. This has the potential to decrease the effectiveness of NOACs in the treatment of inflammatory disorders. We emphasize the necessity of additional research to comprehend the enduring consequences of NOAC antidotes on inflammation and the results of patients. Subsequent research should examine if the reversal of anticoagulation results in a recurrence of inflammatory processes and develop methods to maintain a balance between anticoagulation reversal and the retention of anti-inflammatory benefits. This thorough approach will offer useful insights for optimizing therapy procedures and enhancing patient care.
